# VARIANCE REDUCED HALPERN ITERATION FOR FINITE-SUM MONOTONE INCLUSIONS

**Xufeng Cai**[*1]    **Ahmet Alacaoglu**[*2]    **Jelena Diakonikolas**[1]

[1]Department of Computer Sciences, University of Wisconsin–Madison
[2]Wisconsin Institute for Discovery, University of Wisconsin–Madison
xcai74@wisc.edu, alacaoglu@wisc.edu, jelena@cs.wisc.edu

## ABSTRACT

Machine learning approaches relying on such criteria as adversarial robustness or multi-agent settings have raised the need for solving game-theoretic equilibrium problems. Of particular relevance to these applications are methods targeting finite-sum structure, which generically arises in empirical variants of learning problems in these contexts. Further, methods with computable approximation errors are highly desirable, as they provide verifiable exit criteria. Motivated by these applications, we study finite-sum monotone inclusion problems, which model broad classes of equilibrium problems. Our main contributions are variants of the classical Halpern iteration that employ variance reduction to obtain improved complexity guarantees in which $n$ component operators in the finite sum are *on average* either cocoercive or Lipschitz continuous and monotone, with parameter $L$. The resulting oracle complexity of our methods, which provide guarantees for the last iterate and for the (computable) operator norm residual, is $\widetilde{\mathcal{O}}(n + \sqrt{n}L\varepsilon^{-1})$, improving upon existing methods by a factor up to $\sqrt{n}$. This constitutes the first variance reduction-type result for general finite-sum monotone inclusions and for specific problems as convex-concave optimization when operator norm residual is the optimality measure. We further argue that, up to poly-logarithmic factors, this complexity is unimprovable in the monotone Lipschitz setting; i.e., the provided result is near-optimal.

## 1 INTRODUCTION

We study finite-sum monotone inclusion problems, where the goal is to find $\mathbf{u}_* \in \mathbb{R}^d$ such that

$$\mathbf{0} \in F(\mathbf{u}_*) + G(\mathbf{u}_*), \tag{MI}$$

and where $F(\mathbf{u}) \colon \mathbb{R}^d \to \mathbb{R}^d$ is monotone and Lipschitz, and $G(\mathbf{u}) \colon \mathbb{R}^d \rightrightarrows \mathbb{R}^d$ is maximally monotone and possibly multi-valued. We consider the *finite-sum* structure, i.e., $F(\mathbf{u}) = \frac{1}{n}\sum_{i=1}^{n} F_i(\mathbf{u})$.

As is standard, we assume access to (*i*) the resolvent of $\eta G$ for $\eta > 0$, meaning that for any $\mathbf{u}$ we can find $\bar{\mathbf{u}}$ such that $\mathbf{u} - \bar{\mathbf{u}} \in \eta G(\bar{\mathbf{u}})$ (generalizing the *proximal operator*); and (*ii*) evaluations of component operators $F_i$. We measure oracle complexity of an algorithm in terms of evaluations of component operators $F_i$ and the resolvent operator of $\eta G$.

The considered finite-sum settings are widespread in machine learning; see e.g., Johnson & Zhang (2013); Defazio et al. (2014); Schmidt et al. (2017); Gower et al. (2020). While finite-sum *minimization* problems are well-studied, recent applications in areas such as generative adversarial networks (Goodfellow et al., 2014), robust machine learning (Madry et al., 2018), and multi-agent reinforcement learning (Zhang et al., 2021) require solving more general equilibrium problems. Such tasks are neatly unified under the umbrella of monotone inclusion (MI), which has a rich history within optimization theory and operations research (Facchinei & Pang, 2003).

An important special case of (MI) is the *variational inequality (VI)* problem defined as below, where $G$ is the subdifferential of the indicator function of a closed convex set $C \subseteq \mathbb{R}^d$:

$$\text{Find } \mathbf{u}_* \in C \text{ such that } \langle F(\mathbf{u}_*), \mathbf{u} - \mathbf{u}_* \rangle \geq 0 \;\; \forall \mathbf{u} \in C. \tag{VI}$$

---

[*]Equal contribution.

A more specialized template that results in interesting examples of VI and monotone inclusion problems is min-max optimization $\min_{\mathbf{x}} \max_{\mathbf{y}} f(\mathbf{x}) - g(\mathbf{y}) + \Phi(\mathbf{x}, \mathbf{y})$, where $\Phi$ is convex-concave with Lipschitz gradients and $f, g$ are proper convex lower semicontinuous (l.s.c.). This maps to (MI) by setting $\mathbf{u} = \begin{pmatrix} \mathbf{x} \\ \mathbf{y} \end{pmatrix}$, $F(\mathbf{u}) = \begin{pmatrix} \nabla_{\mathbf{x}} \Phi(\mathbf{x}, \mathbf{y}) \\ -\nabla_{\mathbf{y}} \Phi(\mathbf{x}, \mathbf{y}) \end{pmatrix}$, and $G(\mathbf{u}) = \begin{pmatrix} \partial f(\mathbf{x}) \\ \partial g(\mathbf{y}) \end{pmatrix}$.

**Optimality measures.** A standard optimality measure for solving (VI) problems is the *duality gap* (Facchinei & Pang, 2003), defined as

$$\mathrm{Gap}(\mathbf{u}) = \max_{\mathbf{v} \in C} \langle F(\mathbf{v}), \mathbf{u} - \mathbf{v} \rangle. \tag{Gap}$$

However, (Gap) has significant drawbacks especially when the domain $C$ is unbounded, which is often the case. A common way to get around this is to use the *restricted duality gap* (Nesterov, 2007), which requires the identification of a compact set including the iterates. However, such a set generally affects the constants in the convergence bounds, and the restricted duality gap is not as interpretable as the duality gap. Additional drawbacks are that (*i*) neither of these measures is efficiently computable in general, (*ii*) the guarantees for these measures are typically obtained for an *average* or the *best* iterate, and (*iii*) such duality gap guarantees are generally not possible to obtain outside of monotone operator (convex-concave in the case of min-max optimization) settings.

An alternative optimality measure, which we focus on in this paper and argue to be more general than the duality gap, is the *residual* defined as

$$\mathrm{Res}_{F+G}(\mathbf{u}) = \|F(\mathbf{u}) + g(\mathbf{u})\|, \tag{Res}$$

where $g(\mathbf{u}) \in G(\mathbf{u})$ and hence $\mathrm{dist}(F(\mathbf{u}) + G(\mathbf{u}), \mathbf{0}) = \min_{g(\mathbf{u}) \in G(\mathbf{u})} \|F(\mathbf{u}) + g(\mathbf{u})\| \leq \mathrm{Res}_{F+G}(\mathbf{u})$.

The complexity results for (Res) can be translated to duality gap, but not vice versa (see, e.g., Diakonikolas (2020, Section 1.2)). Moreover, this measure is in most cases computable since the algorithms typically have access to $F(\mathbf{u}) + g(\mathbf{u})$, which will become clearer in the sequel. Further, all our results are for the *last iterate* and the residual error guarantees are possible even for some classes of structured non-monotone operators.

**Context.** When restricted to the complexity results in terms of the duality gap, there exist optimal algorithms for finite-sum monotone VIs (Palaniappan & Bach, 2016; Carmon et al., 2019; Alacaoglu & Malitsky, 2022). These works show how to take advantage of the finite-sum structure with variance reduction techniques to improve the complexity compared to deterministic algorithms such as the extragradient method. However, as described earlier, these results do not translate to guarantees on the residual. Even more, we cannot expect these existing variance reduced algorithms to have optimal rates for the residual, since in the deterministic case they reduce to algorithms that are known to be suboptimal for residual guarantees (Golowich et al., 2020).

On the other hand, even in the deterministic case, algorithms that are optimal for the residual error (in terms of oracle/iteration complexity) were obtained only recently (Diakonikolas, 2020; Yoon & Ryu, 2021). These results are based on variants of the classical Halpern iteration (Halpern, 1967) developed for solving fixed point equations with nonexpansive maps. Despite further developments relaxing the initial assumptions on $F$ and $G$ (Tran-Dinh & Luo, 2021; Lee & Kim, 2021; Cai et al., 2022b; Cai & Zheng, 2023; Kovalev & Gasnikov, 2022) and addressing stochastic approximation settings (Cai et al., 2022a; Chen & Luo, 2022), none of the existing results consider the finite-sum structure nor lead to the $\sqrt{n}$ improvements expected from variance reduction approaches in such settings.

**Our contributions.** We obtain the first variance-reduced complexity results for standard classes of (MI) problems that could lead to a $\sqrt{n}$ improvement compared to methods without variance reduction, in line with similar results obtained for the less general minimization and min-max problems (focusing only on the duality gap guarantees) in prior work; see Table 1 in App. A. In particular:

- When $F$ satisfies average $\frac{1}{L}$-cocoercivity (see Assumption 3), we obtain an algorithm with oracle complexity $\widetilde{\mathcal{O}}(n + \sqrt{n}L\varepsilon^{-1})$. To obtain this result, we incorporate recursive variance reduction (Li et al., 2021) into constrained one-step Halpern iteration. While a similar strategy has been employed in Cai et al. (2022a) to address stochastic approximation (infinite sum) settings, their analysis is more complicated and their oracle complexity is strictly worse than ours whenever $n$ is not too large (roughly, when $n = o(1/\varepsilon^3)$).

- When $F$ is monotone and $L$-Lipschitz in expectation (see Assumption 2), we obtain an algorithm with oracle complexity $\widetilde{\mathcal{O}}(n + \sqrt{n}L\varepsilon^{-1})$. This result is enabled by a variant of Halpern iteration employing inexact resolvent evaluations of $\eta(F + G)$ for $\eta > 0$. While this strategy is similar to the approach taken in Diakonikolas (2020) to address corresponding settings without the finite-sum considerations, unlike their work, our result is enabled by employing a stochastic variance reduced algorithm from Alacaoglu & Malitsky (2022). A critical difference is that we carry out a stochastic error analysis with a new inexactness criterion, due to the randomized nature of the inner algorithm. To obtain our result, we generalize the analysis for strongly monotone inclusion from Alacaoglu & Malitsky (2022) to the composite form (MI) with a maximal monotone $G$.

The complexity results presented above are for the expected residual in view of (Res). Due to the *computability* of the residual, our results can also be easily translated to hold in high probability with logarithmic dependence on confidence parameter. See the discussion after Remark 4.4 for details.

**Further related work.** Monotone inclusion and fixed point problems with finite-sum cocoercive operators have been the focus of study in several recent papers. Davis (2023) presented a possible speedup with variance reduction for root-finding problems with average cocoercivity only w.r.t. the solution point, but requiring additional quasi strong monotonicity assumption. Loizou et al. (2021); Gorbunov et al. (2022); Beznosikov et al. (2023) used similar assumptions to solve the VI problems and provided convergence results for stochastic gradient descent-ascent methods. A more general expected residual assumption was considered in Choudhury et al. (2023), but this work only proved $\mathcal{O}(\varepsilon^{-4})$ complexity for residual norm guarantees in our setting, which is suboptimal for finite-sum monotone problems when $n = o(\varepsilon^{-4})$. Similar less desirable $\mathcal{O}(\varepsilon^{-2})$ complexity on the residual norm was also obtained in Morin & Giselsson (2022) with component cocoercivity. Tran-Dinh & Luo (2023) considered random coordinate methods for root finding problems with cocoercivity, which is a different setting that does not improve upon the overall complexity over deterministic algorithms. For the finite-sum monotone Lipschitz setting, Johnstone et al. (2024) obtained $\mathcal{O}(\varepsilon^{-8})$ complexity for a generalized version of the residual and left open the problem of obtaining a better complexity for the residual norm by a stochastic method in this setting, which our results address.

## 2 PRELIMINARIES

We denote as $\|\cdot\|$ the $\ell_2$ norm. We say that an operator $T\colon \mathbb{R}^d \to \mathbb{R}^d$ is *(i) monotone* if for $\forall \mathbf{u}, \mathbf{v} \in \mathbb{R}^d\colon \langle T(\mathbf{u}) - T(\mathbf{v}), \mathbf{u} - \mathbf{v}\rangle \geq 0$; *(ii) $L_F$-Lipschitz* if for $\forall \mathbf{u}, \mathbf{v} \in \mathbb{R}^d\colon \|T(\mathbf{u}) - T(\mathbf{v})\| \leq L_F\|\mathbf{u} - \mathbf{v}\|$; *(iii) $\frac{1}{L}$-cocoercive* if for $\forall \mathbf{u}, \mathbf{v} \in \mathbb{R}^d\colon \langle T(\mathbf{u}) - T(\mathbf{v}), \mathbf{u} - \mathbf{v}\rangle \geq \frac{1}{L}\|T(\mathbf{u}) - T(\mathbf{v})\|^2$. Monotonicity can be defined in the standard way for a multi-valued operator $T\colon \mathbb{R}^d \rightrightarrows \mathbb{R}^d$. Note that $\frac{1}{L}$-cocoercivity implies monotonicity and $L$-Lipschitzness, but not vice versa. *Maximal monotone* operators are those whose graph is not properly contained in the graph of any other monotone operator where graph is defined in the standard way for an operator $T$ as $\mathrm{gra}T = \{(\mathbf{x}, \mathbf{y})\colon \mathbf{y} \in T(\mathbf{x})\}$. Common examples for this class include subdifferentials of proper convex l.s.c. functions. For further discussion on these properties, see Bauschke & Combettes (2011).

Given an operator $T$, its *resolvent* is defined as $J_{\eta T} = (\mathrm{Id} + \eta T)^{-1}$ for some $\eta > 0$, i.e.,

$$\bar{\mathbf{u}} \in J_{\eta T}(\mathbf{u}) \iff \frac{1}{\eta}(\mathbf{u} - \bar{\mathbf{u}}) \in T(\bar{\mathbf{u}}).$$

Important instances of resolvents include the proximal operator obtained when $T = \partial g$ for a convex function $g$ and projection $P_C$ obtained when $T = \partial \delta_C$ for the indicator function $\delta_C$ of a closed convex set $C$. An important and useful property of the resolvent operator $J_{\eta T}$ is that it is single valued and nonexpansive (1-Lipschitz) when $T$ is maximally monotone.

Our work leverages the classical Halpern iteration (Halpern, 1967), commonly used for solving fixed point equations $\mathbf{x} = T(\mathbf{x})$ with nonexpansive operators $T$. Halpern iteration is defined by

$$\mathbf{u}_{k+1} = \lambda_k \mathbf{u}_0 + (1 - \lambda_k)T(\mathbf{u}_k), \tag{Hal}$$

where $\lambda_k$ is a step parameter typically set to be of the order-$\frac{1}{k}$. To address (MI), several variants of (Hal) have been proposed, with different choices of the nonexpansive operator $T$. Notable examples most relevant to our work include $T = J_{\eta G} \circ (\mathrm{Id} - \eta F)$ for cocoercive settings and

$T = J_{\eta(F+G)}$ for monotone Lipschitz settings (Diakonikolas, 2020). We defer other background information about the techniques used in the paper to Appendix A, due to space constraints.

We start with the common standard assumption that we require in all of the results.

**Assumption 1.** The operator $F\colon \mathbb{R}^d \to \mathbb{R}^d$ is monotone and $L_F$-Lipschitz, and the operator $G\colon \mathbb{R}^d \rightrightarrows \mathbb{R}^d$ is maximally monotone. Their solution set is nonempty, i.e., $(F + G)^{-1}(\mathbf{0}) \neq \emptyset$.

The next two assumptions characterize the two separate settings we consider in the sequel.

**Assumption 2.** The operator $F\colon \mathbb{R}^d \to \mathbb{R}^d$ is $L_Q$-Lipschitz in expectation, meaning that given an oracle $F_\xi$ and distribution $Q$ such that $\mathbb{E}_{\xi \sim Q}[F_\xi(\mathbf{u})] = F(\mathbf{u})$, we have for any $\mathbf{u}, \mathbf{v} \in \mathbb{R}^d$,

$$\mathbb{E}_{\xi \sim Q}\|F_\xi(\mathbf{u}) - F_\xi(\mathbf{v})\|^2 \leq L_Q^2 \|\mathbf{u} - \mathbf{v}\|^2.$$

This is the main requirement used in Section 4. This assumption holds, for example, when each $F_i$ is Lipschitz-continuous, and is standard for analyzing variance reduced algorithms (see e.g., Palaniappan & Bach (2016); Carmon et al. (2019); Alacaoglu & Malitsky (2022) and also Table 1).

Alternatively, in Section 3 we assume that $F$ is cocoercive on average, which can be regarded as cocoercivity in expectation with uniform sampling.

**Assumption 3.** The operator $F\colon \mathbb{R}^d \to \mathbb{R}^d$ is $\frac{1}{L}$-cocoercive on average, i.e., for any $\mathbf{u}, \mathbf{v} \in \mathbb{R}^d$

$$\langle F(\mathbf{u}) - F(\mathbf{v}), \mathbf{u} - \mathbf{v} \rangle \geq \tfrac{1}{nL} \sum_{i=1}^n \|F_i(\mathbf{u}) - F_i(\mathbf{v})\|^2.$$

This assumption holds, for example, when each $F_i$ is cocoercive. In the minimization case, this corresponds to the smoothness of component functions, which is standard in variance reduction literature (Allen-Zhu, 2017; Nguyen et al., 2017). In the case of fixed point problems, it is implied by the nonexpansiveness of component operators. An example of this case is given as a convex feasibility problem in Malitsky (2019, Section 5.2). Similar assumptions also arise in Davis (2023); Morin & Giselsson (2022); Tran-Dinh & Luo (2023); Loizou et al. (2021).

**Oracle complexity.** As the standard convention for finite-sum problems (Palaniappan & Bach, 2016; Carmon et al., 2019; Alacaoglu & Malitsky, 2022), we measure the oracle complexity of an algorithm by the number of calls to $F_i$ to make an optimality measure small (the number of calls to the resolvent of $\eta G$ for $\eta > 0$ is of the same order). Since our variance reduced estimators compute the full operator values with some probability, per-iteration costs are random and our complexity results are on the *expected* number of calls to $F_i$. This is also a standard way to measure complexity with single-loop variance reduced algorithms (Li et al., 2021; Kovalev et al., 2020). It is possible to obtain deterministic per iteration costs by exchanging to multi-loop algorithms (Carmon et al., 2019; Alacaoglu & Malitsky, 2022), which we avoid for simplicity.

## 3 Algorithm and Analysis in the Cocoercive Case

In this section, we work under Assumption 3. The main reason that we study this case separately is because we can provide a simpler single-loop algorithm under cocoercivity. We use the following stochastic variant of the constrained Halpern iteration

$$\mathbf{u}_{k+1} = J_{\eta G}\big(\lambda_k \mathbf{u}_0 + (1 - \lambda_k)\mathbf{u}_k - \eta \widetilde{F}(\mathbf{u}_k)\big), \tag{3.1}$$

where $\widetilde{F}$ is the variance-reduced PAGE estimator (Li et al., 2021). We summarize our approach in Alg. 1, and defer the details of the PAGE estimator to Appendix A, due to space constraints.

A similar constrained Halpern iteration scheme has been analyzed in Cai et al. (2022b); Kovalev & Gasnikov (2022) with an extrapolation step, but only for deterministic settings. For the stochastic counterpart, Alg. 1 can be seen as a simpler constrained version of Cai et al. (2022a), with a single parameter for the constant batch size that (unlike in the algorithm of Cai et al. (2022a)), independent of the accuracy $\varepsilon$, the component variance, the norm of iterate differences, and a stage-wise choice of $p_k$. The reason we are able to simplify the batch size selection comes from our focus on the finite-sum problems, whereas Cai et al. (2022a) considered infinite-sum problems.

---

**Algorithm 1** Halpern iteration with variance reduction

---

**Input:** $\mathbf{u}_0 \in \mathbb{R}^d$, step size $\eta = \frac{1}{4L}$, batch size $b = \lceil \sqrt{n} \rceil$, $\lambda_1 = \frac{2}{5}$
$\mathbf{u}_1 = J_{\frac{\eta}{2\lambda_1}G}\big(\mathbf{u}_0 - \frac{\eta}{2\lambda_1}F(\mathbf{u}_0)\big), \quad \widetilde{F}(\mathbf{u}_1) = F(\mathbf{u}_1)$

**for** $k = 1, 2, \ldots$ **do**

$$\lambda_k = \frac{2}{k+4}, \quad p_{k+1} = \begin{cases} \frac{4}{k+5} & \forall k \leq \sqrt{n} \\ \frac{4}{\sqrt{n}+5} & \forall k \geq \sqrt{n} \end{cases}$$

$\mathbf{u}_{k+1} = J_{\eta G}(\lambda_k \mathbf{u}_0 + (1 - \lambda_k)\mathbf{u}_k - \eta \widetilde{F}(\mathbf{u}_k))$
Sample $\mathcal{S}_{k+1} \subseteq \{1, \ldots, n\}$ without replacement and uniformly at random with $|\mathcal{S}_{k+1}| = b$
$$\widetilde{F}(\mathbf{u}_{k+1}) = \begin{cases} F(\mathbf{u}_{k+1}) & \text{w.p. } p_{k+1}, \\ \widetilde{F}(\mathbf{u}_k) + \frac{1}{b}\sum_{i \in \mathcal{S}_{k+1}}\big(F_i(\mathbf{u}_{k+1}) - F_i(\mathbf{u}_k)\big) & \text{w.p. } 1 - p_{k+1}. \end{cases}$$

---

The key technical ingredient in our analysis is the following lemma, which shows that, in expectation, in each iteration $k$ we can reduce a potential function by a factor $(1 - \lambda_k)$. This potential reduction, in turn, can be translated into the residual error decay at rate $\lambda_k = \mathcal{O}(1/k)$, as shown in Theorem 3.2 below. Due to space constraints, the proofs are deferred to Appendix B.

**Lemma 3.1.** *Let Assumptions 1 and 3 hold. Then, for the iterates $\mathbf{u}_k$ of Algorithm 1 and the potential function $\mathcal{C}_k$ defined by*

$$\mathcal{C}_k = \frac{\eta}{2\lambda_k}\|F(\mathbf{u}_k) + \mathbf{g}_k\|^2 + \langle F(\mathbf{u}_k) + \mathbf{g}_k, \mathbf{u}_k - \mathbf{u}_0 \rangle + c_k\|F(\mathbf{u}_k) - \widetilde{F}(\mathbf{u}_k)\|^2, \quad (3.2)$$

*we have that $\mathbb{E}[\mathcal{C}_{k+1}] \leq (1 - \lambda_k)\mathbb{E}[\mathcal{C}_k]$ for $k \geq 1$, where $\mathbf{g}_{k+1} = \frac{1}{\eta}\big(\lambda_k \mathbf{u}_0 + (1-\lambda_k)\mathbf{u}_k - \eta \widetilde{F}(\mathbf{u}_k) - \mathbf{u}_{k+1}\big) \in G(\mathbf{u}_{k+1})$ and $c_k = \frac{(\sqrt{n}+2)(k+4)}{4L}$.*

Our potential function in (3.2) allows us to go beyond the deterministic setting analyzed in Kovalev & Gasnikov (2022), by handling the variance of the estimator $\widetilde{F}$, which also helps us avoid the more complicated induction-based argument in Cai et al. (2022a). Another important aspect in the above bound is that $c_1$ can be of the order of $\sqrt{n}$. Hence, to make sure that we do not introduce spurious dependence on $n$, it is critical that the first two iterations of the algorithm evaluate the full operator.

The following theorem states our main convergence result for this section.

**Theorem 3.2.** *Let Assumptions 1 and 3 hold. Then, for the iterates $\mathbf{u}_k$ of Algorithm 1, we have*

$$\mathbb{E}[\text{Res}_{F+G}(\mathbf{u}_k)] \leq \big(\mathbb{E}[\text{Res}^2_{F+G}(\mathbf{u}_k)]\big)^{1/2} \leq \frac{16L\|\mathbf{u}_0 - \mathbf{u}_*\|}{k+4}.$$

*In particular, given accuracy $\varepsilon > 0$, to return a point $\mathbf{u}_K$ such that $\mathbb{E}[\text{Res}_{F+G}(\mathbf{u}_K)] \leq \varepsilon$, the stochastic oracle complexity of Algorithm 1 is $\widetilde{\mathcal{O}}\big(n + \frac{\sqrt{n}L\|\mathbf{u}_0 - \mathbf{u}_*\|}{\varepsilon}\big)$.*

Observe that we use a large batch size $|\mathcal{S}_k| \approx \sqrt{n}$ to obtain our improvement from the employed variance reduction strategy, which is a common practice for stochastic algorithms with residual guarantees (Cai et al., 2022a; Lee & Kim, 2021). Prior work (Pethick et al., 2023) that avoids a large batch size requires $\mathcal{O}(\varepsilon^{-4})$ complexity and only provides residual guarantees on the best iterate. We also argue that in the finite-sum case, there is no inherent disadvantage of using $\mathcal{O}(\sqrt{n})$ samples in every iteration since we provably show that this leads to a better dependence on $n$ in the final oracle complexity compared to deterministic algorithms, which would use $n$ samples every iteration.

To compare with prior results, we first note that deterministic Halpern iteration for constrained VIs with cocoercive operators yields $\widetilde{\mathcal{O}}(nL_F \varepsilon^{-1})$ complexity (Diakonikolas, 2020), for which our result in Theorem 3.2 replaces $nL_F$ with $\sqrt{n}L$ and can provide improvements up to $\sqrt{n}$ depending on the relationship between $L$ and $L_F$ (see examples in Palaniappan & Bach (2016); Carmon et al. (2019); Alacaoglu & Malitsky (2022)). Moreover, compared to complexity results $\mathcal{O}(L_F \varepsilon^{-3})$ and $\widetilde{\mathcal{O}}(L_F \varepsilon^{-2})$ for algorithms developed for the infinite-sum stochastic settings in Cai et al. (2022a); Chen & Luo (2022), we provide improvements in the regime $\varepsilon = o(1/\sqrt{n})$, assuming $L_F \approx L$.

An important implication of this result on cocoercive inclusions is for finite-sum minimization where variance reduction has been studied extensively. The state-of-the-art results with direct

algorithms are due to Lan et al. (2019) and Song et al. (2020), that provide oracle complexity $\widetilde{\mathcal{O}}(n + \sqrt{n}L\varepsilon^{-1}\|\mathbf{u}_0 - \mathbf{u}_*\|)$ for the objective suboptimality. This result can be translated to the norm of the prox-mapping to get $\mathbb{E}[\mathrm{Res}_{F+G}(\mathbf{u}_{\mathrm{out}})] \leq \varepsilon$ with complexity $\widetilde{\mathcal{O}}(n + \sqrt{n}L\|\mathbf{u}_0 - \mathbf{u}_*\|\varepsilon^{-1})$ which is the same as our result when specified to the case $F = \nabla f$ for a smooth convex function $f$ and $G = \partial g$ for a regularizer $g$. This complexity is not optimal for smooth convex minimization and has been improved for unconstrained minimization or with indirect algorithms (Zhou et al., 2022; Allen-Zhu, 2018). Our results provide the best-known guarantees (among direct approaches) with a single-loop algorithm. Single-loop versions of Katyusha (Allen-Zhu, 2017) were studied in Kovalev et al. (2020) and Qian et al. (2021), albeit without guarantees for the non-strongly convex case.

## 4    ALGORITHM AND ANALYSIS IN THE MONOTONE & LIPSCHITZ CASE

In this section, we consider the more standard setting where $F$ is monotone and $L_Q$-expected Lipschitz for an oracle distribution $Q$. We note that our results apply for general sampling distributions $Q$ under which Assumption 2 holds; for concrete examples of beneficial non-uniform sampling distributions, see Remark 4.4. We omit the subscript and denote $L = L_Q$ for brevity in this section, since the context is clear. To obtain the desired complexity, we make use of the resolvent operator $J_{\eta(F+G)}(\mathbf{u})$ for some fixed $\eta > 0$ (specified later in this section). In particular, we adapt the stochastic Halpern iteration to the following single-valued and cocoercive operator

$$P^\eta(\mathbf{u}) := \mathbf{u} - J_{\eta(F+G)}(\mathbf{u}).$$

Indeed, for any $\eta > 0$, finding a point $\mathbf{u}$ such that $\mathbb{E}[\|P^\eta(\mathbf{u})\|] \leq \eta\varepsilon$ is sufficient to approximate (MI) of $F + G$, as summarized in the following proposition with the proof deferred to Appendix C.

**Proposition 4.1.** *For any fixed $\eta > 0$, let $P^\eta(\mathbf{u}) = \mathbf{u} - J_{\eta(F+G)}(\mathbf{u})$. If $\|P^\eta(\mathbf{u})\| \leq \eta\varepsilon$ for some $\varepsilon > 0$, then we have $\mathrm{Res}_{F+G}(\bar{\mathbf{u}}) \leq \varepsilon$ with $\bar{\mathbf{u}} = \mathbf{u} - P^\eta(\mathbf{u}) = J_{\eta(F+G)}(\mathbf{u})$.*

This standard proposition gives us a simple way to convert the guarantees on $\|P^\eta(\mathbf{u})\|$ to the residual norm (Res) conceptually, and we later provide a computable approximation for $\bar{\mathbf{u}}$ in Cor. 4.3. If $P^\eta$ can be computed exactly, then (MI) can be solved by the standard, deterministic Halpern iteration, as $P^\eta(\mathbf{u})$ is nonexpansive. However, the exact evaluation of resolvent operators only happens in special cases, and even for those cases, the computation is usually prohibitive when $n$ is large for the operator $F$. Instead, one can efficiently approximate the resolvent by solving a finite-sum strongly monotone VI problem, for which we provide more details in Section 4.2.

In the rest of the section, we provide an overview of the analysis and main technical results. Due to space constraints, the proofs are deferred to Appendix C.

### 4.1    INEXACT HALPERN ITERATION WITH STOCHASTIC ERROR

Denoting the resolvent approximation by $\widetilde{J}_{\eta(F+G)}$, we use the inexact Halpern iteration as follows

$$\mathbf{u}_{k+1} = \lambda_k \mathbf{u}_0 + (1 - \lambda_k)\widetilde{J}_{\eta(F+G)}(\mathbf{u}_k) = \lambda_k \mathbf{u}_0 + (1 - \lambda_k)(\mathbf{u}_k - P^\eta(\mathbf{u}_k)) - (1 - \lambda_k)\mathbf{e}_k, \quad (4.1)$$

where $\mathbf{e}_k = J_{\eta(F+G)}(\mathbf{u}_k) - \widetilde{J}_{\eta(F+G)}(\mathbf{u}_k)$ is the approximation error. To efficiently compute $\widetilde{J}_{\eta(F+G)}$ to a certain accuracy, we use the variance-reduced forward-reflected-backward method (VR−FoRB, Alg. 3) proposed in Alacaoglu & Malitsky (2022) as our subsolver for each iteration. We summarize our approach in Alg. 2, and defer our detailed discussion of VR−FoRB to Section 4.2.

Halpern iteration with inexact resolvent computation has been shown to maintain $\mathcal{O}(1/k)$ convergence rate for *deterministic* problems (Diakonikolas, 2020), provided the approximation error $\|\mathbf{e}_k\|$ is sufficiently small. The critical difference is that we can no longer use the stopping criterion for the inner algorithm therein, due to the randomized nature of VR−FoRB. Their inexactness criterion $\|\mathbf{e}_k\| \leq \frac{\varepsilon}{4k(k+1)}$ for each iteration $k$ requires a pre-fixed accuracy $\varepsilon$ and also leads to the bound on the number of inner iterations to depend on $J_{\eta(F+G)}(\mathbf{u}_k)$ which is not feasible empirically. The latter is simply because the initial distance to the solution of the subproblem appears in the complexity bound (see, e.g., Theorem 4.6). To overcome these issues, we use a different inexactness criterion $\mathbb{E}_k[\|\mathbf{e}_k\|^2] \leq \frac{\|P^\eta(\mathbf{u}_k)\|}{(k+2)^8}$, conditional on the algorithmic randomness up to and including iteration $k$, which is, in fact, related to an old idea of warm-starting (Rockafellar, 1976, eq. (B)). Such

---

**Algorithm 2** Inexact Halpern iteration with VR−FoRB

---

**Input:** $\mathbf{u}_0 \in \mathbb{R}^d$, $L = L_Q$ with the distribution $Q = \{q_i\}_{i=1}^n$, $n$, $\eta = \frac{\sqrt{n}}{L}$

    **for** $k = 0, 1, 2, \ldots$ **do**

        $\lambda_k = \frac{1}{k+2}$, $M_k = \lceil 56(n + \sqrt{n}) \log(2k + 4) \rceil$

        $\widetilde{J}_{\eta(F+G)}(\mathbf{u}_k) = \text{VR−FoRB}(\mathbf{u}_k, M_k, \text{Id} + \eta(F + G) - \mathbf{u}_k, Q)$

        $\mathbf{u}_{k+1} = \lambda_k \mathbf{u}_0 + (1 - \lambda_k)\widetilde{J}_{\eta(F+G)}(\mathbf{u}_k)$

---

a criterion can be guaranteed to hold by setting the number of inner iterations to be a sufficiently large, yet computable, number depending only on known constants, using the convergence results of VR−FoRB from Section 4.2. We summarize these results in the following theorem.

**Theorem 4.2.** *Let Assumptions 1 and 2 hold. Then, for the iterates $\mathbf{u}_k$ of Algorithm 2, we have that* $\mathbb{E}_k[\|\mathbf{e}_k\|^2] \leq \frac{\|P^\eta(\mathbf{u}_k)\|}{(k+2)^8}$ *conditional on the algorithm randomness up to iteration $k$, and*

$$\mathbb{E}[\|P^\eta(\mathbf{u}_k)\|] \leq \left(\mathbb{E}[\|P^\eta(\mathbf{u}_k)\|^2]\right)^{1/2} \leq \frac{7\|\mathbf{u}_0 - \mathbf{u}_*\|}{k}.$$

*Moreover, given accuracy $\varepsilon > 0$, to return a point $\mathbf{u}_K$ such that $\mathbb{E}[\|P^\eta(\mathbf{u}_K)\|] \leq \eta\varepsilon$ with $\eta = \frac{\sqrt{n}}{L}$, the stochastic oracle complexity is $\tilde{\mathcal{O}}\left(n + \frac{\sqrt{n}L\|\mathbf{u}_0 - \mathbf{u}_*\|}{\varepsilon}\right)$.*

The final step is to characterize the precise point at which we have the residual guarantees.

**Corollary 4.3.** *Let Assumptions 1 and 2 hold and let $\mathbf{u}_K$ be as defined in Theorem 4.2. Then, for* $\mathbf{u}_{\text{out}} = \text{VR−FoRB}(\mathbf{u}_K, \lceil 42(n + \sqrt{n}) \log(19n) \rceil, \text{Id} + \eta(F + G) - \mathbf{u}_K, Q)$ *with $\eta = \frac{\sqrt{n}}{L}$,*

$$\mathbb{E}[\text{Res}_{F+G}(\mathbf{u}_{\text{out}})] \leq 2\varepsilon.$$

*The total stochastic oracle complexity for producing $\mathbf{u}_{\text{out}}$ is $\widetilde{\mathcal{O}}\left(n + \frac{\sqrt{n}L\|\mathbf{u}_0 - \mathbf{u}_*\|}{\varepsilon}\right)$.*

**Remark 4.4.** Non-uniform sampling $Q$ can be beneficial in terms of lowering the Lipschitz constant $L_Q$ and thus the overall algorithm complexity. As a specific example, consider the matrix game

$$\min_{\mathbf{x} \in \mathbb{R}^{m_1}} \max_{\mathbf{y} \in \mathbb{R}^{m_2}} \langle \boldsymbol{A}\mathbf{x}, \mathbf{y} \rangle + \delta_{\Delta^{m_1}}(\mathbf{x}) + \delta_{\Delta^{m_2}}(\mathbf{y})$$

for $\boldsymbol{A} \in \mathbb{R}^{m_2 \times m_1}$, the simplices $\Delta^{m_1}$, $\Delta^{m_2}$, where $\delta$ is the indicator function. With $\mathbf{u} = \binom{\mathbf{x}}{\mathbf{y}}$, we have $F(\mathbf{u}) = \binom{\boldsymbol{A}^\top \mathbf{y}}{-\boldsymbol{A}\mathbf{x}}$ and $G(\mathbf{u}) = \binom{\partial\delta_{\Delta^{m_1}}(\mathbf{x})}{\partial\delta_{\Delta^{m_2}}(\mathbf{y})}$. Denote the $i$-th row and the $j$-th column of $\boldsymbol{A}$ by $\boldsymbol{A}_{i:}$ and $\boldsymbol{A}_{:j}$, respectively. Let $\|\cdot\|_2$ be the operator norm for a matrix, and $\|\cdot\|_F$ be its Frobenius norm. Consider the standard importance sampling for $Q$, i.e., we sample $\xi = (i, j) \sim Q$ such that

$$F_\xi(\mathbf{u}) = \begin{pmatrix} \frac{1}{q_i^{(1)}} \boldsymbol{A}_{i:}\mathbf{y}_i \\ -\frac{1}{q_j^{(2)}} \boldsymbol{A}_{:j}\mathbf{x}_j \end{pmatrix}, \quad \mathbb{P}_Q[\xi = (i, j)] = q_i^{(1)} q_j^{(2)}, \quad q_i^{(1)} = \frac{\|\boldsymbol{A}_{i:}\|_2^2}{\|\boldsymbol{A}\|_F^2}, \quad q_j^{(2)} = \frac{\|\boldsymbol{A}_{:j}\|_2^2}{\|\boldsymbol{A}\|_F^2}.$$

It is easy to verify that $\mathbb{E}_{\xi \sim Q}[F_\xi(\mathbf{u})] = F(\mathbf{u})$, and we have $L_Q = \|A\|_F$ while $L_F = \|A\|_2$. Since it is possible for $\|A\|_F$ and $\|A\|_2$ to be of the same order, in those cases the improvement from the variance reduction approaches (including ours) is of the order $\sqrt{\frac{2m_1 m_2}{m_1 + m_2}}$ (order $\sqrt{m_1}$ for square matrices). Similar conclusions can be drawn more generally for linearly constrained nonsmooth convex optimization problems; see (Alacaoglu & Malitsky, 2022, Section 4) and App. C.5 for details.

In addition to complexity bounds for the expected residual, our results also have a direct consequence for high probability guarantees. In particular, since our result clearly implies $\text{Res}_{F+G}(\mathbf{u}_{\text{out}}) \leq \varepsilon$ with a constant probability by Markov's inequality and since the residual is computable, one can run the algorithm multiple times and monitor the residual, to obtain a high probability guarantee with logarithmic dependence on the confidence level. See, for example, Zhou et al. (2022); Allen-Zhu (2018) where such a confidence boosting mechanism is used in a similar context.

A few other remarks are in order here. First, our results imply the gap guarantee results in prior work (Alacaoglu & Malitsky, 2022; Carmon et al., 2019). On the other hand, the algorithms in these

works are bound to be suboptimal for the residual since they reduce to the exragradient algorithm in the case $n = 1$, which is suboptimal for the residual guarantee (Golowich et al., 2020). Second, residual guarantees for these variance reduced algorithms are currently unknown. Third, the implication to gap guarantees also ensures the near-optimality of our results since such complexity is known to be unimprovable for the gap guarantees (Alacaoglu & Malitsky, 2022; Han et al., 2024).

Next, compared to deterministic algorithms with $\tilde{\mathcal{O}}(nL_F\varepsilon^{-1})$ complexity for the residual (Diakonikolas, 2020; Yoon & Ryu, 2021), we replace $nL_F$ with $\sqrt{n}L$. This brings improvements in important cases discussed in Palaniappan & Bach (2016); Carmon et al. (2019); Alacaoglu & Malitsky (2022), including linearly constrained problems and matrix games discussed in Remark 4.4. This mirrors the recent improvements for the duality gap guarantees for VIs where Alacaoglu & Malitsky (2022); Carmon et al. (2019) showed $O(n + \sqrt{n}L\varepsilon^{-1})$ complexity instead of $\mathcal{O}(nL_F\varepsilon^{-1})$ of deterministic methods (Nemirovski, 2004).

Finally, we show the extension to cohypomonotone settings defined by the existence of $\rho > 0$ such that $\langle F(\mathbf{u}) - F(\mathbf{v}), \mathbf{u} - \mathbf{v} \rangle \geq -\rho\|F(\mathbf{u}) - F(\mathbf{v})\|^2$ in the following corollary for completeness, with justifications in Appendix C.4. This result provides a better dependence on $n$ compared to previous results with the drawback of a more restrictive bound for $\rho$ (roughly, $\rho < \frac{L}{\sqrt{n}L_F^2}$) and using $G \equiv \mathbf{0}$.

**Corollary 4.5.** *[Cohypomonotone] Assume that $F$ is maximally $\rho$-cohypomonotone and $L$-expected Lipschitz and $G \equiv \mathbf{0}$. Given $\varepsilon > 0$, Alg. 2 returns a point $\mathbf{u}_K$ such that $\left(\mathbb{E}[\|P^\eta(\mathbf{u}_K)\|^2]\right)^{1/2} \leq \eta\varepsilon$ with $\widetilde{O}\left(\left(n + \sqrt{n}\frac{\eta L+1}{1-\rho\eta L_F^2}\right)\left(\frac{\|\mathbf{u}_0 - \mathbf{u}_*\|}{\eta\varepsilon} + 1\right)\right)$ stochastic oracle complexity, for any positive $\eta$ such that $\rho < \min\left(\frac{\eta}{2}, \frac{1}{\eta L_F^2}\right)$. With $\eta = \frac{\sqrt{n}}{L}$ as before, this corresponds to $\rho < \min\left(\frac{L}{\sqrt{n}L_F^2}, \frac{\sqrt{n}}{2L}\right)$.*

### 4.2 RANDOMIZED APPROXIMATION OF THE RESOLVENT

Approximating $J_{\eta(F+G)}(\mathbf{u}^+)$ for $\mathbf{u}^+ \in \mathbb{R}^d$ corresponds to solving the finite-sum MI defined as:

$$\text{Find } \bar{\mathbf{u}} \text{ such that} \quad \mathbf{0} \in \eta F(\bar{\mathbf{u}}) + \eta G(\bar{\mathbf{u}}) + \bar{\mathbf{u}} - \mathbf{u}^+. \tag{4.2}$$

It is immediate that the solution to (4.2) of the operator $\eta(F + G) + \text{Id} - \mathbf{u}^+$ corresponds to $J_{\eta(F+G)}(\mathbf{u}^+)$ by the definition of the resolvent. Note that $\eta(F + G) + \text{Id} - \mathbf{u}^+$ can be represented as a sum of two operators $\bar{F}^\eta(\mathbf{u}; \mathbf{u}^+)$ and $\eta G(\mathbf{u})$, where

$$\bar{F}^\eta(\mathbf{u}; \mathbf{u}^+) := \eta F(\mathbf{u}) + \mathbf{u} - \mathbf{u}^+ = \frac{1}{n}\sum_{i=1}^{n}\bar{F}_i^\eta(\mathbf{u}; \mathbf{u}^+), \quad \bar{F}_i^\eta(\mathbf{u}; \mathbf{u}^+) := \eta F_i(\mathbf{u}) + \mathbf{u} - \mathbf{u}^+. \tag{4.3}$$

It is simple to verify that $\bar{F}^\eta(\mathbf{u}; \mathbf{u}^+)$ is 1-strongly monotone and $(\eta L + 1)$-average Lipschitz w.r.t. $\mathbf{u}$; we provide a proof in Appendix C.1 for completeness, and $\eta G$ is still maximally monotone as $\eta > 0$. Hence below we use a more general notation and will set $A(\mathbf{u}) = \bar{F}^\eta(\mathbf{u}; \mathbf{u}^+)$ and $B = \eta G$. As mentioned before, we use VR$-$FoRB from (Alacaoglu & Malitsky, 2022, Algorithm 4) to solve (4.2), which we present as Alg. 3. We now state its convergence result under strong monotonicity.

---

**Algorithm 3** VR$-$FoRB($\mathbf{u}, M, A + B, Q$) (Alacaoglu & Malitsky, 2022, Algorithm 4)

---

**Input:** $\mathbf{v}_0 = \mathbf{w}_0 = \mathbf{w}_{-1} = \mathbf{u}$, $p = \frac{1}{n}$, $\alpha = 1 - p$, $\tau = \frac{\sqrt{p(1-p)}}{2L_A}$, distribution $Q = \{q_i\}_{i=1}^{n}$

**for** $k = 0, 1, \dots, M - 1$ **do**
$\quad \hat{\mathbf{v}}_k = \alpha\mathbf{v}_k + (1 - \alpha)\mathbf{w}_k$
$\quad$Sample $i \in \{1, \dots, n\}$ according to $Q$
$\quad \mathbf{v}_{k+1} = J_{\tau B}(\hat{\mathbf{v}}_k - \tau[A(\mathbf{w}_k) - (nq_i)^{-1}A_i(\mathbf{w}_{k-1}) + (nq_i)^{-1}A_i(\mathbf{v}_k)])$
$\quad \mathbf{w}_{k+1} = \begin{cases} \mathbf{v}_{k+1} & \text{w.p. } p \\ \mathbf{w}_k & \text{w.p. } 1 - p \end{cases}$

---

**Theorem 4.6.** *Let $A\colon \mathbb{R}^d \to \mathbb{R}^d$ be monotone and $L_A$-Lipschitz in expectation with $A = \frac{1}{n}\sum_{i=1}^{n}A_i$. Let $B\colon \mathbb{R}^d \rightrightarrows \mathbb{R}^d$ be maximally monotone, and $A + B$ be $\mu$-strongly monotone with $\mu > 0$ and $\mathbf{v}_* := (A + B)^{-1}(\mathbf{0}) \neq \emptyset$. Given $\bar{\varepsilon} > 0$, Alg. 3 returns $\mathbf{v}_M$ with $\mathbb{E}[\|\mathbf{v}_M - \mathbf{v}_*\|^2] \leq \bar{\varepsilon}^2$ in $\lceil 14(n + \frac{\sqrt{n}L_A}{\mu})\log\frac{\sqrt{6}\|\mathbf{v}_0 - \mathbf{v}_*\|}{\bar{\varepsilon}}\rceil$ iterations and $\mathcal{O}\left((n + \frac{\sqrt{n}L_A}{\mu})\log\frac{\|\mathbf{v}_0 - \mathbf{v}_*\|}{\bar{\varepsilon}}\right)$ oracle queries.*

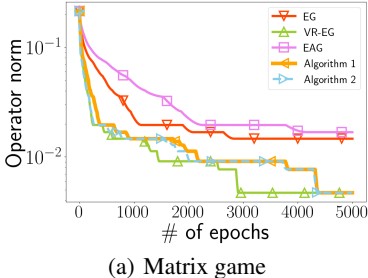 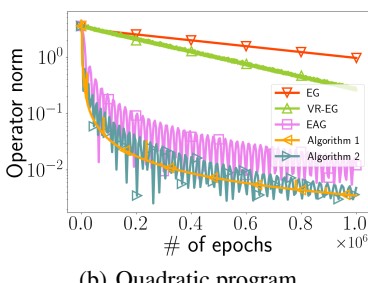

(a) Matrix game            (b) Quadratic program

Figure 1: Comparison of Alg. 1, Alg. 2, deterministic extragradient (EG), extra anchored gradient (EAG), variance reduced extragradient (VR-EG) on the matrix games and quadratic programming.

We remark that only almost sure convergence was proved for $\mathtt{VR-FoRB}$ in Alacaoglu & Malitsky (2022). We show its non-asymptotic linear convergence which is needed for our main result in Theorem 4.2. A similar rate is in (Alacaoglu & Malitsky, 2022, Theorem 6) for strongly monotone VIs, but for a different algorithm – variance reduced extragradient. Our result can be seen as a "single-call" alternative to this method, for the slightly more general setting of strongly monotone inclusions. We provide this to keep the generality of our setting and also for possible interest in its own right, since such algorithms have been popular (Cai & Zheng, 2023; Hsieh et al., 2019).

## 5 NUMERICAL EXPERIMENTS AND DISCUSSION

We provide preliminary numerical results for our proposed algorithms[1]. We compare Alg. 1 and Alg. 2 with existing algorithms on matrix games and a special quadratic program used for lower bound derivations in Ouyang & Xu (2021). We emphasize that our main contributions are theoretical, while the provided examples are mainly for illustration with two goals in mind: (*i*) verify our improved complexity bounds compared to deterministic Halpern-based methods (Cai et al., 2022b), (*ii*) show benefits of our algorithms compared to prior variance reduced algorithms (Alacaoglu & Malitsky, 2022) for *difficult* problem instances used for establishing lower bounds.

We compare our algorithms with extragradient (EG) (Korpelevich, 1977), constrained anchored extragradient (EAG) (Cai et al., 2022b), and variance-reduced extragradient (VR-EG) (Alacaoglu & Malitsky, 2022, Alg. 1). First problem is a matrix game with simplex constraints, i.e., $\min_{\mathbf{x} \in \Delta^{m_1}} \max_{\mathbf{y} \in \Delta^{m_2}} \langle A\mathbf{x}, \mathbf{y} \rangle$ with $m_1 = m_2 = 500$. We use the policeman and burglar matrix (Nemirovski, 2013). Second problem we consider is a quadratic program from Ouyang & Xu (2021) equivalent to the problem $\min_{\mathbf{x} \in \mathbb{R}^{m_1}} \max_{\mathbf{y} \in \mathbb{R}^{m_2}} \frac{1}{2}\mathbf{x}^\top H\mathbf{x} - h^\top \mathbf{x} - \langle A\mathbf{x} - b, \mathbf{y} \rangle$. This problem was used in Ouyang & Xu (2021) for establishing lower bounds for min-max optimization and we use this example with $m_1 = m_2 = 200$ to show the efficacy of Halpern-type algorithms. We use uniform sampling for all the algorithms, set $M_k = \lfloor 0.05n \log(k + 2) \rfloor$ for Alg. 2, and tune the stepsizes for each method individually. We implement all the algorithms in Python, and run the experiments on Google Colab standard CPU backend. We provide further details in App. D.

We summarize our numerical results and plot the operator norm against the number of epochs in Fig. 1, where one epoch means $n$ individual component operator evaluations. Operator norm stands for $\|F(\mathbf{u})\|$ for the unconstrained case, and we follow the convention and use the norm of gradient mapping, i.e., $\sqrt{\|\mathbf{x} - \mathcal{P}_{\Delta^{m_1}}(\mathbf{x} - A^\top \mathbf{y})\|^2 + \|\mathbf{y} - \mathcal{P}_{\Delta^{m_2}}(\mathbf{y} + A\mathbf{x})\|^2}$ for the matrix game (which our guarantees can be directly translated to, see for example (Cai et al., 2022b, Fact 1)). We observe that (*i*) our variance reduced Alg. 1 and Alg. 2 converge faster than deterministic methods in both cases, validating our complexity results; (*ii*) VR-EG exhibits slightly faster convergence than our Halpern-type algorithms in Fig. 1(a) (see similar empirical observations and comments in Park & Ryu (2022); Tran-Dinh (2023)), however VR-EG suffers stagnation while our algorithms progress on the difficult worst-case problem, as shown in Fig. 1(b).

Due to space constraints, conclusions and discussions about future directions appear in Appendix E.

---

[1] Code is available at https://github.com/zephyr-cai/Finite-Sum-Halpern-Iteration.

## ACKNOWLEDGMENTS

This research was supported in part by the NSF grant 2023239, the NSF grant 2007757, the NSF grant 2224213, the AFOSR award FA9550-21-1-0084, the Office of Naval Research under contract number N00014-22-1-2348.

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

Table 1: Comparison of our results with state of the art in the monotone Lipschitz settings, in terms of stochastic oracle complexity required to output $\mathbf{x}_{\text{output}}$ with $\mathbb{E}[\text{Res}_{F+G}(\mathbf{x}_{\text{output}})] \leq \epsilon$ in Column 2 and $\mathbb{E}[\text{Gap}(\mathbf{x}_{\text{output}})] \leq \epsilon$ in Column 3. We refer to Section 1 for the discussion regarding the difference in optimality measures $\text{Res}_{F+G}$ and Gap and the importance of getting results on $\text{Res}_{F+G}$.

| Reference | Complexity for $\text{Res}_{F+G}$ | Complexity for Gap | Assumption | High Probability Result |
|---|---|---|---|---|
| Kovalev & Gasnikov (2022) | $\mathcal{O}(nL_F\varepsilon^{-1})$ | $\mathcal{O}(nL_F\varepsilon^{-1})$ | Assumption 1 | N/A |
| Nemirovski (2004) | $\mathcal{O}(nL_F^2\varepsilon^{-2})$ | $\mathcal{O}(nL_F\varepsilon^{-1})$ | Assumption 1 | N/A |
| Cai et al. (2022a) | $\mathcal{O}((\sigma^2 L + L^3)\varepsilon^{-3})$ | $\mathcal{O}((\sigma^2 L + L^3)\varepsilon^{-3})$ | Assumption 1, 2, $G \equiv \mathbf{0}$ $\mathbb{E}_i\|F_i(\mathbf{x}) - F(\mathbf{x})\|^2 \leq \sigma^2$ | $-$ |
| Luo et al. (2021) | $\widetilde{\mathcal{O}}(\sigma^2\varepsilon^{-2} + L_F\varepsilon^{-1})$ | $\widetilde{\mathcal{O}}(\sigma^2\varepsilon^{-2} + L_F\varepsilon^{-1})$ | Assumption 1, $G \equiv \mathbf{0}$ $F = \begin{pmatrix} \nabla_{\mathbf{x}}\Phi(\mathbf{x},\mathbf{y}) \\ -\nabla_{\mathbf{y}}\Phi(\mathbf{x},\mathbf{y}) \end{pmatrix}$ $\mathbb{E}_i\|F_i(\mathbf{x}) - F(\mathbf{x})\|^2 \leq \sigma^2$ | $-$ |
| Carmon et al. (2019) | $-$ | $\widetilde{\mathcal{O}}(n + \sqrt{n}L\varepsilon^{-1})$ | Assumption 1, 2 bounded domain (cf. Sec 5.4 in (Carmon et al., 2019)) | $-$ |
| Palaniappan & Bach (2016) | $-$ | $\widetilde{\mathcal{O}}(n + \sqrt{n}L\varepsilon^{-1})$ | Assumption 1, 2 bounded domain (cf. (C) in Sec. 2 in (Palaniappan & Bach, 2016)) | $-$ |
| Alacaoglu & Malitsky (2022) | $-$ | $\mathcal{O}(n + \sqrt{n}L\varepsilon^{-1})$ | Assumption 1, 2 (cf. Assumption 1(iv) in (Alacaoglu & Malitsky, 2022)) | $-$ |
| [Our results, Theorem 4.2] | $\widetilde{\mathcal{O}}(n + \sqrt{n}L\varepsilon^{-1})$ | $\widetilde{\mathcal{O}}(n + \sqrt{n}L\varepsilon^{-1})$ | Assumption 1, 2 | ✓ |

# A  BACKGROUND

We first provide Table 1 for the comparison of our results with the state-of-the-art in the monotone Lipschitz case.

**VI algorithms, extragradient, FB, PP.**  Two of the most fundamental algorithms for solving VI problems with monotone operators and monotone inclusions are Forward-Backward (FB) and proximal-point (PP) algorithms, see for example (Facchinei & Pang, 2003; Rockafellar, 1976). For problem (MI), FB iterates as

$$\mathbf{u}_{k+1} = J_{\tau G}(\mathbf{u}_k - \tau F(\mathbf{u}_k)),$$

and converges when $F$ is cocoercive or under other restrictive assumptions such as strong monotonicity of $F + G$. On the other hand, PP, for the same problem, iterates as

$$\mathbf{u}_{k+1} = J_{\tau(F+G)}(\mathbf{u}_k),$$

and does not require cocoercivity of $F$. However, the computation of $J_{\tau(F+G)}$ is in general nontrivial even when $J_{\tau G}$ can be computed efficiently. Hence, the advantage of not requiring cocoercivity comes at the cost of a more expensive iteration.

Extragradient (EG) (Korpelevich, 1977) is a classical algorithm that gets the best of both worlds and, for problem (VI), it iterates as

$$\mathbf{u}_{k+1/2} = P_C(\mathbf{u}_k - \tau F(\mathbf{u}_k))$$
$$\mathbf{u}_{k+1} = P_C(\mathbf{u}_k - \tau F(\mathbf{u}_{k+1/2})).$$

This method converges with $L$-Lipschitz $F$ and only uses $J_{\tau G}$ for $G = \partial\delta_C$. It turns out that PP and EG have the optimal convergence rates for the gap whereas they are suboptimal for guarantees on the residual, see Golowich et al. (2020).

**Variance reduction.** The main idea of variance reduction is to use an estimator $\widetilde{F}$ such that $\mathbb{E}\|\widetilde{F}(\mathbf{u}) - F(\mathbf{u})\|^2$ gets progressively smaller as we run the algorithm. There are several different estimators that are used in minimization such as SVRG (Johnson & Zhang, 2013; Kovalev et al., 2020), SARAH/SPIDER/PAGE (Nguyen et al., 2017; Fang et al., 2018; Li et al., 2021) or SAGA (Defazio et al., 2014; Schmidt et al., 2017). For reference, given $F = \frac{1}{n}\sum_{i=1}^{n} F_i$, SVRG estimator is written as

$$\widetilde{F}(\mathbf{u}_k) = F(\mathbf{w}_k) - F_i(\mathbf{w}_k) + F_i(\mathbf{u}_k),$$

for a randomly selected index $i \in \{1, \ldots, n\}$ and a suitably selected point $\mathbf{w}_k$. A common choice is to select $\mathbf{w}_k$ to be updated only once every couple of epochs. By using this estimator, recent work Alacaoglu & Malitsky (2022) showed how to obtain variance reduced extragradient algorithms with optimal complexity for the gap in terms of both the number of operators $n$ and the desired accuracy $\varepsilon$.

Another estimator that is popular for minimization problems is the PAGE estimator (Li et al., 2021), written for operators as:

$$\widetilde{F}(\mathbf{u}_k) = \begin{cases} F(\mathbf{u}_k), & \text{w.p. } p_k \\ \widetilde{F}(\mathbf{u}_{k-1}) + \frac{1}{b}\sum_{i\in\mathcal{S}_k}\left(F_i(\mathbf{u}_k) - F_i(\mathbf{u}_{k-1})\right), & \text{w.p. } 1 - p_k, \end{cases} \tag{A.1}$$

where $p_k$ and the mini-batch $\mathcal{S}_k$ with $b = |\mathcal{S}_k|$ are the parameters. Even though this estimator has been shown to have unique benefits for minimization, it has not find much use for operators for finite-sum case. It was recently used by Cai et al. (2022a) for operators given as an expectation. Here we introduce a useful result on the variance bound of the PAGE estimator, for our later analysis. Note that this lemma is a slight modification of (Li et al., 2021, Lemma 3) by using without replacement sampling.

**Lemma A.1.** *Let the minibatch $\mathcal{S}_k$ be uniformly sampled from $[n]$ without replacement. Then the variance of $\widetilde{F}$ defined by Eq. (A.1) satisfies the following recursive bound. For all $k \geq 1$, it holds that*

$$\mathbb{E}[\|\widetilde{F}(\mathbf{u}_k) - F(\mathbf{u}_k)\|^2] \leq (1 - p_k)\mathbb{E}[\|\widetilde{F}(\mathbf{u}_{k-1}) - F(\mathbf{u}_{k-1})\|^2]$$
$$+ \frac{(n-b)(1-p_k)}{b(n-1)}\mathbb{E}\left[\sum_{i=1}^{n}\frac{1}{n}\|F_i(\mathbf{u}_k) - F_i(\mathbf{u}_{k-1})\|^2\right]. \tag{A.2}$$

*Proof.* Let $\mathcal{F}_k$ denote the filtration that contains all algorithmic randomness up to and including iteration $\mathbf{u}_k$. Using the definition of $\widetilde{F}$ in (A.1) where $\widetilde{F}(\mathbf{u}_k) = F(\mathbf{u}_k)$ with probability $p_k$, then conditional on $\mathcal{F}_k$, we have for all $k \geq 1$ that

$$\mathbb{E}\left[\|\widetilde{F}(\mathbf{u}_k) - F(\mathbf{u}_k)\|^2 \,\Big|\, \mathcal{F}_k\right]$$
$$= (1 - p_k)\mathbb{E}\left[\left\|\widetilde{F}(\mathbf{u}_{k-1}) + \frac{1}{b}\sum_{i\in\mathcal{S}_k}\left(F_i(\mathbf{u}_k) - F_i(\mathbf{u}_{k-1})\right) - F(\mathbf{u}_k)\right\|^2 \,\Big|\, \mathcal{F}_k\right]$$
$$= (1 - p_k)\mathbb{E}_{\mathcal{S}_k}\left[\left\|\widetilde{F}(\mathbf{u}_{k-1}) + \frac{1}{b}\sum_{i\in\mathcal{S}_k}\left(F_i(\mathbf{u}_k) - F_i(\mathbf{u}_{k-1})\right) - F(\mathbf{u}_k)\right\|^2\right], \tag{A.3}$$

where $\mathbb{E}_{\mathcal{S}_k}$ denotes the expectation with respect to the randomness of $\mathcal{S}_k$. Adding and substracting $F(\mathbf{u}_{k-1})$ in the quadratic, and noticing $\mathbb{E}_{\mathcal{S}_k}\left[\frac{1}{b}\sum_{i\in\mathcal{S}_k}\left(F_i(\mathbf{u}_k) - F_i(\mathbf{u}_{k-1})\right)\right] = F(\mathbf{u}_k) - F(\mathbf{u}_{k-1})$,

we have

$$\mathbb{E}_{\mathcal{S}_k}\left[\left\|\widetilde{F}(\mathbf{u}_{k-1}) + \frac{1}{b}\sum_{i\in\mathcal{S}_k}\left(F_i(\mathbf{u}_k) - F_i(\mathbf{u}_{k-1})\right) - F(\mathbf{u}_k)\right\|^2\right]$$

$$= \mathbb{E}_{\mathcal{S}_k}\left[\left\|\frac{1}{b}\sum_{i\in\mathcal{S}_k}\left(F_i(\mathbf{u}_k) - F_i(\mathbf{u}_{k-1})\right) - \left(F(\mathbf{u}_k) - F(\mathbf{u}_{k-1})\right)\right\|^2\right] + \|F(\mathbf{u}_{k-1}) - \widetilde{F}(\mathbf{u}_{k-1})\|^2$$

$$+ 2\left\langle \widetilde{F}(\mathbf{u}_{k-1}) - F(\mathbf{u}_{k-1}), \mathbb{E}_{\mathcal{S}_k}\left[\frac{1}{b}\sum_{i\in\mathcal{S}_k}\left(F_i(\mathbf{u}_k) - F_i(\mathbf{u}_{k-1})\right) - \left(F(\mathbf{u}_k) - F(\mathbf{u}_{k-1})\right)\right]\right\rangle$$

$$= \mathbb{E}_{\mathcal{S}_k}\left[\left\|\frac{1}{b}\sum_{i\in\mathcal{S}_k}\left(F_i(\mathbf{u}_k) - F_i(\mathbf{u}_{k-1})\right) - \left(F(\mathbf{u}_k) - F(\mathbf{u}_{k-1})\right)\right\|^2\right]$$

$$+ \|F(\mathbf{u}_{k-1}) - \widetilde{F}(\mathbf{u}_{k-1})\|^2. \tag{A.4}$$

With $\mathcal{S}_k$ sampled without replacement according to the uniform distribution, we have (see for example (Lohr, 2021, Section 2.7))

$$\mathbb{E}_{\mathcal{S}_k}\left[\left\|\frac{1}{b}\sum_{i\in\mathcal{S}_k}\left(F_i(\mathbf{u}_k) - F_i(\mathbf{u}_{k-1})\right) - \left(F(\mathbf{u}_k) - F(\mathbf{u}_{k-1})\right)\right\|^2\right]$$

$$\leq \frac{(n-b)}{b(n-1)}\sum_{i=1}^{n}\frac{1}{n}\|F_i(\mathbf{u}_k) - F_i(\mathbf{u}_{k-1}) - \left(F(\mathbf{u}_k) - F(\mathbf{u}_{k-1})\right)\|^2$$

$$\overset{(i)}{\leq} \frac{(n-b)}{b(n-1)}\sum_{i=1}^{n}\frac{1}{n}\|F_i(\mathbf{u}_k) - F_i(\mathbf{u}_{k-1})\|^2,$$

where $(i)$ holds because $\mathbb{E}[\|X - \mathbb{E}[X]\|^2] \leq \mathbb{E}[\|X\|^2]$ for any random variable $X$. Using this estimate on (A.4) and plugging in the result to (A.3) give

$$\mathbb{E}\left[\|\widetilde{F}(\mathbf{u}_k) - F(\mathbf{u}_k)\|^2 \,\Big|\, \mathcal{F}_k\right]$$

$$\leq (1 - p_k)\|F(\mathbf{u}_{k-1}) - \widetilde{F}(\mathbf{u}_{k-1})\|^2 + (1 - p_k)\frac{(n-b)}{b(n-1)}\sum_{i=1}^{n}\frac{1}{n}\|F_i(\mathbf{u}_k) - F_i(\mathbf{u}_{k-1})\|^2.$$

Taking expectation with respect to all the randomness and using the tower rule, we conclude. ∎

## B OMITTED PROOFS FROM SECTION 3 (COCOERCIVE CASE)

We first prove the following lemma on the useful conclusions of our parameter choices, which are essential in the proof of Lemma 3.1.

**Lemma B.1.** *For $k \geq 1$, let*

$$\eta = \frac{1}{4L}, \quad \lambda_k = \frac{2}{k+4}, \quad c_k = \frac{(\sqrt{n}+2)(k+4)}{4L}, \quad b = \left\lceil\sqrt{n}\right\rceil, \quad p_{k+1} = \begin{cases} \frac{4}{k+5} & \forall k \leq \sqrt{n} \\ \frac{4}{\sqrt{n}+5} & \forall k \geq \sqrt{n} \end{cases}$$

*as in Alg. 1 and Lemma 3.1. Then, for $k \geq 1$ it holds that*

$$\frac{2\eta}{\lambda_k} - c_k + \frac{c_{k+1}(1 - p_{k+1})}{1 - \lambda_k} \leq 0, \tag{B.1a}$$

$$\frac{2\eta}{\lambda_k} - \frac{1}{\lambda_k L} + \frac{c_{k+1}(1 - p_{k+1})(n-b)}{b(n-1)(1 - \lambda_k)} \leq 0. \tag{B.1b}$$

*Proof.* We start by manipulating the last term on the left-hand side of (B.1b). By the definition of $b$, we have $b \geq \sqrt{n}$ which implies that

$$\frac{c_{k+1}(1 - p_{k+1})(n-b)}{b(n-1)(1 - \lambda_k)} \leq \frac{c_{k+1}(1 - p_{k+1})(n - \sqrt{n})}{\sqrt{n}(n-1)(1 - \lambda_k)} = \frac{c_{k+1}(1 - p_{k+1})}{(\sqrt{n}+1)(1 - \lambda_k)}.$$

On the one hand, in view of this inequality and $\eta = \frac{1}{4L}$, the following suffices to guarantee Eq. (B.1b):

$$\frac{c_{k+1}(1 - p_{k+1})}{1 - \lambda_k} \leq \frac{\sqrt{n} + 1}{2\lambda_k L}. \tag{B.2}$$

On the other hand, with the definition of $\eta$, Eq. (B.1a) is equivalent to

$$\frac{c_{k+1}(1 - p_{k+1})}{1 - \lambda_k} \leq c_k - \frac{1}{2\lambda_k L}. \tag{B.3}$$

To get the best upper bounds in (B.2) and (B.3), we set $c_k - \frac{1}{2\lambda_k L} = \frac{\sqrt{n}+1}{2\lambda_k L}$ and obtain $c_k = \frac{\sqrt{n}+2}{2\lambda_k L}$, thus $c_{k+1} = \frac{\sqrt{n}+2}{2\lambda_{k+1} L}$. Then the inequalities in (B.2) and (B.3) are equivalent to

$$
\begin{aligned}
1 - p_{k+1} &\leq \frac{(\sqrt{n} + 1)(1 - \lambda_k)}{2\lambda_k L c_{k+1}} \\
&= \frac{\sqrt{n} + 1}{\sqrt{n} + 2} \frac{\lambda_{k+1}(1 - \lambda_k)}{\lambda_k} \\
&= \frac{\sqrt{n} + 1}{\sqrt{n} + 2} \frac{k + 2}{k + 5},
\end{aligned} \tag{B.4}
$$

where we plug in the definition of $c_{k+1}$ for the first equality and the definition of $\lambda_k$ for the last equality. When $k \leq \sqrt{n}$, we have $\frac{\sqrt{n}+1}{\sqrt{n}+2} \geq \frac{k+1}{k+2}$, then it suffices to choose $p_{k+1} = \frac{4}{k+5}$ to ensure that Eq. (B.4) holds. When $k \geq \sqrt{n}$, we have $\frac{k+2}{k+5} \geq \frac{\sqrt{n}+2}{\sqrt{n}+5}$, and it suffices to take $p_{k+1} = \frac{4}{\sqrt{n}+5}$. This is the definition of $p_{k+1}$ and the proof is completed. ∎

**Lemma 3.1.** *Let Assumptions 1 and 3 hold. Then, for the iterates $\mathbf{u}_k$ of Algorithm 1 and the potential function $\mathcal{C}_k$ defined by*

$$\mathcal{C}_k = \frac{\eta}{2\lambda_k} \|F(\mathbf{u}_k) + \mathbf{g}_k\|^2 + \langle F(\mathbf{u}_k) + \mathbf{g}_k, \mathbf{u}_k - \mathbf{u}_0 \rangle + c_k \|F(\mathbf{u}_k) - \widetilde{F}(\mathbf{u}_k)\|^2, \tag{3.2}$$

*we have that $\mathbb{E}[\mathcal{C}_{k+1}] \leq (1 - \lambda_k)\mathbb{E}[\mathcal{C}_k]$ for $k \geq 1$, where $\mathbf{g}_{k+1} = \frac{1}{\eta}\big(\lambda_k\mathbf{u}_0 + (1 - \lambda_k)\mathbf{u}_k - \eta\widetilde{F}(\mathbf{u}_k) - \mathbf{u}_{k+1}\big) \in G(\mathbf{u}_{k+1})$ and $c_k = \frac{(\sqrt{n}+2)(k+4)}{4L}$.*

*Proof.* By Assumption 3 on $F$ and the monotonicity of $G$, we have for $k \geq 1$

$$\frac{1}{nL}\sum_{i=1}^{n}\|F_i(\mathbf{u}_{k+1}) - F_i(\mathbf{u}_k)\|^2 \leq \langle F(\mathbf{u}_{k+1}) + \mathbf{g}_{k+1}, \mathbf{u}_{k+1} - \mathbf{u}_k \rangle - \langle F(\mathbf{u}_k) + \mathbf{g}_k, \mathbf{u}_{k+1} - \mathbf{u}_k \rangle,$$

where $\mathbf{g}_{k+1} \in G(\mathbf{u}_{k+1})$ and $\mathbf{g}_k \in G(\mathbf{u}_k)$. Dividing both sides by $\lambda_k$, we get

$$
\begin{aligned}
&\frac{1}{\lambda_k nL}\sum_{i=1}^{n}\|F_i(\mathbf{u}_{k+1}) - F_i(\mathbf{u}_k)\|^2 \\
&\leq \frac{1}{\lambda_k}\langle F(\mathbf{u}_{k+1}) + \mathbf{g}_{k+1}, \mathbf{u}_{k+1} - \mathbf{u}_k \rangle - \frac{1}{\lambda_k}\langle F(\mathbf{u}_k) + \mathbf{g}_k, \mathbf{u}_{k+1} - \mathbf{u}_k \rangle. \tag{B.5}
\end{aligned}
$$

Recall that by the definition of the resolvent operator and the definition of $\mathbf{u}_{k+1}$, we have for $k \geq 1$

$$\mathbf{g}_{k+1} = \frac{1}{\eta}\big(\lambda_k\mathbf{u}_0 + (1 - \lambda_k)\mathbf{u}_k - \eta\widetilde{F}(\mathbf{u}_k) - \mathbf{u}_{k+1}\big) \in G(\mathbf{u}_{k+1}),$$

which lets us rewrite the algorithm updates as

$$\mathbf{u}_{k+1} = \lambda_k\mathbf{u}_0 + (1 - \lambda_k)\mathbf{u}_k - \eta(\widetilde{F}(\mathbf{u}_k) + \mathbf{g}_{k+1}).$$

By simple rearrangements on this representation of $\mathbf{u}_{k+1}$, we have for $k \geq 1$ that

$$\mathbf{u}_{k+1} - \mathbf{u}_k = \lambda_k(\mathbf{u}_0 - \mathbf{u}_k) - \eta(\widetilde{F}(\mathbf{u}_k) + \mathbf{g}_{k+1}) \tag{B.6a}$$

$$\mathbf{u}_{k+1} - \mathbf{u}_k = \frac{\lambda_k}{1 - \lambda_k}(\mathbf{u}_0 - \mathbf{u}_{k+1}) - \frac{\eta}{1 - \lambda_k}(\widetilde{F}(\mathbf{u}_k) + \mathbf{g}_{k+1}). \tag{B.6b}$$

Plugging Eq. (B.6b) in the first term in the right-hand side of Eq. (B.5) and Eq. (B.6a) in the second term in the right-hand side of Eq. (B.5), we obtain

$$
\frac{1}{\lambda_k n L} \sum_{i=1}^{n} \|F_i(\mathbf{u}_{k+1}) - F_i(\mathbf{u}_k)\|^2
$$

$$
\leq \frac{1}{1-\lambda_k} \left\langle F(\mathbf{u}_{k+1}) + \mathbf{g}_{k+1}, \mathbf{u}_0 - \mathbf{u}_{k+1} \right\rangle - \frac{\eta}{\lambda_k(1-\lambda_k)} \left\langle F(\mathbf{u}_{k+1}) + \mathbf{g}_{k+1}, \widetilde{F}(\mathbf{u}_k) + \mathbf{g}_{k+1} \right\rangle
$$

$$
- \left\langle F(\mathbf{u}_k) + \mathbf{g}_k, \mathbf{u}_0 - \mathbf{u}_k \right\rangle + \frac{\eta}{\lambda_k} \left\langle F(\mathbf{u}_k) + \mathbf{g}_k, \widetilde{F}(\mathbf{u}_k) + \mathbf{g}_{k+1} \right\rangle. \tag{B.7}
$$

We next represent the second and fourth inner products in the right-hand side of Eq. (B.7) with squared norms as

$$
\frac{\eta}{\lambda_k} \left\langle F(\mathbf{u}_k) + \mathbf{g}_k, \widetilde{F}(\mathbf{u}_k) + \mathbf{g}_{k+1} \right\rangle
$$

$$
= \frac{\eta}{2\lambda_k} \left( \|F(\mathbf{u}_k) + \mathbf{g}_k\|^2 + \|\widetilde{F}(\mathbf{u}_k) + \mathbf{g}_{k+1}\|^2 - \|F(\mathbf{u}_k) + \mathbf{g}_k - \widetilde{F}(\mathbf{u}_k) - \mathbf{g}_{k+1}\|^2 \right)
$$

$$
\leq \frac{\eta}{2\lambda_k} \left( \|F(\mathbf{u}_k) + \mathbf{g}_k\|^2 + \|\widetilde{F}(\mathbf{u}_k) + \mathbf{g}_{k+1}\|^2 \right) \tag{B.8}
$$

and

$$
- \frac{\eta}{\lambda_k(1-\lambda_k)} \left\langle F(\mathbf{u}_{k+1}) + \mathbf{g}_{k+1}, \widetilde{F}(\mathbf{u}_k) + \mathbf{g}_{k+1} \right\rangle
$$

$$
= - \frac{\eta}{2\lambda_k(1-\lambda_k)} \left( \|F(\mathbf{u}_{k+1}) + \mathbf{g}_{k+1}\|^2 + \|\widetilde{F}(\mathbf{u}_k) + \mathbf{g}_{k+1}\|^2 - \|F(\mathbf{u}_{k+1}) - \widetilde{F}(\mathbf{u}_k)\|^2 \right). \tag{B.9}
$$

We now estimate the second term on the right-hand side of Eq. (B.9) by Young's inequality as

$$
- \frac{\eta}{2\lambda_k(1-\lambda_k)} \|\widetilde{F}(\mathbf{u}_k) + \mathbf{g}_{k+1}\|^2 = - \left( \frac{\eta}{2\lambda_k} + \frac{\eta}{2(1-\lambda_k)} \right) \|\widetilde{F}(\mathbf{u}_k) + \mathbf{g}_{k+1}\|^2
$$

$$
\leq - \frac{\eta}{2\lambda_k} \|\widetilde{F}(\mathbf{u}_k) + \mathbf{g}_{k+1}\|^2 - \frac{\eta}{4(1-\lambda_k)} \|F(\mathbf{u}_{k+1}) + \mathbf{g}_{k+1}\|^2
$$

$$
+ \frac{\eta}{2(1-\lambda_k)} \|F(\mathbf{u}_{k+1}) - \widetilde{F}(\mathbf{u}_k)\|^2.
$$

We use this estimation in Eq. (B.9) and combine like terms by also using the definition of $\lambda_k$, which gives $-\frac{\eta}{2\lambda_k(1-\lambda_k)} - \frac{\eta}{4(1-\lambda_k)} = -\frac{\eta}{2\lambda_{k+1}(1-\lambda_k)}$, to get

$$
- \frac{\eta}{\lambda_k(1-\lambda_k)} \left\langle F(\mathbf{u}_{k+1}) + \mathbf{g}_{k+1}, \widetilde{F}(\mathbf{u}_k) + \mathbf{g}_{k+1} \right\rangle
$$

$$
\leq - \frac{\eta}{2\lambda_{k+1}(1-\lambda_k)} \|F(\mathbf{u}_{k+1}) + \mathbf{g}_{k+1}\|^2 - \frac{\eta}{2\lambda_k} \|\widetilde{F}(\mathbf{u}_k) + \mathbf{g}_{k+1}\|^2
$$

$$
+ \frac{\eta(1+\lambda_k)}{2\lambda_k(1-\lambda_k)} \|F(\mathbf{u}_{k+1}) - \widetilde{F}(\mathbf{u}_k)\|^2
$$

$$
\leq - \frac{\eta}{2\lambda_{k+1}(1-\lambda_k)} \|F(\mathbf{u}_{k+1}) + \mathbf{g}_{k+1}\|^2 - \frac{\eta}{2\lambda_k} \|\widetilde{F}(\mathbf{u}_k) + \mathbf{g}_{k+1}\|^2
$$

$$
+ \frac{2\eta}{n\lambda_k} \sum_{i=1}^{n} \|F_i(\mathbf{u}_{k+1}) - F_i(\mathbf{u}_k)\|^2 + \frac{2\eta}{\lambda_k} \|F(\mathbf{u}_k) - \widetilde{F}(\mathbf{u}_k)\|^2, \tag{B.10}
$$

where the last step is by Young's inequality, Jensen's inequality and $\frac{1+\lambda_k}{1-\lambda_k} \leq 2$.

Combining Eq. (B.8) and Eq. (B.10), plugging into Eq. (B.7), joining like terms and rearranging give

$$
\frac{1}{1-\lambda_k} \left( \frac{\eta}{2\lambda_{k+1}} \|F(\mathbf{u}_{k+1}) + \mathbf{g}_{k+1}\|^2 + \left\langle F(\mathbf{u}_{k+1}) + \mathbf{g}_{k+1}, \mathbf{u}_{k+1} - \mathbf{u}_0 \right\rangle \right)
$$

$$
\leq \frac{\eta}{2\lambda_k} \|F(\mathbf{u}_k) + \mathbf{g}_k\|^2 + \left\langle F(\mathbf{u}_k) + \mathbf{g}_k, \mathbf{u}_k - \mathbf{u}_0 \right\rangle
$$

$$
+ \frac{1}{n\lambda_k} \left( 2\eta - \frac{1}{L} \right) \sum_{i=1}^{n} \|F_i(\mathbf{u}_{k+1}) - F_i(\mathbf{u}_k)\|^2 + \frac{2\eta}{\lambda_k} \|F(\mathbf{u}_k) - \widetilde{F}(\mathbf{u}_k)\|^2.
$$

Adding $\frac{c_{k+1}}{1-\lambda_k}\|F(\mathbf{u}_{k+1}) - \widetilde{F}(\mathbf{u}_{k+1})\|^2 - c_k\|F(\mathbf{u}_k) - \widetilde{F}(\mathbf{u}_k)\|^2$ to both sides, rearranging, and using the definition of $\mathcal{C}_k$, we obtain

$$\frac{1}{1-\lambda_k}\mathcal{C}_{k+1} \leq \mathcal{C}_k + \frac{c_{k+1}}{1-\lambda_k}\|F(\mathbf{u}_{k+1}) - \widetilde{F}(\mathbf{u}_{k+1})\|^2 + \left(\frac{2\eta}{\lambda_k} - c_k\right)\|F(\mathbf{u}_k) - \widetilde{F}(\mathbf{u}_k)\|^2$$
$$+ \frac{1}{n\lambda_k}\left(2\eta - \frac{1}{L}\right)\sum_{i=1}^{n}\|F_i(\mathbf{u}_{k+1}) - F_i(\mathbf{u}_k)\|^2. \tag{B.11}$$

We recall the result of Lemma A.1 which states, for $k \geq 1$, that

$$\mathbb{E}[\|F(\mathbf{u}_{k+1}) - \widetilde{F}(\mathbf{u}_{k+1})\|^2] \leq (1 - p_{k+1})\mathbb{E}[\|F(\mathbf{u}_k) - \widetilde{F}(\mathbf{u}_k)\|^2]$$
$$+ \frac{(1 - p_{k+1})(n - b)}{b(n - 1)}\mathbb{E}\left[\sum_{i=1}^{n}\frac{1}{n}\|F_i(\mathbf{u}_{k+1}) - F_i(\mathbf{u}_k)\|^2\right]. \tag{B.12}$$

We take the expectation of both sides of (B.11), and then upper bound the resulting second term on the right-hand side of (B.11) by (B.12). As a result, we have for $k \geq 1$ that

$$\frac{1}{1-\lambda_k}\mathbb{E}[\mathcal{C}_{k+1}]$$
$$\leq \mathbb{E}[\mathcal{C}_k] + \left(\frac{2\eta}{\lambda_k} - c_k + \frac{c_{k+1}(1 - p_{k+1})}{1 - \lambda_k}\right)\mathbb{E}\|F(\mathbf{u}_k) - \widetilde{F}(\mathbf{u}_k)\|^2$$
$$+ \left(\frac{2\eta}{\lambda_k} - \frac{1}{\lambda_k L} + \frac{c_{k+1}(1 - p_{k+1})(n - b)}{b(n - 1)(1 - \lambda_k)}\right)\mathbb{E}\left[\sum_{i=1}^{n}\frac{1}{n}\|F_i(\mathbf{u}_{k+1}) - F_i(\mathbf{u}_k)\|^2\right]. \tag{B.13}$$

By Lemma B.1, we have that the second and third terms on the right-hand side of (B.13) are non-positive. Hence, we get the result after multiplying both sides by $1 - \lambda_k$. ∎

**Theorem 3.2.** *Let Assumptions 1 and 3 hold. Then, for the iterates $\mathbf{u}_k$ of Algorithm 1, we have*

$$\mathbb{E}[\text{Res}_{F+G}(\mathbf{u}_k)] \leq \left(\mathbb{E}[\text{Res}^2_{F+G}(\mathbf{u}_k)]\right)^{1/2} \leq \frac{16L\|\mathbf{u}_0 - \mathbf{u}_*\|}{k + 4}.$$

*In particular, given accuracy $\varepsilon > 0$, to return a point $\mathbf{u}_K$ such that $\mathbb{E}[\text{Res}_{F+G}(\mathbf{u}_K)] \leq \varepsilon$, the stochastic oracle complexity of Algorithm 1 is $\widetilde{\mathcal{O}}\left(n + \frac{\sqrt{n}L\|\mathbf{u}_0 - \mathbf{u}_*\|}{\varepsilon}\right)$.*

*Proof.* After iterating the result of Lemma 3.1, we have

$$\mathbb{E}[\mathcal{C}_k] \leq \left(\prod_{i=1}^{k-1}(1 - \lambda_i)\right)\mathbb{E}[\mathcal{C}_1].$$

Since $\lambda_i = \frac{2}{i+4}$, we have

$$\prod_{i=1}^{k-1}(1 - \lambda_i) = \prod_{i=1}^{k-1}\frac{i + 2}{i + 4} = \frac{(k + 1)!/2!}{(k + 3)!/4!} = \frac{12}{(k + 2)(k + 3)},$$

which leads to

$$\mathbb{E}[\mathcal{C}_k] \leq \frac{12}{(k + 2)(k + 3)}\mathbb{E}[\mathcal{C}_1]. \tag{B.14}$$

Next, we bound $\mathbb{E}[\mathcal{C}_1]$, recalling the definition in (3.2). First note that $\widetilde{F}(\mathbf{u}_1) = F(\mathbf{u}_1)$, we know that $\mathcal{C}_1$ does not involve any randomness, i.e., $\mathbb{E}[\mathcal{C}_1] = \mathcal{C}_1$, and have

$$\mathcal{C}_1 = \frac{\eta}{2\lambda_1}\|F(\mathbf{u}_1) + \mathbf{g}_1\|^2 + \langle F(\mathbf{u}_1) + \mathbf{g}_1, \mathbf{u}_1 - \mathbf{u}_0\rangle.$$

Further, by the definition of $\mathbf{u}_1$, we have

$$\mathbf{u}_1 = J_{\frac{\eta}{2\lambda_1}G}\left(\mathbf{u}_0 - \frac{\eta}{2\lambda_1}F(\mathbf{u}_0)\right) = \mathbf{u}_0 - \frac{\eta}{2\lambda_1}F(\mathbf{u}_0) - \frac{\eta}{2\lambda_1}\mathbf{g}_1.$$

With this, we obtain

$$\mathcal{C}_1 = \frac{\eta}{2\lambda_1}\|F(\mathbf{u}_1) + \mathbf{g}_1\|^2 + \langle F(\mathbf{u}_1) + \mathbf{g}_1, \mathbf{u}_1 - \mathbf{u}_0\rangle$$
$$= \frac{\eta}{2\lambda_1}\|F(\mathbf{u}_1) + \mathbf{g}_1\|^2 - \frac{\eta}{2\lambda_1}\langle F(\mathbf{u}_1) + \mathbf{g}_1, F(\mathbf{u}_0) + \mathbf{g}_1\rangle.$$

Decomposing the inner product term above, by adding and subtracting $F(\mathbf{u}_1)$ in the second argument, we obtain

$$\mathcal{C}_1 = \frac{\eta}{2\lambda_1}\|F(\mathbf{u}_1) + \mathbf{g}_1\|^2 - \frac{\eta}{2\lambda_1}\|F(\mathbf{u}_1) + \mathbf{g}_1\|^2 - \frac{\eta}{2\lambda_1}\langle F(\mathbf{u}_1) + \mathbf{g}_1, F(\mathbf{u}_0) - F(\mathbf{u}_1)\rangle$$
$$= \frac{\eta}{2\lambda_1}\langle F(\mathbf{u}_1) + \mathbf{g}_1, F(\mathbf{u}_1) - F(\mathbf{u}_0)\rangle.$$

Plugging in the definition $\mathbf{g}_1 = \frac{2\lambda_1}{\eta}(\mathbf{u}_0 - \mathbf{u}_1) - F(\mathbf{u}_0)$, we obtain

$$\mathcal{C}_1 = \frac{\eta}{2\lambda_1}\left\langle F(\mathbf{u}_1) - F(\mathbf{u}_0) + \frac{2\lambda_1}{\eta}(\mathbf{u}_0 - \mathbf{u}_1), F(\mathbf{u}_1) - F(\mathbf{u}_0)\right\rangle$$
$$= \frac{\eta}{2\lambda_1}\|F(\mathbf{u}_1) - F(\mathbf{u}_0)\|^2 - \langle F(\mathbf{u}_1) - F(\mathbf{u}_0), \mathbf{u}_1 - \mathbf{u}_0\rangle$$
$$\overset{(i)}{\leq} 0,$$

where $(i)$ is by $\frac{1}{L}$-cocoercivity of $F$ and $\frac{\eta}{2\lambda_1} = \frac{5}{16L} < \frac{1}{L}$. So we obtain $\mathbb{E}[\mathcal{C}_k] \leq 0$ in view of (B.14). Recalling the definition of $\mathcal{C}_k$ and noticing the term $c_k\|F(\mathbf{u}_k) - \widetilde{F}(\mathbf{u}_k)\|^2$ is nonnegative, we have

$$\mathbb{E}\left[\frac{\eta(k+4)}{4}\|F(\mathbf{u}_k) + \mathbf{g}_k\|^2 + \langle F(\mathbf{u}_k) + \mathbf{g}_k, \mathbf{u}_k - \mathbf{u}_0\rangle\right] \leq 0.$$

Since $\mathbf{u}_*$ is a solution to (MI), there exists $\mathbf{g}_*$ such that $\mathbf{g}_* \in G(\mathbf{u}_*)$ and $F(\mathbf{u}_*) + \mathbf{g}_* = \mathbf{0}$, then we have

$$\langle F(\mathbf{u}_k) + \mathbf{g}_k, \mathbf{u}_k - \mathbf{u}_0\rangle = \langle F(\mathbf{u}_k) + \mathbf{g}_k, \mathbf{u}_k - \mathbf{u}_*\rangle + \langle F(\mathbf{u}_k) + \mathbf{g}_k, \mathbf{u}_* - \mathbf{u}_0\rangle$$
$$\overset{(i)}{\geq} \langle F(\mathbf{u}_k) + \mathbf{g}_k, \mathbf{u}_* - \mathbf{u}_0\rangle$$
$$\overset{(ii)}{\geq} -\|F(\mathbf{u}_k) + \mathbf{g}_k\|\|\mathbf{u}_0 - \mathbf{u}_*\|,$$

where we use the monotonicity of $F + G$ for $(i)$ and Cauchy-Schwarz inequality for $(ii)$. Noticing that $\mathbb{E}[\|F(\mathbf{u}_k) + \mathbf{g}_k\|] \leq (\mathbb{E}[\|F(\mathbf{u}_k) + \mathbf{g}_k\|^2])^{1/2}$ by Jensen's inequality, then we have

$$\frac{\eta(k+4)}{4}\mathbb{E}[\|F(\mathbf{u}_k) + \mathbf{g}_k\|^2] \leq \|\mathbf{u}_0 - \mathbf{u}_*\|\left(\mathbb{E}[\|F(\mathbf{u}_k) + \mathbf{g}_k\|^2]\right)^{1/2}$$

Completing the square and then solving for the quadratic gives

$$\left(\mathbb{E}[\mathrm{Res}_{F+G}^2(\mathbf{u}_k)]\right)^{1/2} = \left(\mathbb{E}[\|F(\mathbf{u}_k) + \mathbf{g}_k\|^2]\right)^{1/2} \leq \frac{16L\|\mathbf{u}_0 - \mathbf{u}_*\|}{k+4}.$$

Given $\varepsilon > 0$, to return a point $\mathbf{u}_K$ such that $\left(\mathbb{E}[\|F(\mathbf{u}_K) + \mathbf{g}_K\|^2]\right)^{1/2} \leq \varepsilon$, which also guarantees that $\mathbb{E}[\|F(\mathbf{u}_K) + \mathbf{g}_K\|] \leq \varepsilon$ by Jensen's inequality, the total number of iterations required is $K = \lceil\frac{16L\|\mathbf{u}_0 - \mathbf{u}_*\|}{\varepsilon}\rceil$. To obtain the total number of stochastic oracle queries, we let $m_k$ be the number of individual operator evaluations at iteration $k$ (in which we compute $\mathbf{u}_{k+1}$ and $\widetilde{F}(\mathbf{u}_{k+1})$ as Alg. 1) for $k \geq 1$, and $M = 2n + \sum_{k=1}^{K-1} m_k$ be the total number of individual operator evaluations to return $\mathbf{u}_K$. Conditioned on the filtration $\mathcal{F}_k$ that contains all algorithm randomness up to and not including iteration $k$, we have for $k \geq 1$

$$\mathbb{E}[m_k \mid \mathcal{F}_k] = p_{k+1}n + (1 - p_{k+1})2b = \begin{cases} \frac{4}{k+5}n + 2\frac{k+1}{k+5}\lceil\sqrt{n}\rceil & \text{if } k \leq \sqrt{n}, \\ \frac{4}{\sqrt{n}+5}n + 2\frac{\sqrt{n}+1}{\sqrt{n}+5}\lceil\sqrt{n}\rceil & \text{if } k \geq \sqrt{n}. \end{cases}$$

Taking expectation w.r.t. all randomness and summing from $k = 1$ to $K$, we obtain

$$\mathbb{E}[M] = 2n + \mathbb{E}\Big[\sum_{k=1}^{K-1} m_k\Big]$$

$$= 2n + \sum_{k=1}^{\lfloor\sqrt{n}\rfloor} \mathbb{E}[m_k] + \sum_{k=\lceil\sqrt{n}\rceil}^{K-1} \mathbb{E}[m_k]$$

$$\leq 2n + 4n \sum_{k=1}^{\lfloor\sqrt{n}\rfloor} \frac{1}{k+5} + 2\lceil\sqrt{n}\rceil^2 + 6\sqrt{n}(K - \sqrt{n})$$

$$\leq 4n + 4\sqrt{n} + 2 + 4n\log(\sqrt{n} + 5) + \frac{96\sqrt{n}L\|\mathbf{u}_0 - \mathbf{u}_*\|}{\varepsilon} = \widetilde{\mathcal{O}}\Big(n + \frac{\sqrt{n}L\|\mathbf{u}_0 - \mathbf{u}_*\|}{\varepsilon}\Big),$$

which completes the proof. $\blacksquare$

## C OMITTED PROOFS FROM SECTION 4 (MONOTONE & LIPSCHITZ CASE)

**Proposition 4.1.** *For any fixed $\eta > 0$, let $P^\eta(\mathbf{u}) = \mathbf{u} - J_{\eta(F+G)}(\mathbf{u})$. If $\|P^\eta(\mathbf{u})\| \leq \eta\varepsilon$ for some $\varepsilon > 0$, then we have $\text{Res}_{F+G}(\bar{\mathbf{u}}) \leq \varepsilon$ with $\bar{\mathbf{u}} = \mathbf{u} - P^\eta(\mathbf{u}) = J_{\eta(F+G)}(\mathbf{u})$.*

*Proof.* By the definition of resolvent operator, we have

$$\mathbf{u} - \bar{\mathbf{u}} \in \eta F(\bar{\mathbf{u}}) + \eta G(\bar{\mathbf{u}}) \iff \mathbf{u} \in \mathbf{u} - P^\eta(\mathbf{u}) + \eta F(\bar{\mathbf{u}}) + \eta G(\bar{\mathbf{u}}) \iff \mathbf{0} \in \eta F(\bar{\mathbf{u}}) + \eta G(\bar{\mathbf{u}}) - P^\eta(\mathbf{u}).$$

So we have $\|\eta F(\bar{\mathbf{u}}) + \eta\bar{\mathbf{g}}\| \leq \eta\varepsilon$, thus $\|F(\bar{\mathbf{u}}) + \bar{\mathbf{g}}\| \leq \varepsilon$, where $\bar{\mathbf{g}} = \frac{1}{\eta}(\mathbf{u} - \bar{\mathbf{u}}) - F(\bar{\mathbf{u}}) \in G(\bar{\mathbf{u}})$. $\blacksquare$

In the rest of this section, for readability, we first provide, in Section C.1, the proofs for the convergence of Alg. 3 from Section 4.2. Then, in Section C.2, we give the proofs for the convergence of inexact Halpern iteration from Section 4.1.

### C.1 APPROXIMATING THE RESOLVENT

**Lemma C.1.** *Let $F : \mathbb{R}^d \to \mathbb{R}^d$ be monotone and $L_Q$-Lipschitz in expectation as in Assumption 2. Then for $\mathbf{u}^+ \in \mathbb{R}^d$, the function $\mathbf{u} \mapsto \bar{F}^\eta(\mathbf{u}; \mathbf{u}^+) = \eta F(\mathbf{u}) + \mathbf{u} - \mathbf{u}^+$ defined in Eq. (4.3) is 1-strongly monotone and $(\eta L_Q + 1)$-Lipschitz in expectation.*

*Proof.* Let $L = L_Q$ for brevity. For $\mathbf{u}^+ \in \mathbb{R}^d$, strong monotonicity clearly follows since for any $\mathbf{u}, \mathbf{v} \in \mathbb{R}^d$, we have

$$\langle \bar{F}^\eta(\mathbf{u}; \mathbf{u}^+) - \bar{F}^\eta(\mathbf{v}; \mathbf{u}^+), \mathbf{u} - \mathbf{v} \rangle = \eta\langle F(\mathbf{u}) - F(\mathbf{v}), \mathbf{u} - \mathbf{v} \rangle + \|\mathbf{u} - \mathbf{v}\|^2 \geq \|\mathbf{u} - \mathbf{v}\|^2,$$

where we use monotonicity of $F$ for the last inequality. Further, since $F$ is $L$-Lipschitz in expectation with oracle $F_\xi$ as Assumption 2, we have $\bar{F}_\xi^\eta(\mathbf{u}; \mathbf{u}^+) = \eta F_\xi(\mathbf{u}) + \mathbf{u} - \mathbf{u}^+$ such that $\mathbb{E}_{\xi\sim Q}[\bar{F}_\xi^\eta(\mathbf{u}; \mathbf{u}^+)] = \eta F(\mathbf{u}) + \mathbf{u} - \mathbf{u}^+ = \bar{F}^\eta(\mathbf{u}; \mathbf{u}^+)$. Then for any $\mathbf{u}, \mathbf{v} \in \mathbb{R}^d$,

$$\mathbb{E}_{\xi\sim Q}[\|\bar{F}_\xi^\eta(\mathbf{u}; \mathbf{u}^+) - \bar{F}_\xi^\eta(\mathbf{v}; \mathbf{u}^+)\|^2]$$

$$= \mathbb{E}_{\xi\sim Q}[\|\eta F_\xi(\mathbf{u}) - \eta F_\xi(\mathbf{v}) + \mathbf{u} - \mathbf{v}\|^2]$$

$$= \eta^2\mathbb{E}_{\xi\sim Q}[\|F_\xi(\mathbf{u}) - F_\xi(\mathbf{v})\|^2] + \|\mathbf{u} - \mathbf{v}\|^2 + 2\eta\mathbb{E}_{\xi\sim Q}[\langle F_\xi(\mathbf{u}) - F_\xi(\mathbf{v}), \mathbf{u} - \mathbf{v} \rangle].$$

Using $L$-Lipschitzness of $F$ in expectation and Cauchy-Schwarz inequality for the quantity above, we obtain

$$\mathbb{E}_{\xi\sim Q}[\|\bar{F}_\xi^\eta(\mathbf{u}; \mathbf{u}^+) - \bar{F}_\xi^\eta(\mathbf{v}; \mathbf{u}^+)\|^2]$$

$$\leq (\eta^2 L^2 + 1)\|\mathbf{u} - \mathbf{v}\|^2 + 2\eta\|\mathbf{u} - \mathbf{v}\|\mathbb{E}_{\xi\sim Q}[\|F_\xi(\mathbf{u}) - F_\xi(\mathbf{v})\|]$$

$$\overset{(i)}{\leq} (\eta^2 L^2 + 2\eta L + 1)\|\mathbf{u} - \mathbf{v}\|^2 = (\eta L + 1)^2\|\mathbf{u} - \mathbf{v}\|^2.$$

where $(i)$ is due to $\mathbb{E}_{\xi\sim Q}[\|F_\xi(\mathbf{u}) - F_\xi(\mathbf{v})\|] \leq L\|\mathbf{u} - \mathbf{v}\|$ by Assumption 2 and Jensen's inequality. Hence, $\bar{F}^\eta(\mathbf{u}; \mathbf{u}^+)$ is $(\eta L + 1)$-Lipschitz in expectation. This finishes the proof. $\blacksquare$

We now move to the convergence proof for `VR−FoRB` of Alacaoglu & Malitsky (2022) which incorporates the loopless SVRG variance reduction technique (Kovalev et al., 2020; Hofmann et al., 2015) into FoRB method (Malitsky & Tam, 2020).

**Theorem 4.6.** *Let* $A\colon \mathbb{R}^d \to \mathbb{R}^d$ *be monotone and* $L_A$-*Lipschitz in expectation with* $A = \frac{1}{n}\sum_{i=1}^n A_i$. *Let* $B\colon \mathbb{R}^d \rightrightarrows \mathbb{R}^d$ *be maximally monotone, and* $A + B$ *be* $\mu$-*strongly monotone with* $\mu > 0$ *and* $\mathbf{v}_* := (A+B)^{-1}(\mathbf{0}) \neq \emptyset$. *Given* $\bar{\varepsilon} > 0$, *Alg. 3 returns* $\mathbf{v}_M$ *with* $\mathbb{E}[\|\mathbf{v}_M - \mathbf{v}_*\|^2] \leq \bar{\varepsilon}^2$ *in* $\left\lceil 14\big(n + \frac{\sqrt{n}L_A}{\mu}\big) \log \frac{\sqrt{6}\|\mathbf{v}_0 - \mathbf{v}_*\|}{\bar{\varepsilon}} \right\rceil$ *iterations and* $\mathcal{O}\big((n + \frac{\sqrt{n}L_A}{\mu}) \log \frac{\|\mathbf{v}_0 - \mathbf{v}_*\|}{\bar{\varepsilon}}\big)$ *oracle queries.*

*Proof.* First, following the proof of (Alacaoglu & Malitsky, 2022, Theorem 22) with the addition of strong monotonicity of $A + B$ (*cf.* the equation after Eq. (56) in Alacaoglu & Malitsky (2022) which had used only monotonicity), and taking expectation w.r.t. all randomness on both sides, we plug in our parameter choices $\tau = \frac{\sqrt{p(1-p)}}{2L_A}$ and $\alpha = 1 - p$, and then obtain

$$
\begin{aligned}
&(1 - p + 2\tau\mu)\mathbb{E}[\|\mathbf{v}_{k+1} - \mathbf{v}_*\|^2] + \mathbb{E}[\|\mathbf{w}_{k+1} - \mathbf{v}_*\|^2] + p\mathbb{E}[\|\mathbf{v}_{k+1} - \mathbf{w}_k\|^2] \\
&+ 2\tau\mathbb{E}[\langle A(\mathbf{v}_{k+1}) - A(\mathbf{w}_k), \mathbf{v}_* - \mathbf{v}_{k+1}\rangle] \\
&\leq (1 - p)\mathbb{E}[\|\mathbf{v}_k - \mathbf{v}_*\|^2] + \mathbb{E}[\|\mathbf{w}_k - \mathbf{v}_*\|^2] + p\mathbb{E}[\|\mathbf{v}_k - \mathbf{w}_{k-1}\|^2] \\
&+ 2\tau\mathbb{E}[\langle A(\mathbf{v}_k) - A(\mathbf{w}_{k-1}), \mathbf{v}_* - \mathbf{v}_k\rangle] - \frac{p}{2}\mathbb{E}[\|\mathbf{v}_k - \mathbf{w}_{k-1}\|^2] \\
&- \frac{1-p}{2}\mathbb{E}[\|\mathbf{v}_{k+1} - \mathbf{v}_k\|^2].
\end{aligned}
\tag{C.1}
$$

To simplify the notation, we define

$$
\begin{aligned}
a_k &= \frac{1}{2}\mathbb{E}[\|\mathbf{v}_k - \mathbf{v}_*\|^2], \\
b_k &= \frac{1-p}{2}\mathbb{E}[\|\mathbf{v}_k - \mathbf{v}_*\|^2] + \mathbb{E}[\|\mathbf{w}_k - \mathbf{v}_*\|^2] + p\mathbb{E}[\|\mathbf{v}_k - \mathbf{w}_{k-1}\|^2] \\
&\quad + 2\tau\mathbb{E}[\langle A(\mathbf{v}_k) - A(\mathbf{w}_{k-1}), \mathbf{v}_* - \mathbf{v}_k\rangle].
\end{aligned}
$$

We first note that $b_k \geq 0$. Indeed, using Young's inequality, Lipschitzness of $A$ and the definition of $\tau$ in Algorithm 3, we have

$$
\begin{aligned}
|2\tau\langle A(\mathbf{v}_k) - A(\mathbf{w}_{k-1}), \mathbf{v}_* - \mathbf{v}_k\rangle| &\leq 2\tau L_A \|\mathbf{v}_k - \mathbf{w}_{k-1}\| \|\mathbf{v}_k - \mathbf{v}_*\| \\
&\leq \frac{4\tau^2 L_A^2}{2(1-p)}\|\mathbf{v}_k - \mathbf{w}_{k-1}\|^2 + \frac{1-p}{2}\|\mathbf{v}_k - \mathbf{v}_*\|^2 \\
&= \frac{p}{2}\|\mathbf{v}_k - \mathbf{w}_{k-1}\|^2 + \frac{1-p}{2}\|\mathbf{v}_k - \mathbf{v}_*\|^2.
\end{aligned}
\tag{C.2}
$$

Then we have, by plugging in the definitions of $a_k, b_k$ in Eq. (C.1) and discarding the last term on the right-hand side, that

$$
(1 - p + 4\tau\mu)a_{k+1} + b_{k+1} \leq (1 - p)a_k + b_k - \frac{p}{2}\mathbb{E}[\|\mathbf{v}_k - \mathbf{w}_{k-1}\|^2].
\tag{C.3}
$$

For a constant $c > 0$ to be set later, let us write the right-hand side of Eq. (C.3) as

$$
\begin{aligned}
(1 - p)a_k + b_k &= (1-p)(1+c)a_k + (1-c)b_k + c\mathbb{E}[\|\mathbf{w}_k - \mathbf{v}_*\|^2] + pc\mathbb{E}[\|\mathbf{v}_k - \mathbf{w}_{k-1}\|^2] \\
&\quad + 2\tau c\mathbb{E}[\langle A(\mathbf{v}_k) - A(\mathbf{w}_{k-1}), \mathbf{v}_* - \mathbf{v}_k\rangle].
\end{aligned}
\tag{C.4}
$$

By using the definition of $\mathbf{w}_k$ in Algorithm 3 and the tower rule, we have

$$
\begin{aligned}
c\mathbb{E}[\|\mathbf{w}_k - \mathbf{v}_*\|^2] &\leq 2c\mathbb{E}[\|\mathbf{v}_k - \mathbf{v}_*\|^2] + 2c\mathbb{E}[\|\mathbf{v}_k - \mathbf{w}_k\|^2] \\
&= 2c\mathbb{E}[\|\mathbf{v}_k - \mathbf{v}_*\|^2] + 2c(1-p)\mathbb{E}[\|\mathbf{v}_k - \mathbf{w}_{k-1}\|^2].
\end{aligned}
$$

Combining this estimate with Eq. (C.2), and using trivial facts that $cp \leq c$ and $1 - p \leq 1$ to further manipulate Eq. (C.4) gives

$$
\begin{aligned}
(1 - p)a_k + b_k &\leq (1-p)(1+c)a_k + (1-c)b_k + \frac{5c}{2}\mathbb{E}[\|\mathbf{v}_k - \mathbf{v}_*\|^2] + \frac{7c}{2}\mathbb{E}[\|\mathbf{v}_k - \mathbf{w}_{k-1}\|^2] \\
&\leq (1 - p + 6c)a_k + (1-c)b_k + \frac{7c}{2}\mathbb{E}[\|\mathbf{v}_k - \mathbf{w}_{k-1}\|^2].
\end{aligned}
\tag{C.5}
$$

We now let $\frac{7c}{2} \leq \frac{p}{2}$ and use the estimation Eq. (C.5) in Eq. (C.3) to get

$$(1 - p + 4\tau\mu)a_{k+1} + b_{k+1} \leq (1 - p + 6c)a_k + (1 - c)b_k. \tag{C.6}$$

Let us also set $6c \leq 3\tau\mu$ and then

$$
\begin{aligned}
(1 - p + 4\tau\mu)a_{k+1} + b_{k+1} &\leq (1 - p + 3\tau\mu)a_k + (1 - c)b_k \\
&= \Big(1 - \frac{\tau\mu}{1 - p + 4\tau\mu}\Big)(1 - p + 4\tau\mu)a_k + (1 - c)b_k \\
&\leq \Big(1 - \min\big\{\frac{\tau\mu}{1 + 4\tau\mu}, c\big\}\Big)\big((1 - p + 4\tau\mu)a_k + b_k\big),
\end{aligned}
$$

where the last step used nonnegativity of $b_k$. Let us also define $c = \min\big\{\frac{p}{7}, \frac{\tau\mu}{2}\big\}$. By iterating the inequality and using $b_k \geq 0$, we have

$$(1 - p + 4\tau\mu)a_k \leq \Big(1 - \min\big\{\frac{\tau\mu}{1 + 4\tau\mu}, c\big\}\Big)^k \big((1 - p + 4\tau\mu)a_0 + b_0\big) \tag{C.7}$$

thus

$$\mathbb{E}[\|\mathbf{v}_k - \mathbf{v}_*\|^2] \leq 2\Big(1 - \min\big\{\frac{\tau\mu}{1 + 4\tau\mu}, c\big\}\Big)^k \Big(a_0 + \frac{1}{1 - p + 4\tau\mu}b_0\Big).$$

Further using $\mathbf{v}_0 = \mathbf{w}_0 = \mathbf{w}_{-1}$ (see Alg. 3) and definitions of $a_k, b_k$, we have

$$
\begin{aligned}
a_0 + \frac{1}{1 - p + 4\tau\mu}b_0 &= \frac{1}{2}\|\mathbf{v}_0 - \mathbf{v}_*\|^2 + \frac{1}{1 - p + 4\tau\mu}\Big(\frac{1 - p}{2}\|\mathbf{v}_0 - \mathbf{v}_*\|^2 + \|\mathbf{v}_0 - \mathbf{v}_*\|^2\Big) \\
&\overset{(i)}{\leq} \Big(\frac{1}{2} + \frac{1}{2} + \frac{1}{1 - p}\Big)\|\mathbf{v}_0 - \mathbf{v}_*\|^2 = \frac{2 - p}{1 - p}\|\mathbf{v}_0 - \mathbf{v}_*\|^2,
\end{aligned}
$$

where we use $1 - p + 4\tau\mu \geq 1 - p$ for $(i)$. So we obtain

$$\mathbb{E}[\|\mathbf{v}_k - \mathbf{v}_*\|^2] \leq \Big(1 - \min\big\{\frac{\tau\mu}{1 + 4\tau\mu}, c\big\}\Big)^k \frac{4 - 2p}{1 - p}\|\mathbf{v}_0 - \mathbf{v}_*\|^2.$$

Given $\bar{\varepsilon} > 0$, we now see the exact number of iterations to guarantee $\mathbb{E}[\|\mathbf{v}_k - \mathbf{u}_*\|^2] \leq \bar{\varepsilon}^2$. We have

$$
\begin{aligned}
\mathbb{E}[\|\mathbf{v}_k - \mathbf{v}_*\|^2] &\leq \Big(1 - \min\big\{\frac{\tau\mu}{1 + 4\tau\mu}, c\big\}\Big)^k \frac{4 - 2p}{1 - p}\|\mathbf{v}_0 - \mathbf{v}_*\|^2 \\
&\leq 6\exp\Big(-\min\big\{\frac{\tau\mu}{1 + 4\tau\mu}, c\big\}k\Big)\|\mathbf{v}_0 - \mathbf{v}_*\|^2, \tag{C.8}
\end{aligned}
$$

where we use $p = \frac{1}{n}$ and assume $n \geq 2$ without loss of generality, thus

$$k \geq 2\max\Big\{\frac{1 + 4\tau\mu}{\tau\mu}, \frac{1}{c}\Big\}\log\Big(\frac{\sqrt{6}\|\mathbf{v}_0 - \mathbf{v}_*\|}{\bar{\varepsilon}}\Big) = 2\max\Big\{\frac{1 + 4\tau\mu}{\tau\mu}, \frac{7}{p}, \frac{2}{\tau\mu}\Big\}\log\frac{\sqrt{6}\|\mathbf{v}_0 - \mathbf{v}_*\|}{\bar{\varepsilon}},$$

using $c = \min\big\{\frac{p}{7}, \frac{\tau\mu}{2}\big\}$. Plugging in the choices that $p = \frac{1}{n}$ and $\tau = \frac{\sqrt{p(1-p)}}{2L_A} = \frac{\sqrt{n-1}}{2nL_A}$, we have

$$k \geq 2\max\Big\{4 + \frac{2nL_A}{\mu\sqrt{n-1}}, 7n, \frac{4nL_A}{\mu\sqrt{n-1}}\Big\}\log\Big(\frac{\sqrt{6}\|\mathbf{v}_0 - \mathbf{v}_*\|}{\bar{\varepsilon}}\Big), \tag{C.9}$$

For simplicity, we assume without loss of generality that $n \geq 2$, then we have $\frac{4nL_A}{\mu\sqrt{n-1}} \leq \frac{7\sqrt{n}L_A}{\mu}$. So it suffices to choose $k = \lceil 14\big(n + \frac{\sqrt{n}L_A}{\mu}\big)\log\frac{\sqrt{6}\|\mathbf{v}_0 - \mathbf{v}_*\|}{\bar{\varepsilon}}\rceil$. Further, we notice that Alg. 3 has constant expected per-iteration cost $pn + 2 = 3$ by $p = \frac{1}{n}$, so we have the oracle complexity to be $\mathcal{O}\Big(\big(n + \frac{\sqrt{n}L_A}{\mu}\big)\log\frac{\|\mathbf{v}_0 - \mathbf{v}_*\|}{\bar{\varepsilon}}\Big)$. ∎

### C.2 INEXACT HALPERN ITERATION

**Lemma C.2.** *Let $F$ be monotone and Lipschitz, and $G$ be maximally monotone. Assume that conditional on the algorithm randomness up to iteration $k$, we have $\mathbb{E}_k[\|\mathbf{e}_k\|^2] \leq \frac{\|P^\eta(\mathbf{u}_k)\|^2}{(k+2)^8} = \frac{\|\mathbf{u}_k - J_{\eta(F+G)}(\mathbf{u}_k)\|^2}{(k+2)^8}$ for all $k \geq 1$ and $\mathbb{E}[\|\mathbf{e}_0\|^2] \leq \frac{\|P^\eta(\mathbf{u}_0)\|^2}{27}$. Then the iterates $\mathbf{u}_k$ generated by Alg. 2 satisfy*

$$\mathbb{E}[\|\mathbf{u}_k - \mathbf{u}_*\|^2] \leq 2\|\mathbf{u}_0 - \mathbf{u}_*\|^2 \tag{C.10}$$

*where $\mathbf{u}_*$ is the solution point such that $J_{\eta(F+G)}(\mathbf{u}_*) = \mathbf{u}_*$.*

*Proof.* By Eq. (4.1), definition of $P^\eta$ and noticing $J_{\eta(F+G)}(\mathbf{u}_*) = \mathbf{u}_*$, we have

$$
\begin{aligned}
&\|\mathbf{u}_k - \mathbf{u}_*\| \\
&= \|\lambda_{k-1}(\mathbf{u}_0 - \mathbf{u}_*) + (1-\lambda_{k-1})(J_{\eta(F+G)}(\mathbf{u}_{k-1}) - J_{\eta(F+G)}(\mathbf{u}_*)) - (1-\lambda_{k-1})\mathbf{e}_{k-1}\| \\
&\overset{(i)}{\leq} \lambda_{k-1}\|\mathbf{u}_0 - \mathbf{u}_*\| + (1-\lambda_{k-1})\|J_{\eta(F+G)}(\mathbf{u}_{k-1}) - J_{\eta(F+G)}(\mathbf{u}_*)\| + (1-\lambda_{k-1})\|\mathbf{e}_{k-1}\| \\
&\overset{(ii)}{\leq} \lambda_{k-1}\|\mathbf{u}_0 - \mathbf{u}_*\| + (1-\lambda_{k-1})\|\mathbf{u}_{k-1} - \mathbf{u}_*\| + (1-\lambda_{k-1})\|\mathbf{e}_{k-1}\|,
\end{aligned}
$$

where we use the triangle inequality for $(i)$ and $(ii)$ is due to nonexpansiveness of resolvent. Iterating this inequality until $k = 1$, we obtain

$$
\begin{aligned}
\|\mathbf{u}_k - \mathbf{u}_*\| \leq &\underbrace{\left( \lambda_{k-1} + \prod_{i=1}^{k}(1-\lambda_{i-1}) + \sum_{i=1}^{k-1}\left(\lambda_{i-1}\prod_{j=i+1}^{k}(1-\lambda_{j-1})\right) \right)}_{\mathcal{T}_{[1]}}\|\mathbf{u}_0 - \mathbf{u}_*\| \\
&+ \sum_{i=1}^{k}\left( \|\mathbf{e}_{i-1}\| \underbrace{\prod_{j=i}^{k}(1-\lambda_{j-1})}_{\mathcal{T}_{[2]}^{(i)}} \right).
\end{aligned}
$$

Plugging in $\lambda_i = \frac{1}{i+2}$ gives

$$\mathcal{T}_{[1]} = \frac{1}{k+1} + \prod_{i=1}^{k}\frac{i}{i+1} + \sum_{i=1}^{k-1}\frac{1}{i+1}\frac{i+1}{k+1} = 1, \quad \mathcal{T}_{[2]}^{(i)} = \prod_{j=i}^{k}\frac{j}{j+1} = \frac{i}{k+1}$$

and consequently,

$$\|\mathbf{u}_k - \mathbf{u}_*\| \leq \|\mathbf{u}_0 - \mathbf{u}_*\| + \frac{1}{k+1}\sum_{i=1}^{k}i\|\mathbf{e}_{i-1}\|.$$

Squaring the terms on both sides and taking expectation w.r.t. all randomness on both sides, we obtain

$$
\begin{aligned}
\mathbb{E}[\|\mathbf{u}_k - \mathbf{u}_*\|^2] &\leq \mathbb{E}\left[\|\mathbf{u}_0 - \mathbf{u}_*\| + \frac{1}{k+1}\sum_{i=1}^{k}i\|\mathbf{e}_{i-1}\|\right]^2 \\
&\overset{(i)}{\leq} \frac{3}{2}\|\mathbf{u}_0 - \mathbf{u}_*\|^2 + \frac{3}{(k+1)^2}\mathbb{E}\left[\sum_{i=1}^{k}i\|\mathbf{e}_{i-1}\|\right]^2 \\
&\overset{(ii)}{\leq} \frac{3}{2}\|\mathbf{u}_0 - \mathbf{u}_*\|^2 + \frac{3}{k+1}\mathbb{E}\left[\sum_{i=1}^{k}i^2\|\mathbf{e}_{i-1}\|^2\right],
\end{aligned}
$$

where we use Young's inequality for $(i)$ and the fact $\left(\sum_{i=1}^{k}x_i\right)^2 \leq k\sum_{i=1}^{k}x_i^2$ for any $x_i \in \mathbb{R}$ for $(ii)$. Since $\mathbb{E}_k[\|\mathbf{e}_k\|^2] \leq \frac{\|P^\eta(\mathbf{u}_k)\|^2}{(k+2)^8} = \frac{\|\mathbf{u}_k - J_{\eta(F+G)}(\mathbf{u}_k)\|^2}{(k+2)^8}$ for all $k \geq 1$, and consequently,

$\mathbb{E}[\|\mathbf{e}_k\|^2] \leq \frac{\mathbb{E}[\|P^\eta(\mathbf{u}_k)\|^2]}{(k+2)^8}$ by tower rule, and $\mathbb{E}[\|\mathbf{e}_0\|^2] \leq \frac{\|P^\eta(\mathbf{u}_0)\|^2}{27}$, we have

$$\mathbb{E}[\|\mathbf{u}_k - \mathbf{u}_*\|^2] \leq \frac{3}{2}\|\mathbf{u}_0 - \mathbf{u}_*\|^2 + \frac{3}{k+1}\sum_{i=2}^{k} i^2 \frac{\mathbb{E}[\|P^\eta(\mathbf{u}_{i-1})\|^2]}{(i+1)^8} + \frac{\|P^\eta(\mathbf{u}_0)\|^2}{9(k+1)}$$

$$\leq \frac{3}{2}\|\mathbf{u}_0 - \mathbf{u}_*\|^2 + \frac{3}{k+1}\sum_{i=1}^{k-1} \frac{\mathbb{E}[\|P^\eta(\mathbf{u}_i)\|^2]}{(i+1)^6} + \frac{\|P^\eta(\mathbf{u}_0)\|^2}{9(k+1)}.$$

Noticing that $P^\eta(\mathbf{u}_*) = \mathbf{0}$ and $P^\eta$ is 2-Lipschitz by nonexpansiveness of the resolvent operator, we have for all $i \geq 0$

$$\|P^\eta(\mathbf{u}_i)\|^2 = \|P^\eta(\mathbf{u}_i) - P^\eta(\mathbf{u}_*)\|^2 \leq 4\|\mathbf{u}_i - \mathbf{u}_*\|^2,$$

which leads to

$$\mathbb{E}[\|\mathbf{u}_k - \mathbf{u}_*\|^2] \leq \frac{3}{2}\|\mathbf{u}_0 - \mathbf{u}_*\|^2 + \frac{12}{k+1}\sum_{i=1}^{k-1} \frac{\mathbb{E}[\|\mathbf{u}_i - \mathbf{u}_*\|^2]}{(i+1)^6} + \frac{4\|\mathbf{u}_0 - \mathbf{u}_*\|^2}{9(k+1)}. \qquad (\text{C.11})$$

We claim that for all $k \geq 1$:

$$\mathbb{E}[\|\mathbf{u}_k - \mathbf{u}_*\|^2] \leq 2\|\mathbf{u}_0 - \mathbf{u}_*\|^2.$$

We prove this claim by induction. First, for the base case $k = 1$, we have by Eq. (C.11)

$$\mathbb{E}[\|\mathbf{u}_1 - \mathbf{u}_*\|^2] \leq \frac{3}{2}\|\mathbf{u}_0 - \mathbf{u}_*\|^2 + \frac{2}{9}\|\mathbf{u}_0 - \mathbf{u}_*\|^2 \leq 2\|\mathbf{u}_0 - \mathbf{u}_*\|^2.$$

Suppose that $\mathbb{E}[\|\mathbf{u}_i - \mathbf{u}_*\|^2] \leq 2\|\mathbf{u}_0 - \mathbf{u}_*\|^2$ for all $i \leq k - 1$, then for the case $k \geq 2$, we have by Eq. (C.11)

$$\mathbb{E}[\|\mathbf{u}_k - \mathbf{u}_*\|^2] \leq \frac{3}{2}\|\mathbf{u}_0 - \mathbf{u}_*\|^2 + \frac{12}{k+1}\sum_{i=1}^{k-1} \frac{\mathbb{E}[\|\mathbf{u}_i - \mathbf{u}_*\|^2]}{(i+1)^6} + \frac{4\|\mathbf{u}_0 - \mathbf{u}_*\|^2}{9(k+1)}$$

$$\leq \frac{3}{2}\|\mathbf{u}_0 - \mathbf{u}_*\|^2 + \frac{24\|\mathbf{u}_0 - \mathbf{u}_*\|^2}{k+1}\Big(\sum_{i=1}^{\infty} \frac{1}{i^6} - 1\Big) + \frac{4\|\mathbf{u}_0 - \mathbf{u}_*\|^2}{9(k+1)}$$

$$\overset{(ii)}{\leq} 2\|\mathbf{u}_0 - \mathbf{u}_*\|^2,$$

where we use $k \geq 2$ and $\sum_{i=1}^{\infty} \frac{1}{i^6} = \frac{\pi^6}{945}$ for $(i)$. So our claim holds. ∎

**Lemma C.3.** *Let $F$ be monotone and Lipschitz and let $G$ be maximally monotone. Assume that conditional on the algorithm randomness up to iteration $k$, we have $\mathbb{E}_k[\|\mathbf{e}_k\|^2] \leq \frac{\|P^\eta(\mathbf{u}_k)\|^2}{(k+2)^8}$ for all $k \geq 1$ and $\mathbb{E}[\|\mathbf{e}_0\|^2] \leq \frac{\|P^\eta(\mathbf{u}_0)\|^2}{27}$. Then for the iterates $\mathbf{u}_k$ generated by Alg. 2, we have*

$$\mathbb{E}[\mathcal{C}_{k+1}] - \mathbb{E}[\mathcal{C}_k] \leq \frac{16\|\mathbf{u}_0 - \mathbf{u}_*\|^2}{(k+2)^2}, \qquad (\text{C.12})$$

*where*

$$\mathcal{C}_k = \frac{k(k+1)}{2}\|P^\eta(\mathbf{u}_k)\|^2 + (k+1)\langle P^\eta(\mathbf{u}_k), \mathbf{u}_k - \mathbf{u}_0 \rangle. \qquad (\text{C.13})$$

*Proof.* By Eq. (4.1) and the definition of $\mathbf{e}_k$, we have

$$\mathbf{u}_{k+1} - \mathbf{u}_k = \lambda_k(\mathbf{u}_0 - \mathbf{u}_k) - (1 - \lambda_k)P^\eta(\mathbf{u}_k) - (1 - \lambda_k)\mathbf{e}_k, \qquad (\text{C.14})$$

$$\mathbf{u}_{k+1} - \mathbf{u}_k = \frac{\lambda_k}{1 - \lambda_k}(\mathbf{u}_0 - \mathbf{u}_{k+1}) - P^\eta(\mathbf{u}_k) - \mathbf{e}_k. \qquad (\text{C.15})$$

Noticing that $P^\eta$ is $\frac{1}{2}$-cocoercive by the nonexpansiveness of the resolvent operator and using the equations above, we have

$$\frac{1}{2}\|P^\eta(\mathbf{u}_{k+1}) - P^\eta(\mathbf{u}_k)\|^2 \leq \langle P^\eta(\mathbf{u}_{k+1}) - P^\eta(\mathbf{u}_k), \mathbf{u}_{k+1} - \mathbf{u}_k \rangle$$

$$= \frac{\lambda_k}{1 - \lambda_k} \langle P^\eta(\mathbf{u}_{k+1}), \mathbf{u}_0 - \mathbf{u}_{k+1} \rangle - \langle P^\eta(\mathbf{u}_{k+1}), P^\eta(\mathbf{u}_k) + \mathbf{e}_k \rangle$$

$$- \lambda_k \langle P^\eta(\mathbf{u}_k), \mathbf{u}_0 - \mathbf{u}_k \rangle + (1 - \lambda_k)\big(\|P^\eta(\mathbf{u}_k)\|^2 + \langle P^\eta(\mathbf{u}_k), \mathbf{e}_k \rangle\big)$$

In this inequality, expanding $\|P^\eta(\mathbf{u}_{k+1}) - P^\eta(\mathbf{u}_k)\|^2$ and rearranging the terms, we obtain

$$\frac{1}{2}\|P^\eta(\mathbf{u}_{k+1})\|^2 + \frac{\lambda_k}{1 - \lambda_k} \langle P^\eta(\mathbf{u}_{k+1}), \mathbf{u}_{k+1} - \mathbf{u}_0 \rangle$$

$$\leq \left(\frac{1}{2} - \lambda_k\right)\|P^\eta(\mathbf{u}_k)\|^2 + \lambda_k \langle P^\eta(\mathbf{u}_k), \mathbf{u}_k - \mathbf{u}_0 \rangle + \langle \mathbf{e}_k, (1 - \lambda_k)P^\eta(\mathbf{u}_k) - P^\eta(\mathbf{u}_{k+1}) \rangle.$$

Plugging in $\lambda_k = \frac{1}{k+2}$ and multiplying by $(k+1)(k+2)$ on both sides, we can bound the consecutive change of $\mathcal{C}_k$ as below

$$\mathcal{C}_{k+1} - \mathcal{C}_k = \frac{(k+1)(k+2)}{2}\|P^\eta(\mathbf{u}_{k+1})\|^2 - \frac{k(k+1)}{2}\|P^\eta(\mathbf{u}_k)\|^2$$

$$+ (k+2)\langle P^\eta(\mathbf{u}_{k+1}), \mathbf{u}_{k+1} - \mathbf{u}_0 \rangle - (k+1)\langle P^\eta(\mathbf{u}_k), \mathbf{u}_k - \mathbf{u}_0 \rangle$$

$$\leq (k+1)(k+2)\left\langle \mathbf{e}_k, \frac{k+1}{k+2}P^\eta(\mathbf{u}_k) - P^\eta(\mathbf{u}_{k+1}) \right\rangle. \tag{C.16}$$

Noticing that $P^\eta(\mathbf{u}_*) = \mathbf{0}$ and $P^\eta$ is 2-Lipschitz by nonexpansiveness of the resolvent operator, we have

$$\left\langle \mathbf{e}_k, \frac{k+1}{k+2}P^\eta(\mathbf{u}_k) - P^\eta(\mathbf{u}_{k+1}) \right\rangle = \left\langle \mathbf{e}_k, \frac{k+1}{k+2}(P^\eta(\mathbf{u}_k) - P^\eta(\mathbf{u}_*)) - (P^\eta(\mathbf{u}_{k+1}) - P^\eta(\mathbf{u}_*)) \right\rangle$$

$$\overset{(i)}{\leq} 2\|\mathbf{e}_k\|\left(\frac{k+1}{k+2}\|\mathbf{u}_k - \mathbf{u}_*\| + \|\mathbf{u}_{k+1} - \mathbf{u}_*\|\right)$$

$$\overset{(ii)}{\leq} (k+2)^4\|\mathbf{e}_k\|^2 + \frac{2}{(k+2)^4}\big(\|\mathbf{u}_k - \mathbf{u}_*\|^2 + \|\mathbf{u}_{k+1} - \mathbf{u}_*\|^2\big),$$

where we use Cauchy-Schwarz inequality, triangle inequality and Lipschitzness of $P^\eta$ for $(i)$, and $(ii)$ is due to $\frac{k+1}{k+2} \leq 1$ and Young's inequality. With this and by taking expectation w.r.t. all randomness on both sides for Eq. (C.16) and using the results from Lemma C.2, we have for $k \geq 1$

$$\mathbb{E}[\mathcal{C}_{k+1}] - \mathbb{E}[\mathcal{C}_k] \leq (k+1)(k+2)\mathbb{E}\left[\left\langle \mathbf{e}_k, \frac{k+1}{k+2}P^\eta(\mathbf{u}_k) - P^\eta(\mathbf{u}_{k+1}) \right\rangle\right]$$

$$\leq (k+1)(k+2)\mathbb{E}\left[(k+2)^4\|\mathbf{e}_k\|^2 + \frac{2}{(k+2)^4}\big(\|\mathbf{u}_k - \mathbf{u}_*\|^2 + \|\mathbf{u}_{k+1} - \mathbf{u}_*\|^2\big)\right]$$

$$\overset{(i)}{\leq} \frac{4\mathbb{E}[\|\mathbf{u}_k - \mathbf{u}_*\|^2]}{(k+2)^2} + \frac{2}{(k+2)^2}\big(\mathbb{E}[\|\mathbf{u}_k - \mathbf{u}_*\|^2] + \mathbb{E}[\|\mathbf{u}_{k+1} - \mathbf{u}_*\|^2]\big)$$

$$\overset{(ii)}{\leq} \frac{16\|\mathbf{u}_0 - \mathbf{u}_*\|^2}{(k+2)^2},$$

where $(i)$ is due to $\mathbb{E}[\|\mathbf{e}_k\|^2] \leq \frac{\|P^\eta(\mathbf{u}_k)\|^2}{(k+2)^8}$ and $\|P^\eta(\mathbf{u}_k)\|^2 \leq 4\|\mathbf{u}_k - \mathbf{u}_*\|^2$ and we use $\mathbb{E}[\|\mathbf{u}_k - \mathbf{u}_*\|^2] \leq 2\|\mathbf{u}_0 - \mathbf{u}_*\|^2$ for $(ii)$. ∎

**Theorem 4.2.** *Let Assumptions 1 and 2 hold. Then, for the iterates $\mathbf{u}_k$ of Algorithm 2, we have that $\mathbb{E}_k[\|\mathbf{e}_k\|^2] \leq \frac{\|P^\eta(\mathbf{u}_k)\|}{(k+2)^8}$ conditional on the algorithm randomness up to iteration $k$, and*

$$\mathbb{E}[\|P^\eta(\mathbf{u}_k)\|] \leq \big(\mathbb{E}[\|P^\eta(\mathbf{u}_k)\|^2]\big)^{1/2} \leq \frac{7\|\mathbf{u}_0 - \mathbf{u}_*\|}{k}.$$

*Moreover, given accuracy $\varepsilon > 0$, to return a point $\mathbf{u}_K$ such that $\mathbb{E}[\|P^\eta(\mathbf{u}_K)\|] \leq \eta\varepsilon$ with $\eta = \frac{\sqrt{n}}{L}$, the stochastic oracle complexity is $\tilde{\mathcal{O}}\big(n + \frac{\sqrt{n}L\|\mathbf{u}_0 - \mathbf{u}_*\|}{\varepsilon}\big)$.*

*Proof.* We first prove $\mathbb{E}_k[\|\mathbf{e}_k\|^2] \leq \frac{\|P^\eta(\mathbf{u}_k)\|^2}{(k+2)^8}$ by our number of inner iterations in Alg. 2. Given $\varepsilon_k > 0$, noticing that $\|P^\eta(\mathbf{u}_k)\|^2 = \|\mathbf{u}_k - J_{\eta(F+G)}(\mathbf{u}_k)\|^2$ and using the convergence results in Theorem 4.6, we have our subsolver Alg. 3 with initial point $\mathbf{u}_k$ returns $\widetilde{J}_{\eta(F+G)}(\mathbf{u}_k)$ such that $\mathbb{E}_k[\|\widetilde{J}_{\eta(F+G)}(\mathbf{u}_k) - J_{\eta(F+G)}(\mathbf{u}_k)\|^2] \leq \varepsilon_k^2$ with number of iterations

$$\left\lceil 14\big(n + \sqrt{n}(\eta L + 1)\big) \log \frac{\sqrt{6}\|P^\eta(\mathbf{u}_k)\|}{\varepsilon_k} \right\rceil,$$

where we also use that $\bar{F}^\eta$ is 1-strongly monotone and $(\eta L + 1)$-Lipschitz in expectation (see Lem. C.1). So it suffices to choose $M_k \geq \left\lceil 56\big(n + \sqrt{n}(\eta L + 1)\big) \log\big(2(k+2)\big) \right\rceil$ to reach the accuracy $\varepsilon_k = \frac{\|P^\eta(\mathbf{u}_k)\|}{(k+2)^4}$.

Then with Assumptions 1 and 2, we have that the assumptions of Lemmas C.2 and C.3 hold. Summing up Eq. (C.12) in Lemma C.3 for $k = 1, \ldots, k-1$ gives

$$\mathbb{E}[\mathcal{C}_k] \leq \mathbb{E}[\mathcal{C}_1] + 16\|\mathbf{u}_0 - \mathbf{u}_*\|^2 \sum_{i=1}^{k-1} \frac{1}{(i+2)^2} \leq \mathbb{E}[\mathcal{C}_1] + 7\|\mathbf{u}_0 - \mathbf{u}_*\|^2. \tag{C.17}$$

Next, we bound $\mathbb{E}[\mathcal{C}_1]$. Recall that $P^\eta$ is $\frac{1}{2}$-cocoercive and $\mathbf{u}_1 - \mathbf{u}_0 = -\frac{1}{2}P^\eta(\mathbf{u}_0) - \frac{1}{2}\mathbf{e}_0$ by Eq. (C.14). Then, we have

$$\langle P^\eta(\mathbf{u}_1) - P^\eta(\mathbf{u}_0), \mathbf{u}_1 - \mathbf{u}_0 \rangle \geq \frac{1}{2}\|P^\eta(\mathbf{u}_1) - P^\eta(\mathbf{u}_0)\|^2$$

$$\iff \|P^\eta(\mathbf{u}_1)\|^2 \leq \langle P^\eta(\mathbf{u}_1), P^\eta(\mathbf{u}_0) \rangle - \langle \mathbf{e}_0, P^\eta(\mathbf{u}_1) - P^\eta(\mathbf{u}_0) \rangle,$$

which leads to

$$\begin{aligned}
\mathcal{C}_1 &= \|P^\eta(\mathbf{u}_1)\|^2 + 2\langle P^\eta(\mathbf{u}_1), \mathbf{u}_1 - \mathbf{u}_0 \rangle \\
&= \|P^\eta(\mathbf{u}_1)\|^2 - \langle P^\eta(\mathbf{u}_1), P^\eta(\mathbf{u}_0) + \mathbf{e}_0 \rangle \\
&\leq \langle \mathbf{e}_0, P^\eta(\mathbf{u}_0) - 2P^\eta(\mathbf{u}_1) \rangle. 
\end{aligned} \tag{C.18}$$

Note that

$$\begin{aligned}
\langle \mathbf{e}_0, P^\eta(\mathbf{u}_0) - 2P^\eta(\mathbf{u}_1) \rangle &= 2\langle \mathbf{e}_0, P^\eta(\mathbf{u}_0) - P^\eta(\mathbf{u}_1) \rangle - \langle \mathbf{e}_0, P^\eta(\mathbf{u}_0) \rangle \\
&\overset{(i)}{\leq} 3\|\mathbf{e}_0\|^2 + \frac{1}{2}\|P^\eta(\mathbf{u}_0) - P^\eta(\mathbf{u}_1)\|^2 + \frac{1}{4}\|P^\eta(\mathbf{u}_0)\|^2 \\
&\overset{(ii)}{\leq} 3\|\mathbf{e}_0\|^2 + 2\|\mathbf{u}_0 - \mathbf{u}_1\|^2 + \frac{1}{4}\|P^\eta(\mathbf{u}_0)\|^2, 
\end{aligned} \tag{C.19}$$

where we use Young's inequality for $(i)$; and $(ii)$ is due to $P^c$ being 2-Lipschitz. For the second term in the right-hand side of (C.19), we use $\mathbf{u}_1 - \mathbf{u}_0 = -\frac{1}{2}P^\eta(\mathbf{u}_0) - \frac{1}{2}\mathbf{e}_0$ and Young's inequality again, and obtain

$$2\|\mathbf{u}_0 - \mathbf{u}_1\|^2 = \frac{1}{2}\|P^\eta(\mathbf{u}_0) + \mathbf{e}_0\|^2 \leq \frac{3}{2}\|\mathbf{e}_0\|^2 + \frac{3}{4}\|P^\eta(\mathbf{u}_0)\|^2.$$

With this, (C.19) becomes

$$\langle \mathbf{e}_0, P^\eta(\mathbf{u}_0) - 2P^\eta(\mathbf{u}_1) \rangle \leq \frac{9}{2}\|\mathbf{e}_0\|^2 + \|P^\eta(\mathbf{u}_0)\|^2.$$

Taking expectation w.r.t. all randomness on both sides and noticing that $\mathbb{E}[\|\mathbf{e}_0\|^2] \leq \frac{\|P^\eta(\mathbf{u}_0)\|^2}{27}$ and $\|P^\eta(\mathbf{u}_0)\|^2 \leq 4\|\mathbf{u}_0 - \mathbf{u}_*\|^2$, we have

$$\mathbb{E}[\langle \mathbf{e}_0, P^\eta(\mathbf{u}_0) - 2P^c(\mathbf{u}_1) \rangle] \leq 5\|\mathbf{u}_0 - \mathbf{u}_*\|^2,$$

which, in view of (C.18), leads to

$$\mathbb{E}[\mathcal{C}_1] \leq \mathbb{E}[\langle \mathbf{e}_0, P^\eta(\mathbf{u}_0) - 2P^\eta(\mathbf{u}_1) \rangle] \leq 5\|\mathbf{u}_0 - \mathbf{u}_*\|^2.$$

As a result, we obtain in (C.17) that

$$\mathbb{E}[\mathcal{C}_k] \leq \mathbb{E}[\mathcal{C}_1] + 7\|\mathbf{u}_0 - \mathbf{u}_*\|^2 \leq 12\|\mathbf{u}_0 - \mathbf{u}_*\|^2. \tag{C.20}$$

On the other hand, since $P^\eta$ is monotone and $P^\eta(\mathbf{u}_*) = \mathbf{0}$, we have

$$
\begin{aligned}
\langle P^\eta(\mathbf{u}_k), \mathbf{u}_k - \mathbf{u}_0 \rangle &= \langle P^\eta(\mathbf{u}_k) - P^\eta(\mathbf{u}_*), \mathbf{u}_k - \mathbf{u}_* \rangle + \langle P^\eta(\mathbf{u}_k), \mathbf{u}_* - \mathbf{u}_0 \rangle \\
&\geq \langle P^\eta(\mathbf{u}_k), \mathbf{u}_* - \mathbf{u}_0 \rangle \\
&\geq - \|P^\eta(\mathbf{u}_k)\| \|\mathbf{u}_0 - \mathbf{u}_*\|,
\end{aligned}
$$

where we use Cauchy-Schwarz inequality.

In view of the definition of $\mathcal{C}_k$ on (3.2), the last estimation gives us the bound

$$
\mathbb{E}[\mathcal{C}_k] \geq \frac{k(k+1)}{2} \mathbb{E}[\|P^\eta(\mathbf{u}_k)\|^2] - (k+1)\|\mathbf{u}_0 - \mathbf{u}_*\| \mathbb{E}[\|P^\eta(\mathbf{u}_k)\|].
$$

Using this bound in (C.20) gives

$$
\frac{k(k+1)}{2} \mathbb{E}[\|P^\eta(\mathbf{u}_k)\|^2] \leq (k+1)\|\mathbf{u}_0 - \mathbf{u}_*\| \mathbb{E}[\|P^\eta(\mathbf{u}_k)\|] + 12\|\mathbf{u}_0 - \mathbf{u}_*\|^2.
$$

By Jensen's inequality, we have

$$
\frac{k(k+1)}{2} \mathbb{E}[\|P^\eta(\mathbf{u}_k)\|^2] \leq (k+1)\|\mathbf{u}_0 - \mathbf{u}_*\| \left(\mathbb{E}[\|P^\eta(\mathbf{u}_k)\|^2]\right)^{1/2} + 12\|\mathbf{u}_0 - \mathbf{u}_*\|^2.
$$

By the larger root of this quadratic inequality w.r.t. $\left(\mathbb{E}[\|P^\eta(\mathbf{u}_k)\|^2]\right)^{1/2}$, we obtain for $k \geq 1$

$$
\begin{aligned}
\left(\mathbb{E}[\|P^\eta(\mathbf{u}_k)\|^2]\right)^{1/2} &\leq \frac{\|\mathbf{u}_0 - \mathbf{u}_*\|}{k} + \sqrt{\frac{\|\mathbf{u}_0 - \mathbf{u}_*\|^2}{k^2} + \frac{24\|\mathbf{u}_0 - \mathbf{u}_*\|^2}{k(k+1)}} \\
&\overset{(i)}{\leq} \frac{2\|\mathbf{u}_0 - \mathbf{u}_*\|}{k} + \frac{2\sqrt{6}\|\mathbf{u}_0 - \mathbf{u}_*\|}{\sqrt{k(k+1)}} \\
&\leq \frac{7\|\mathbf{u}_0 - \mathbf{u}_*\|}{k}.
\end{aligned}
$$

where $(i)$ is due to the elementary inequality $\sqrt{a+b} \leq \sqrt{a} + \sqrt{b}$.

Hence, to guarantee that Algorithm 2 returns a point $\mathbf{u}_K$ such that $\mathbb{E}[\|P^\eta(\mathbf{u}_K)\|^2] \leq \eta^2 \varepsilon^2$, which implies $\mathbb{E}[\|P^\eta(\mathbf{u}_K)\|] \leq \eta \varepsilon$ by Jensen's inequality, we need $K = \left\lceil \frac{7\|\mathbf{u}_0 - \mathbf{u}_*\|}{\eta \varepsilon} \right\rceil$ outer iterations.

We now look at the cost of each inner loop to estimate $J_{\eta(F+G)}(\mathbf{u}_k)$. Notice that for the subsolver $\mathtt{VR-FoRB}$, we take the number of inner iterations $M_k = \left\lceil 28 \max\left(n + \sqrt{n}(\eta L + 1)\right) \log\left(2(k+2)\right) \right\rceil$ with the constant oracle query cost (in expectation) for one iteration of $\mathtt{VR-FoRB}$. Hence, we have $\mathcal{O}\left((n + \sqrt{n}(\eta L + 1)) \log(k+2)\right)$ oracle queries for $k$-th inner loop. Then for the total number of oracle queries, we note that

$$
\sum_{k=1}^{K} \left((n + \sqrt{n}(\eta L + 1)) \log(k+2)\right) = \mathcal{O}\left((n + \sqrt{n}(\eta L + 1)) K \log(K + 2)\right).
$$

Plugging in the choice of $K$ and suppressing the logarithm terms, we obtain $\tilde{\mathcal{O}}\left((n + \sqrt{n}(\eta L + 1))(\frac{\|\mathbf{u}_0 - \mathbf{u}_*\|}{\eta \varepsilon} + 1)\right)$ oracle complexity. Taking $\eta = \frac{\sqrt{n}}{L}$, we have the claimed result. ∎

## C.3  ESTIMATION OF THE OUTPUT

**Corollary 4.3.** *Let Assumptions 1 and 2 hold and let $\mathbf{u}_K$ be as defined in Theorem 4.2. Then, for $\mathbf{u}_{\mathrm{out}} = \mathtt{VR-FoRB}(\mathbf{u}_K, \lceil 42(n + \sqrt{n}) \log(19n) \rceil, \mathrm{Id} + \eta(F+G) - \mathbf{u}_K, Q)$ with $\eta = \frac{\sqrt{n}}{L}$,*

$$
\mathbb{E}[\mathrm{Res}_{F+G}(\mathbf{u}_{\mathrm{out}})] \leq 2\varepsilon.
$$

*The total stochastic oracle complexity for producing $\mathbf{u}_{\mathrm{out}}$ is $\widetilde{\mathcal{O}}\left(n + \frac{\sqrt{n}L\|\mathbf{u}_0 - \mathbf{u}_*\|}{\varepsilon}\right)$.*

*Proof.* We combine Lemma C.4 and Theorem 4.2. ∎

**Lemma C.4.** *Let Assumptions 1 and 2 hold and $\mathbf{u}_k$ be such that $(\mathbb{E}[\|P^\eta(\mathbf{u}_k)\|^2])^{1/2} \leq \eta\varepsilon$ with $\eta = \frac{\sqrt{n}}{L}$. Then, for $\mathbf{v}_{\text{out}}$ outputted by $\mathtt{VR-FoRB}(\mathbf{u}_k, M, \mathrm{Id} + \eta(F+G) - \mathbf{u}_k)$, we have that*

$$\mathbb{E}[\mathrm{Res}_{F+G}(\mathbf{v}_{\text{out}})] \leq 2\varepsilon, \tag{C.21}$$

*where $M = \lceil 42(n + \sqrt{n}) \log(19n) \rceil$ and complexity of this step is $\mathcal{O}(n \log n)$.*

*Proof.* Let us denote $\mathbf{u}_k^* = J_{\eta(F+G)}(\mathbf{u}_k)$, i.e., $(\mathbb{E}[\|P^\eta(\mathbf{u}_k)\|^2])^{1/2} = (\mathbb{E}[\|\mathbf{u}_k - \mathbf{u}_k^*\|^2])^{1/2} \leq \eta\varepsilon$, and consider the uniform sampling for brevity. Note that we have in this case $A(\mathbf{u}) = \eta F(\mathbf{u}) + \mathbf{u} - \mathbf{u}_k$, $A_i(\mathbf{u}) = \eta F_i(\mathbf{u}) + \mathbf{u} - \mathbf{u}_k$ and $B = \eta G$. By the update rule of $\mathtt{VR-FoRB}$ (where we use the index $t$ for the inner loop to prevent confusion), we have $\alpha = 1 - p$ and then for $t \geq 1$

$$\mathbf{v}_{t+1} + \tau B(\mathbf{v}_{t+1}) \ni (1-p)\mathbf{v}_t + p\mathbf{w}_t - \tau\tilde{A}(\mathbf{v}_t)$$

$$\iff A(\mathbf{v}_{t+1}) + B(\mathbf{v}_{t+1}) \ni \frac{1-p}{\tau}(\mathbf{v}_t - \mathbf{v}_{t+1}) + \frac{p}{\tau}(\mathbf{w}_t - \mathbf{v}_{t+1}) + A(\mathbf{v}_{t+1}) - \tilde{A}(\mathbf{v}_t),$$

where $\tilde{A}(\mathbf{v}_t) = A(\mathbf{w}_t) - A_i(\mathbf{w}_{t-1}) + A_i(\mathbf{v}_t)$ and we also have the implicit definition $\eta\mathbf{g}_{t+1} = \frac{(1-p)}{\tau}\mathbf{v}_t + \frac{p}{\tau}\mathbf{w}_t - \tilde{A}(\mathbf{v}_t) - \frac{1}{\tau}\mathbf{v}_{t+1} \in B(\mathbf{v}_{t+1}) = \eta G(\mathbf{v}_{t+1})$ since $B = \eta G$.

By using the definitions $A(\mathbf{u}) = \eta F(\mathbf{u}) + \mathbf{u} - \mathbf{u}_k$, $A_i(\mathbf{u}) = \eta F_i(\mathbf{u}) + \mathbf{u} - \mathbf{u}_k$, $\tilde{A}(\mathbf{v}_t) = \eta F(\mathbf{w}_t) + \mathbf{w}_t - \mathbf{u}_k - \eta F_i(\mathbf{w}_{t-1}) - \mathbf{w}_{t-1} + \eta F_i(\mathbf{v}_t) + \mathbf{v}_t$ and rearranging we get

$$\eta F(\mathbf{v}_{t+1}) + \eta\mathbf{g}_{t+1} = \frac{1-p}{\tau}(\mathbf{v}_t - \mathbf{v}_{t+1}) + \frac{p}{\tau}(\mathbf{w}_t - \mathbf{v}_{t+1})$$
$$+ \eta F(\mathbf{v}_{t+1}) - \eta F(\mathbf{w}_t)$$
$$+ \eta F_i(\mathbf{w}_{t-1}) - \eta F_i(\mathbf{v}_t)$$
$$+ (\mathbf{u}_k - \mathbf{v}_t) + (\mathbf{w}_{t-1} - \mathbf{w}_t).$$

Note that we have $\mathbb{E}[\|\mathbf{w}_t - \mathbf{v}_t\|] = (1-p)\mathbb{E}[\|\mathbf{w}_{t-1} - \mathbf{v}_t\|]$ and $\mathbb{E}[\|\mathbf{w}_t - \mathbf{w}_{t-1}\|] = p\mathbb{E}[\|\mathbf{v}_t - \mathbf{w}_{t-1}\|]$ by Alg. 3 and the tower rule. As a result, triangle inequalities and Lipschitzness of $F$ give

$$\eta\mathbb{E}[\mathrm{Res}_{F+G}(\mathbf{v}_{t+1})] \leq \frac{1-p}{\tau}\mathbb{E}[\|\mathbf{v}_t - \mathbf{v}_{t+1}\|] + \left(\frac{p}{\tau} + \eta L_F\right)\mathbb{E}[\|\mathbf{v}_{t+1} - \mathbf{w}_t\|]$$
$$+ \eta L\mathbb{E}[\|\mathbf{v}_t - \mathbf{w}_{t-1}\|] + \mathbb{E}[\|\mathbf{v}_t - \mathbf{u}_k^*\|] + \mathbb{E}[\|\mathbf{u}_k - \mathbf{u}_k^*\|]$$
$$+ \mathbb{E}[\|\mathbf{w}_t - \mathbf{w}_{t-1}\|]$$
$$\leq \left(\frac{1}{\tau} + \eta L_F + 1\right)\mathbb{E}[\|\mathbf{v}_t - \mathbf{v}_{t+1}\|]$$
$$+ \left(\frac{(1-p)p}{\tau} + (1-p)\eta L_F + \eta L + p\right)\mathbb{E}[\|\mathbf{v}_t - \mathbf{w}_{t-1}\|]$$
$$+ \mathbb{E}[\|\mathbf{v}_{t+1} - \mathbf{u}_k^*\|] + \mathbb{E}[\|\mathbf{u}_k - \mathbf{u}_k^*\|]$$
$$\leq \left(2\tau^{-1} + 3\eta L + 3\right)\bar{\varepsilon} + \eta\varepsilon, \tag{C.22}$$

where the last step is because $L_F \leq L$ and given accuracy $\bar{\varepsilon} > 0$, the output of $\mathtt{VR-FoRB}$ gives for $t = M$

$$\mathbb{E}[\|\mathbf{v}_{t+1} - \mathbf{u}_k^*\|^2] \leq \bar{\varepsilon}^2, \quad \mathbb{E}[\|\mathbf{v}_{t+1} - \mathbf{v}_t\|^2] \leq \bar{\varepsilon}^2, \quad \mathbb{E}[\|\mathbf{v}_t - \mathbf{w}_{t-1}\|^2] \leq \bar{\varepsilon}^2. \tag{C.23}$$

We now bound the oracle complexity and the number of iterations to get these bounds in (C.23).

Denote by $\mathbb{E}_k$ the expectation conditioned on all the randomness up to and including $\mathbf{u}_k$. Recall that $A$ is $(\eta L + 1)$-Lipschitz in expectation and $A + B$ is 1-strongly monotone, and we have our parameters to be $p = \frac{1}{n}$, $\eta = \frac{\sqrt{n}}{L}$ and $\tau\mu = \frac{\sqrt{p(1-p)}}{2(\eta L+1)} = \frac{\sqrt{n-1}}{2n(\sqrt{n}+1)} \leq \frac{1}{2n}$. Then following the same derivation from Eq. (C.1) to Eq. (C.5) in the proof of Theorem 4.6, we first do not discard the last term on the right-hand side of Eq. (C.1), and obtain (*cf.* (C.3))

$$(1 - p + 4\tau\mu)a_{t+1} + b_{t+1} \leq (1-p)a_t + b_t$$
$$- \frac{p}{2}\mathbb{E}_k[\|\mathbf{v}_t - \mathbf{w}_{t-1}\|^2] - \frac{1-p}{2}\mathbb{E}_k[\|\mathbf{v}_{t+1} - \mathbf{v}_t\|^2]. \tag{C.24}$$

Further, without using $p \leq 1$ in (C.4), we have (*cf.* (C.5))

$$(1-p)a_t + b_t \leq (1-p+6c)a_t + (1-c)b_t + \left(\frac{3cp}{2} + 2c\right) \mathbb{E}_k[\|\mathbf{v}_t - \mathbf{w}_{t-1}\|^2]$$

$$\leq (1-p+6c)a_t + (1-c)b_t + \frac{3p^2 + 4p}{14}\mathbb{E}_k[\|\mathbf{v}_t - \mathbf{w}_{t-1}\|^2],$$

where the last line used $c \leq \frac{p}{7}$. Combining with (C.24) gives (*cf.* (C.6))

$$(1-p+4\tau\mu)a_{t+1} + b_{t+1} \leq (1-p+6c)a_t + (1-c)b_t$$
$$- \frac{3p(1-p)}{14}\mathbb{E}_k[\|\mathbf{v}_t - \mathbf{w}_{t-1}\|^2] - \frac{1-p}{2}\mathbb{E}_k[\|\mathbf{v}_{t+1} - \mathbf{v}_t\|^2].$$
(C.25)

With the same derivation as obtaining (C.8), we have after using $4\tau\mu \geq 0$ and $4\tau\mu \leq 2p$

$$\frac{1-p}{2}\mathbb{E}_k[\|\mathbf{v}_{t+1} - \mathbf{v}_t\|^2] + \frac{3p(1-p)}{14}\mathbb{E}_k[\|\mathbf{v}_t - \mathbf{w}_{t-1}\|^2] + \frac{1-p}{2}\mathbb{E}_k[\|\mathbf{v}_{t+1} - \mathbf{u}_k^*\|^2]$$
$$\leq \frac{5}{2}\exp\left(-\min\left\{\frac{\tau\mu}{1+4\tau\mu}, c\right\}(t+1)\right)\|\mathbf{v}_0 - \mathbf{u}_k^*\|^2$$

where we also use that the solution of the inner subproblem is $\mathbf{u}_k^*$. Unrolling the expectation and using that VR$-$FoRB is initialized as $\mathbf{v}_0 = \mathbf{u}_k$, we have

$$\frac{3p(1-p)}{14}\left(\mathbb{E}[\|\mathbf{v}_{t+1} - \mathbf{v}_t\|^2] + \mathbb{E}[\|\mathbf{v}_t - \mathbf{w}_{t-1}\|^2] + \mathbb{E}[\|\mathbf{v}_{t+1} - \mathbf{u}_k^*\|^2]\right)$$
$$\leq \frac{5}{2}\exp\left(-\min\left\{\frac{\tau\mu}{1+4\tau\mu}, c\right\}(t+1)\right)\mathbb{E}[\|\mathbf{u}_k - \mathbf{u}_k^*\|^2]$$
$$\leq \frac{5}{2}\exp\left(-\min\left\{\frac{\tau\mu}{1+4\tau\mu}, c\right\}(t+1)\right)\eta^2\varepsilon^2,$$
(C.26)

where the last step is by $\mathbb{E}[\|\mathbf{u}_k - J_{\eta(F+G)}(\mathbf{u}_k)\|^2] = \mathbb{E}[\|\mathbf{u}_k - \mathbf{u}_k^*\|^2] \leq \eta^2\varepsilon^2$. Hence, as in the end of Theorem 4.6, we have that (C.23) holds in $\lceil 7\left(n + \sqrt{n}(\eta L + 1)\right)\log\left(\frac{35\eta^2\varepsilon^2}{3p(1-p)\bar{\varepsilon}^2}\right)\rceil$ iterations and with complexity $\mathcal{O}\left(n\log\left(\frac{n\varepsilon}{L\bar{\varepsilon}}\right)\right)$ as $\eta = \frac{\sqrt{n}}{L}$. In particular, we choose $\bar{\varepsilon} = \frac{\eta\varepsilon}{2\tau^{-1}+3\eta L+3} = \frac{\varepsilon}{\frac{4nL(\sqrt{n}+1)}{\sqrt{n(n-1)}}+3L+\frac{3L}{\sqrt{n}}}$ on (C.22) we get (C.21), and also have $\frac{\varepsilon}{\bar{\varepsilon}} \leq 10\sqrt{n}L + 6L$. Plugging in this value gives the number of iterations as $\lceil 42(n + \sqrt{n})\log(19n)\rceil$ and overall complexity as $\mathcal{O}(n\log n)$. ∎

## C.4 PROOF FOR THE COHYPOMONOTONE EXTENSION

**Corollary 4.5.** *[Cohypomonotone] Assume that $F$ is maximally $\rho$-cohypomonotone and $L$-expected Lipschitz and $G \equiv 0$. Given $\varepsilon > 0$, Alg. 2 returns a point $\mathbf{u}_K$ such that $\left(\mathbb{E}[\|P^\eta(\mathbf{u}_K)\|^2]\right)^{1/2} \leq \eta\varepsilon$ with $\widetilde{O}\left(\left(n + \sqrt{n}\frac{\eta L+1}{1-\rho\eta L_F^2}\right)\left(\frac{\|\mathbf{u}_0 - \mathbf{u}_*\|}{\eta\varepsilon} + 1\right)\right)$ stochastic oracle complexity, for any positive $\eta$ such that $\rho < \min\left(\frac{\eta}{2}, \frac{1}{\eta L_F^2}\right)$. With $\eta = \frac{\sqrt{n}}{L}$ as before, this corresponds to $\rho < \min\left(\frac{L}{\sqrt{n}L_F^2}, \frac{\sqrt{n}}{2L}\right)$.*

*Proof.* By definition, we have that $\eta F$ is maximally $\frac{\rho}{\eta}$-cohypomonotone. When $\frac{\rho}{\eta} \leq \frac{1}{2}$, we have that $J_{\eta F}$ is single-valued and nonexpansive (Bauschke et al., 2021, Prop. 3.7, Thm. 2.17). Subproblem in this case is finding $\bar{\mathbf{u}}$ such that

$$\mathbf{0} \in (\mathrm{Id} + \eta F)(\bar{\mathbf{u}}) - \mathbf{u}_k.$$

Since $F$ is $\rho$-cohypomonotone and $L_F$-Lipschitz, we have that

$$\langle F(\mathbf{u}) - F(\mathbf{v}), \mathbf{u} - \mathbf{v}\rangle \geq -\rho L_F^2\|\mathbf{u} - \mathbf{v}\|^2.$$

As a result, our subproblem is $(1 - \rho\eta L_F^2)$ strongly monotone and $(\eta L + 1)$-Lipschitz in expectation (see Lem. C.1).

In summary, to ensure that the resolvent is nonexpansive, single-valued and the subproblem is strongly monotone, we require

$$\frac{\rho}{\eta} \le \frac{1}{2} \quad \text{and} \quad \rho\eta L_F^2 < 1,$$

Of course, these bounds are optimized with $\eta = \frac{\sqrt{2}}{L_F}$, which leads to the requirement $\rho < \frac{1}{\sqrt{2}L_F}$. However, this choice of $\eta$ does not give the best oracle complexity with finite-sum form. In particular, by using a standard deterministic extragradient algorithm as subsolver in our framework (see e.g., (Diakonikolas, 2020, App A.3) for a proof for extragradient, with additional adaptivity that one can drop for simplicity), this would give complexity $\widetilde{O}\left(\frac{nL_F}{\varepsilon}\left(\frac{1}{1-\sqrt{2}\rho L_F}\right)\right)$. Using a variance reduced solver with $\eta = \frac{\sqrt{2}}{L_F}$ does not improve this.

Hence, we pick $\eta = \frac{\sqrt{n}}{L}$ as before. We can then use the same estimations as in the proof of Theorem 4.2 by only changing the strong monotonicity parameter for the inner subproblem which affects the complexity of the inner subsolver and hence the final complexity. ∎

## C.5 Details about Remark 4.4

We now derive the bounds for $L_F$ and $L_Q$ for this important special case covering matrix games and linearly constrained convex optimization.

It is straightforward from the definition $F(\mathbf{u}) = \begin{pmatrix} \boldsymbol{A}^\top\mathbf{y} \\ -\boldsymbol{A}\mathbf{x} \end{pmatrix}$ that

$$L_F = \|\boldsymbol{A}\|_2.$$

Following Carmon et al. (2019) and Alacaoglu & Malitsky (2022), we derive for any $\mathbf{u} = \begin{pmatrix} \mathbf{x} \\ \mathbf{y} \end{pmatrix}$,

$$
\begin{aligned}
\mathbb{E}_{\xi \sim Q}\|F_\xi(\mathbf{u})\|_2^2 &= \sum_{i=1}^{m_2} q_i^{(1)} \times \frac{1}{(q_i^{(1)})^2}\|\boldsymbol{A}_{i:}\mathbf{y}_i\|_2^2 + \sum_{j=1}^{m_1} q_j^{(2)} \times \frac{1}{(q_j^{(2)})^2}\|\boldsymbol{A}_{:j}\mathbf{x}_j\|_2^2 \\
&= \sum_{i=1}^{m_2} \frac{1}{q_i^{(1)}}\|\boldsymbol{A}_{i:}\mathbf{y}_i\|_2^2 + \sum_{j=1}^{m_1} \frac{1}{q_j^{(2)}}\|\boldsymbol{A}_{:j}\mathbf{x}_j\|_2^2 \\
&= \sum_{i=1}^{m_2} \frac{1}{q_i^{(1)}}\|\boldsymbol{A}_{i:}\|_2^2(\mathbf{y}_i)^2 + \sum_{j=1}^{m_1} \frac{1}{q_j^{(2)}}\|\boldsymbol{A}_{:j}\|_2^2(\mathbf{x}_j)^2 \\
&= \|\boldsymbol{A}\|_F^2 \left(\sum_{i=1}^{m_2}(\mathbf{y}_i)^2 + \sum_{j=1}^{m_1}(\mathbf{x}_j)^2\right) \\
&= \|\boldsymbol{A}\|_F^2\|\mathbf{u}\|_2^2.
\end{aligned}
$$

In view of this derivation, linearity of $F_\xi$ and Assumption 2, we conclude that

$$L_Q = \|\boldsymbol{A}\|_F. \tag{C.27}$$

As a result, we have in this case that $L_Q \le \sqrt{\operatorname{rank}(\boldsymbol{A})}L_F$.

In our analyses of this section, we use the simplified assumption that computation of $F_i$ is $n$ times cheaper than $F = \frac{1}{n}\sum_{i=1}^{n} F_i$ which gave the choice $p = \frac{1}{n}$. This is the most natural assumption given a generic $F$. This was the setting also in previous works such as Carmon et al. (2019); Alacaoglu & Malitsky (2022) when dealing with a general $F$ with a finite sum form. On the other hand, our bounds could have been also written in terms of $p$ (the probability for full operator evaluations for $\mathbf{w}_{k+1}$ in Alg. 3), as in Alacaoglu & Malitsky (2022). In this case, the general form for $p$ in terms of the costs of $F$ and $F_i$, denoted for simplicity as $\operatorname{Cost}(F)$ and $\operatorname{Cost}(F_i)$ would be $p = \frac{\operatorname{Cost}(F_i)}{\operatorname{Cost}(F)}$.

However, for specific examples such as matrix games or linearly constrained optimization, we would use $p = \frac{m_1+m_2}{2m_1 m_2}$ given a dense matrix $\boldsymbol{A}$. We refer to Carmon et al. (2019); Palaniappan & Bach (2016); Alacaoglu & Malitsky (2022) for more details about this representation. This choice gives rise to the claimed complexity improvements in Remark 4.4 in view of the derivation of $L_F$ and $L_Q$ provided above.

## D    EXPERIMENT DETAILS

In this section, we provide further details about our experiment setup. For the matrix game case (also mentioned in Remark 4.4), we solve the problem

$$\min_{\mathbf{x} \in \mathbb{R}^{m_1}} \max_{\mathbf{y} \in \mathbb{R}^{m_2}} \langle \boldsymbol{A}\mathbf{x}, \mathbf{y} \rangle + \delta_{\Delta^{m_1}}(\mathbf{x}) + \delta_{\Delta^{m_2}}(\mathbf{y})$$

for $\boldsymbol{A} \in \mathbb{R}^{m_1 \times m_2}$, the simplices $\Delta^{m_1}, \Delta^{m_2}$, where $\delta$ is the indicator function. We use the policeman and burglar matrix from Nemirovski (2013) with $m_1 = m_2 = 500$, where the entries are given by $A_{ij} = \boldsymbol{z}_i(1 - \exp(-\theta|i - j|))$ with $\theta = 0.8$ and $\boldsymbol{z} \sim \mathcal{N}(\mathbf{0}, I_{m_1})$. For the computation of $J_{\eta G}$, which corresponds to projection onto the simplex in this case, we use (Condat, 2016, Algorithm 1).

For the test case of Lagrangian of a quadratic program, we use the saddle function from Ouyang & Xu (2021) as follows

$$\min_{\mathbf{x} \in \mathbb{R}^{m_1}} \max_{\mathbf{y} \in \mathbb{R}^{m_2}} \frac{1}{2}\mathbf{x}^\top \boldsymbol{H}\mathbf{x} - \boldsymbol{h}^\top \mathbf{x} - \langle \boldsymbol{A}\mathbf{x} - \boldsymbol{b}, \mathbf{y} \rangle,$$

where $m_1 = m_2 = 200$, $\boldsymbol{H} = 2\boldsymbol{A}^\top \boldsymbol{A}$ and

$$\boldsymbol{A} = \frac{1}{4}\begin{bmatrix} & & & -1 & 1 \\ & & \cdots & \cdots & \\ & -1 & 1 & & \\ -1 & 1 & & & \\ 1 & & & & \end{bmatrix} \in \mathbb{R}^{m_1 \times m_2}, \quad \boldsymbol{b} = \frac{1}{4}\begin{bmatrix} 1 \\ 1 \\ \cdots \\ 1 \\ 1 \end{bmatrix} \in \mathbb{R}^{m_1}, \quad \boldsymbol{h} = \frac{1}{4}\begin{bmatrix} 0 \\ 0 \\ \cdots \\ 0 \\ 1 \end{bmatrix} \in \mathbb{R}^{m_1}.$$

Also, for Alg. 2, we directly measure the residual on its output point $\mathbf{u}_K$, without doing another approximation step as in Corollary 4.3 for simplicity. This is guaranteed by $\mathbb{E}[\|P^\eta(\mathbf{u}_k)\|] = \mathbb{E}[\|\mathbf{u}_k - J_{\eta(F+G)}(\mathbf{u}_k)\|] = \mathcal{O}(1/k)$, and $\mathbf{u}_k$ can be a good empirical approximation of $J_{\eta(F+G)}(\mathbf{u}_k)$ as the algorithm proceeds.

## E    FURTHER DISCUSSION AND PERSPECTIVES

We showed complexity guarantees for variance reduced algorithms that improve the best-known results for minimizing the residual in finite-sum monotone inclusions. Our improvements mirror those that were shown for the duality gap for finite-sum VIs in the recent literature; see e.g., Palaniappan & Bach (2016); Carmon et al. (2019); Alacaoglu & Malitsky (2022).

Our result for the cocoercive case is with a direct algorithm whereas for the Lipschitz monotone case, we have an indirect algorithm. In particular, our algorithm in the latter case works by solving randomized approximations to the resolvent, which can be seen as an inexact Halpern iteration (Diakonikolas, 2020). An important related open question is the development of direct algorithms that achieve the same complexity guarantee that we showed for the Lipschitz monotone case, which is optimal up to log factors.

It is worth pointing out that our results and this open question closely resemble the recent development of improved duality gap guarantees for finite-sum monotone VIs with variance reduction. In particular, it was the work of Palaniappan & Bach (2016) that provided the first variance reduced variational inequality algorithm which was an indirect procedure based on the Catalyst proximal point framework and forward-backward algorithm. This work already showed the benefit of variance reduction compared to deterministic algorithms for strongly monotone inclusions and monotone VIs with bounded domains by using standard reductions using regularization.

A direct algorithm for the important special case of matrix games was provided by Carmon et al. (2019). At the same time, this work also handled the general monotone VI case with an indirect approach. Other direct algorithms, given in Chavdarova et al. (2019) for the strongly monotone case and in Alacaoglu et al. (2021) for the monotone case, were simple but did not improve the complexity bounds compared to deterministic algorithms. The work of Alacaoglu & Malitsky (2022) obtained direct and single-loop algorithms with complexity improvements for general monotone VIs, nearly five years after the indirect result of Palaniappan & Bach (2016). An alternative direct approach focusing on finite-sum saddle point problems was also studied in Yazdandoost Hamedani

& Jalilzadeh (2023). Soon after the direct algorithms, matching lower bounds for the duality gap were also provided for finite-sum monotone VIs (Han et al., 2024).

In this context, our results for the monotone Lipschitz case provided the first improvement with variance reduction for residual guarantees. This can be seen as corresponding to the results of Palaniappan & Bach (2016); Carmon et al. (2019) that had indirect algorithms with complexity improvements for the duality gap for monotone VIs. What remains to be done to complete the picture for finite-sum monotone inclusions and monotone VIs is developing direct algorithms with tight complexity guarantees for the residual, similar to the process that we have seen for duality gap guarantees.

