# OpenReview forum: "Variance Reduced Halpern Iteration for Finite-Sum Monotone Inclusions"
_ICLR.cc/2024/Conference — ICLR 2024 poster_

### Official Review · Reviewer_9YKM · 2023-10-24

**Soundness:** 3 good
**Presentation:** 3 good
**Contribution:** 3 good
**Rating:** 6
**Confidence:** 3

**Summary:**

The paper studies finite-sum monotone inclusion problems. When assuming the operator is co-coercive, the paper proposes a direct single-loop algorithm combining PAGE and Halpern iteration that achieves $O(n+\sqrt{n}\epsilon^{-1})$ convergence rate. When assuming the operator is monotone and Lipschitz, the paper proposes an indirect inexact proximal point-based method with VR-FoRB as the inner-problem solver that also achieves $O(n+\sqrt{n}\epsilon^{-1})$ convergence rate. Both convergence rates are last-iterate guarantees on the operator norm.

**Strengths:**

The paper is well-written and explains every detail of the algorithms and their contributions. The discussions and comparisons to previous works clearly reflect their differences and improvements. Although the algorithms and techniques are based on previous works, the obtained last-iterate guarantees on the operator norm for finite-sum monotone inclusion problems are new in the related literature.

**Weaknesses:**

1. PAGE was originally designed for nonconvex minimization problems and SVRG/SAGA is a common choice for convex problems. Although the problem to be solved is monotone, Algorithm 1 chooses PAGE as the base algorithm. Could the authors explain why? What happens if SVRG is used?

2. I don't see any dependence and requirement on $L_F$ for both Algorithms 1 and 2. Is the assumption that $F$ is $L_F$-Lipschitz used anywhere in the analysis? Why is it required other than allowing easier comparisons with existing results? I also think the discussions on the relationship among $L_F$, $L$, and $L_Q$ should be clearly stated in the paper instead of just referring to existing works. It would be better if the paper just had one remark saying something like $L_F\leq L\leq\sqrt{n}L_F$. This allows a better understanding of the regime when the improvements happen.

3. I think a discussion about how and where the additional logarithmic factors in the convergence results of both algorithms come from would be great. I assume they come from different sources and thus require different techniques and efforts to get rid of (if possible).

4. I still have questions on how to check the stopping criteria for the inner-problems in Algorithm 2. Is $E\Vert e_k\Vert^2$ something computable since it requires the exact solution $J_{\eta(F+G)}(u_k)$?

5. What are the examples of operators that are non-monotone but co-monotone other than $F(x)=-x$?

6. In Figure 1(b), EAG and Algorithm 2 tend to have lots of oscillations but Algorithm 1 does not. Some explanations and discussions on this might be good.

7. (Minor) Although the results in the paper are new, I am not so surprised by their algorithms and technical analyses given the rich literature on the related topics. It seems standard that the deterministic version of some problem is first studied and then variance reduction can be applied to improve the rates for its finite-sum extension. The results in this paper are thus expected by combining existing knowledge, especially given its infinite-sum version in (Cai, 2022a). Another small concern is about the practical use of variance reduction methods. Even though they offer plenty of benefits in the theoretical analysis, there are works that report these methods do not have good performances in practice as the theory suggests, e.g., [arXiv:1812.04529]. There is even no official support for variance reduction methods in the popularly used machine learning packages like PyTorch and TensorFlow despite rich literature showing their theoretical advantages on both convex and nonconvex problems.

**Questions:**

See Weaknesses.

---

> ### Author Response · Authors · 2023-11-15
> **Rebuttal by Authors (1/4)**
>
> **Thank you for the feedback and for the questions you raised. We hope that the following clarifications address your question and that you will consider reevaluating our work. Please let us know if there is any additional information you would find helpful.**
>
> ---
>
> #### **Weaknesses**
>
> > PAGE was originally designed for nonconvex minimization problems and SVRG/SAGA is a common choice for convex problems. Although the problem to be solved is monotone, Algorithm 1 chooses PAGE as the base algorithm. Could the authors explain why? What happens if SVRG is used?
>
> We chose PAGE due to its simplicity and compatibility with our analysis. Our results already improve upon the existing work on using variance reduction for cocoercive operators (Davis (2023) and Morin & Giselsson (2022)), and are likely unimprovable (in fact, the results match the lower bound up to a log factor in the more general monotone case and we do not expect there to be a gap between the monotone and the cocoercive case, based on the existing complexity results in other related settings). This is the main reason we did not try SVRG/SAGA – it is unlikely there is any benefit to using them, though it is possible that similar results could be obtained by using alternative techniques, which is beyond the scope of our work.
>
> We emphasize that although SVRG/SAGA may be a common choice for convex problems, this does not mean they are the only choice: the recursive variance reduction method SARAH (equivalent to PAGE) was proposed for the convex/strongly convex settings. Moreover, as discussed in the last paragraph of Section 3, an application of our results (with the PAGE estimator) to finite-sum convex minimization problems leads to the best-known gradient norm guarantees for direct methods, which constitutes another (albeit minor) contribution of our paper.
>
> > I don't see any dependence and requirement on $L_F$ for both Algorithms 1 and 2.
>
> $L_F$ is mainly used for comparison with deterministic algorithms, since they are the state-of-the-art for complexity of operator residual prior to our work and their complexity depends on $L_F$. As mentioned in the first paragraph of Section 4, we omit the subscript and denote $L = L_Q$ for brevity, and we added a more explicit discussion about the relationship of $L$ and $L_F$ in Appendix C.4. Briefly, the reviewer is completely right that $L_F \leq L$ by triangle inequality. However, it might be the case that component Lipschitzness is easier to estimate, especially in the large-scale setting, and $L$ provides a natural upper bound on $L_F$.
>
> > I also think the discussions on the relationship among $L_F$, $L$ and $L_Q$ should be clearly stated in the paper instead of just referring to existing works. It would be better if the paper just had one remark saying something like $L_F \leq L \leq \sqrt{n}L_F$. This allows a better understanding of the regime when the improvements happen.
>
> We added details for the derivation of $L_F$ and $L_Q$ for the important special case of bilinear problems, in Appendix C.4. Does our addition seem satisfactory to the reviewer or do they want us to include more discussions or details?
>
> It is of course possible, as also mentioned in the work of Alacaoglu & Malitsky (2022) that the relationship between $L_F$ and $L_Q$ might prevent improvements for variance reduction (unfortunately, a generic bound such as $L\leq \sqrt{n} L_F$ does not hold when we step away from convex minimization). See for example, the discussions in Section 2.1 and Appendix D in Alacaoglu & Malitsky (2022) for min-max problems. Let us mention that the difference in Lipschitz constants in deterministic and variance reduced algorithms is inherently prevalent for all variance reduction algorithms to our knowledge, see for example the seminal paper on proximal SVRG [Lin, Zhang 2014, discussion right after eq. 9].
>
> [Lin, Zhang, 2014] Xiao, Lin, and Tong Zhang. "A proximal stochastic gradient method with progressive variance reduction." SIAM Journal on Optimization 24.4 (2014): 2057-2075, reference to arXiv:1403.4699.

---

> ### Author Response · Authors · 2023-11-15
> **Rebuttal by Authors (2/4)**
>
> > I think a discussion about how and where the additional logarithmic factors in the convergence results of both algorithms come from would be great. I assume they come from different sources and thus require different techniques and efforts to get rid of (if possible).
>
> This is a good point! We did not focus on removing log factors although we agree it would be an interesting question. We repeat that our results for the residual are the first optimal-up-to-log complexities for finite-sum convex-concave problems, monotone VIs, and monotone inclusions.
>
> For Alg. 1, we chose the probability of full operator evaluation $p_k = \frac{4}{k + 5}$ for the first $\sqrt{n}$ iterations, which contributes $\mathcal{O}(n\log(n))$ oracle queries to the final complexity that hides $\log(n)$ terms. For Alg. 2, each $k$-th inner loop requires $\mathcal{O}((n + \sqrt{n}(\eta L + 1))\log(k + 2))$ oracle queries, which leads to the logarithmic term $\mathcal{O}(\log(K)) = \mathcal{O}(\log(1/\epsilon))$ in the final complexity that we hide. These can be precisely seen in the proofs of Theorem 3.2 and Theorem 4.2, respectively. We did not include logarithmic factors in our theorem statements for simplicity of presentation. It is also because shaving off logarithmic terms in the complexity is beyond the scope of our work, since we already provide the first algorithms with complexity that is optimal (up to a log) in terms of the operator residual for the class of convex-concave problems and more generally, monotone VIs and monotone inclusions.
>
> > I still have questions on how to check the stopping criteria for the inner-problems in Algorithm 2. Is $\mathbb{E} \lVert e_k \rVert^2$ something computable since it requires the exact solution $J_{\eta(F + G)}(u_k)$?
>
> This is a good question and you are completely right that one cannot check the bound on $\mathbb{E}\lVert e_k\rVert^2$: this is the exact difficulty that our analysis overcomes. The precise insight we use for termination: We set an explicit number of iterations (that only depends on $L$, $n$ and $k$) where we **guarantee** that the condition on the error is satisfied (see the first paragraph in the proof of Theorem 4.2) hence the the algorithm does not require checking the error. One cannot check the expectation of the error in the stochastic case, which is the main difference in going from the deterministic to the stochastic case.
>
> > What are the examples of operators that are non-monotone but co-monotone other than $F(x) = -x$?
>
> First, we would like to emphasize that co-monotone operators are just a quick corollary of our main result, for completeness, and this corollary is definitely not a main result of our paper. Our main results are for monotone operators (convex-concave min-max problems as a special case) where we obtain the optimal complexity results for operator residual for the first time, with improvement up to a factor of $\sqrt{n}$ compared to the existing state-of-the-art due to deterministic algorithms.
>
> Classical examples include the inverse of Lipschitz continuous operators (see e.g., Bauschke et al. (2021)), and the gradient operators of the weakly-convex-weakly-concave objectives (see e.g., Example 1 in Lee & Kim, (2021) and Example 3.1 and Appendix B.1 in Pethick et al. (2023)).
>
> [Pethick et al. (2023)] Thomas Pethick, Wanyun Xie, Volkan Cevher. Stable Nonconvex-Nonconcave Training via Linear Interpolation. In Proc. NeurIPS’23, 2023.
>
> > In Figure 1(b), EAG and Algorithm 2 tend to have lots of oscillations but Algorithm 1 does not. Some explanations and discussions on this might be good.
>
> We will add a more detailed discussion in a later revision. From our perspective, the quadratic program for Figure 1(b) is a worst-case problem and will cause oscillations for EG-type algorithms (see e.g., Park & Ryu (2022); Cai & Zheng (2023)). EAG can be seen as EG with anchoring at the initial point, while Alg. 2 uses VR-EG for the subsolver. Alg. 1 has less oscillations because it is a one-step Halpern that is not an EG-type algorithm, and the incorporation of PAGE may help reduce the oscillations as well.
>
> [Cai & Zheng (2023)] Yang Cai, Weiqiang Zheng. “Doubly Optimal No-Regret Learning in Monotone Games”. In Proc. ICML 2023, 2023.

---

> ### Author Response · Authors · 2023-11-15
> **Rebuttal by Authors (3/4)**
>
> > Although the results in the paper are new, I am not so surprised by their algorithms and technical analyses given the rich literature on the related topics. It seems standard that the deterministic version of some problem is first studied and then variance reduction can be applied to improve the rates for its finite-sum extension. The results in this paper are thus expected by combining existing knowledge, especially given its infinite-sum version in (Cai, 2022a).
>
> - First, regarding the “extension from deterministic result to variance reduction”: We respectfully disagree with this point of view. As mentioned in our discussion in App. E: it took more than 5 years to settle the optimal complexity of variance reduced min-max algorithms where the duality gap is the optimality measure. It was definitely not a “direct extension” of deterministic algorithms and required the combination of different ideas in designing stochastic algorithms, in addition to extensive seminal prior works improving oracle complexity using variance reduction for finite-sum minimization.
> - Another pointer to why combination of a deterministic algorithm and variance reduction is not straightforward: papers Chavdarova et al. (2021), Carmon et al. (2019) and Alacaoglu, Malitsky (2022) all combined variance reduction with the *same* algorithm: the extragradient. However, these algorithms are not only different, their properties are also wildly different; Chavdarova et al. (2021)’s algorithm only works with strongly monotone operators, Carmon et al. (2019)’s algorithm works only with bounded domains or for matrix games, and Alacaoglu, Malitsky (2022)’s algorithm can work with the general setting of unbounded convex-concave problems or monotone VIs. This goes to show how subtle and non-straightforward it is to combine variance reduction with primal-dual algorithms. One needs the **right** combination to get the desirable convergence guarantees.
> - Moreover, the techniques we use are quite distinct from the infinite-sum version (Cai et al., 2022a), even in the cocoercive case. Cai et al., 2022a handles the variance reduction by separating the variance term, and their analysis heavily depends on using large batches (depending on $\varepsilon$) to ensure $\mathbb{E}[\lVert F(u_k) - \tilde{F}(u_k)\rVert^2] = \mathcal{O}(\epsilon^2/k)$. It is obvious that one cannot obtain such a variance bound in the finite-sum setting. Instead, we include the variance in our potential function and show that it decreases without batch sizes depending on $\varepsilon$ but only depending on $\sqrt{n}$, by using the structure of finite-sum and taking different $p_k$ (see Alg.1 and Lemma B.1 in Appendix B) than the original PAGE estimator. In the monotone case, our Algorithm 2 is completely different from the one in Cai et al., 2022a, and our result further handles the constrained setting. Additionally, while the complexity results we obtain are near-optimal (optimal up to a log factor) in the finite-sum setting we consider, the result of Cai et al., 2022a for the infinite-sum setting is *suboptimal* (their oracle complexity is $1/\epsilon^3$, whereas the optimal oracle complexity for the infinite sum setting is $1/\epsilon^2$). It would be highly unusual that transferring results from one domain (infinite-sum) to another (finite-sum) would suddenly change the results from highly suboptimal to (near-)optimal.
> - Last but not least, we believe that the community has the fundamental tradition of appreciating the contributions of those works that provide improved (near-optimal) complexity results. Our paper provides the first improvements with variance reduction for operator residual (computable unlike previous results) guarantees for min-max problems. These are near-optimal oracle complexity results for general finite-sum monotone inclusions, and several implications (such as the gradient norm guarantee in finite-sum minimization and the duality gap guarantee in finite-sum convex-concave min-max problems) are also discussed in the paper to show the tightness of our results. Our results also lend themselves to a clean argument resulting in the **first high probability results** with variance reduction for min-max problems.

---

> > ### Author Response · Authors · 2023-11-15
> > **Rebuttal by Authors (4/4)**
> >
> > > Another small concern is about the practical use of variance reduction methods. Even though they offer plenty of benefits in the theoretical analysis, there are works that report these methods do not have good performances in practice as the theory suggests, e.g., [arXiv:1812.04529]. There is even no official support for variance reduction methods in the popularly used machine learning packages like PyTorch and TensorFlow despite rich literature showing their theoretical advantages on both convex and nonconvex problems.
> >
> > We would like to reiterate that the aim of our paper is theoretical, to show the benefit of variance reduction for improving complexity for min-max problems. The use of variance reduction for deep learning is out of the scope of our paper and we agree with the reviewer that it is an interesting topic of inquiry.
> >
> > On the other hand, we note that the investigation of variance reduction for deep learning is still an ongoing area of research, see for example the recent SVRG work (Yin et al., (2023)). Even in the paper [arXiv:1812.04529] the reviewer mentioned, the authors still argued in the conclusion that “..., however we don’t believe that they rule out the use of stochastic variance reduction on deep learning problems. Rather, they suggest avenues for further research. For instance, SVR can be applied adaptively; or on a meta level to learning rates; or scaling matrices; and can potentially be combined with methods like Adagrad and ADAM to yield hybrid methods”.
> >
> > To support our point that this practical aspect is an active research topic: the reviewer can also see the reaction of the first author of [arXiv: 1812.04529] to the new development Yin et al., (2023) on using SVRG for deep learning: https://twitter.com/aaron_defazio/status/1713959068409204785
> >
> > [Yin et al., (2023)] Yin, Yida, et al. "A Coefficient Makes SVRG Effective." arXiv preprint arXiv:2311.05589 (2023).
> >
> > (Please see our paper for other citations.)

---

> > > ### Comment · Reviewer_9YKM · 2023-11-22
> > >
> > > Thanks for your detailed explanations. I appreciate for the authors' effort and this healthy discussions. I raise my score to 6.

---

> > > > ### Author Response · Authors · 2023-11-22
> > > > **Follow-up on the reviewer's feedback**
> > > >
> > > > We really appreciate the follow-up and the increasing support by the reviewer. We will also be very happy to provide further clarifications for any remaining concerns of the reviewer that keeps them “borderline” about our paper.

---

### Official Review · Reviewer_wjm9 · 2023-11-10

**Soundness:** 3 good
**Presentation:** 2 fair
**Contribution:** 3 good
**Rating:** 6
**Confidence:** 3

**Summary:**

The paper addresses game-theoretic equilibrium problems pertinent to adversarial robustness and multi-agent settings, by studying finite-sum monotone inclusion problems. The authors propose variants of the Halpern iteration with variance reduction, yielding improved complexity guarantees for problems where component operators are cocoercive or Lipschitz continuous and monotone on average.

**Strengths:**

In the paper's context, the authors claim an oracle complexity of \\( \mathcal{O}(n + \sqrt{n}L/\varepsilon) \\) under their studied conditions, providing a considerable (theoretical) improvement over prior methodologies.

**Weaknesses:**

*Due to confusing presentation, there is doubt on whether the improvements stem from an innovative analysis approach, or is predominantly artifacts of the specific assumptions employed in their  (restricted) settings.*

**Assumptions and Implications:**
- Considering the decomposition, $\mathbb{E}_{q \sim Q} \left[ \| F_q(u) - F_q(v) \|^2 \right] = \text{Var}_{q \sim Q} \left[ \| F_q(u) - F_q(v) \| \right] + \left( \mathbb{E}_{q \sim Q} \left[ \| F_q(u) - F_q(v) \| \right] \right)^2$, assumptions (particularly 2 and 3) concerning bounded second moment warrant a deeper discussion on their necessity -- *particularly since variance reduction is typically sought in scenarios with large or unbounded variance.* Could the authors provide a discussion of the implications of these assumptions, and whether these could be relaxed without compromising the results?
- The paper suggests that the bounded second moment assumption holds as standard for analyzing variance-reduced algorithms, referencing works such as (Palaniappan & Bach (2016)); (Carmon et al. (2019)); and (Alacaoglu & Malitsky (2022)). While all referenced works are implied to operate under similar conditions, however, it is unspecified whether those works inherently assume boundedness, or *derive it as a result of variance reduction*. It's unclear given the different problem settings, and if they indeed support such an assumption as a given, or if they use variance reduction techniques to bound the second moment, which are significant distinctions.
- Therefore, claiming that such assumptions are standard in the context of variance-reduced algorithms might be misleading. A deeper analysis of the references in question, and how they relate to the specific assumptions made in this paper would be beneficial.

**Presentation and Exposition:**
- The paper's presentation of the problem setting, such as "graph of maximal monotone operators" in Section 2 is unclear. More precise definitions, without obscured notations, would greatly aid in comprehension.
- The discussion post Lemma 3.1 could benefit from elaboration on what is meant by "going beyond" the deterministic setting. Clarification on the nature and implications of the "more complicated induction-argument" that is avoided may also add to the reader's understanding.
- A structured overview would facilitate a more transparent evaluation of the work's context and contributions. To aid in the accurate evaluation of the paper's contributions, a side-by-side comparison with the assumptions, techniques, and results of existing work would be invaluable. A table format could be the most effective way to present this information, providing a quick reference to understand the advancements made. Could the authors provide a table comparing this work to the literature regarding key assumptions and results, such as variance bounds, Lipschitz conditions, sample complexity, and optimality measures?

**Numerical Experiments:**
- The numerical experiments section requires a clearer articulation of its goals and outcomes. The interpretation of the numerical experiments, particularly in Section 5, Figure 1, is not immediately apparent. Could the authors elucidate the objective of these experiments and how they substantiate the paper's claims?

**Clarification on Algorithms:**
- Considering the significant overlap with prior works on Algorithms 1, 2 and 3, would it be possible for the authors to detail the specific analysis differences and their implications? A tabulated summary including convergence rates, settings (deterministic versus stochastic), and the role of approximation errors would be highly informative. This should include the pivotal details, ensuring that all terms are precisely defined. It would also clarify the stochastic elements in the context of the algorithms proposed and how these factors are managed or analyzed differently from the literature.

**Originality:**
- The adaptation of Algorithms 1, 2, and 3 from existing literature and the application of martingale analysis proofs should be accompanied by a clear indication of the novel contributions of this work. Specifically, what is the new insight and proof technique that lead to the reported improvement in sample complexity? Currently, it seems to be mere artifact of the specific assumptions for this problem setting.

**Questions:**

Kindly address the points raised above, which are briefly summarized here:
1. **Clarity of Presentation:** The exposition is particularly convoluted and indirect. It could be re-structured for better comprehension.

2. **Comparison with Existing Literature:** The paper should offer a clearer and more direct comparison with existing works. Such a comparison would be more insightful if it included a discussion on how the assumptions differ from those in the literature and the impact of these differences on the results. Additionally, the citations of relevant literature regarding expected Lipschitz continuity and variance reduction may require a review to ensure accurate representation. *The paper should address whether those cited works indeed operate under analogous assumptions or if there are misalignments which may affect the interpretation of the current work's contributions.*

3. **Assumptions and Implications:** The paper would benefit from a critical examination of its assumptions, particularly Assumptions 2 and 3. Since the paper aims to address variance reduction, *assuming bounded second moments could be contradictory.* It's crucial to investigate whether they overly constrain the scenarios the variance reduction aims to address.

4. **Proof Methodology:** Besides the strict assumptions, it is not immediately clear what novel proof techniques or insights contribute to the improved sample complexity bound. A more explicit articulation of these novel aspects would help delineate the paper's contributions from existing work.

*Overall, I encourage the authors to meticulously delineate the parallels and distinctions, between the analysis, results, & proofs presented in this paper and those in prior studies. Clarifying this may enhance the perceived value of the work.*

$\textbf{[Update:]}$ After a thorough reflection on the merits and limitations, I have decided to bump up my score to a **6**. Still, I encourage the authors to refine their manuscript, including the nuanced discussions on $L$, spectral-norm & other dependencies, and the tail-bounds. (Ideally, mention important caveats in the early sections.)

---

> ### Author Response · Authors · 2023-11-15
> **Rebuttal by Authors (1/3)**
>
> **Thanks for the questions you raised. We hope that the clarifications provided below resolve your questions and that you would consider reevaluating our work. Please let us know if there is any additional information you would find helpful.**
>
> ---
>
> #### **Strengths**
>
> > In the paper's context, the authors claim an oracle complexity of $O(\sqrt{nL/\epsilon})$ under their studied conditions, providing a considerable (theoretical) improvement over prior methodologies.
>
> We point out that the reviewer has misinterpreted our complexity result $\widetilde{O}(n + \sqrt{n}L/\epsilon)$, which is the best known (and also optimal up to log factors) for the problem settings considered in our paper. It is also the first near optimal result for this general problem setting.
>
> ---
>
> #### **Weaknesses**
>
> ##### **Assumptions**:
> The main argument of the reviewer is that our assumptions are different from the related work. This is **not** true and we refer below to the precise locations in the previous work where the exact same assumption is made. This concludes that our improvements are **not** due to stronger assumptions but are due to our new approach that operates under the standard assumptions.
>
> >  assumptions (particularly 2 and 3) concerning bounded second moment warrant a deeper discussion on their necessity; The paper suggests that the bounded second moment assumption holds as standard for analyzing variance-reduced algorithms
>
> **Both of the statements of the reviewer, quoted above, are incorrect.** Our assumption does not require a bounded second moment. Here is an obvious counterexample: Take $F(x) = Ax$ with $A \in \mathbb{R}^{n \times n}$ and $F_i(x) = nA_i x_i$, where $A_i$ is $i$-th column of the matrix $A$ and $x_i$ is the $i$-th element of the vector $x$, and take the oracle distribution $Q$ to be the uniform distribution over $[n]$. Then, our Assumption 2 holds with $L=\sqrt{n}\max_i \lVert A_i \rVert$ whereas the second moment $\mathbb{E}_i[||F_i(x) - F(x)||^2]$ is obviously unbounded for $x \in \mathbb{R}^n$. Could the reviewer further clarify why they think our assumption implies a bounded second moment? For the second statement, please see the next point.
>
> > A deeper analysis of the references in question, and how they relate to the specific assumptions made in this paper would be beneficial.
>
> **In summary, our assumptions are exactly the same with the previous variance reduction methods for min-max problems.** We here provide pointers to the exact same assumption used in prior works.
>
> - *Assumption 2: Lipschitzness in expectation* (holds trivially when all component operators are Lipschitz).
> 1. Assumption 1(iv) in Alacaoglu and Malisky (2022),
> 2. (C) in Section 2 of Palaniappan & Bach (2016) where each operator is assumed to be $L_i$-Lipschitz which satisfies our assumption with $L=\mathbb{E}_{i\sim Q} L_i$,
> 3. Sec 5.4 in Carmon et al. (2019) where they discuss convex-concave problems, they assume that each function has $L_k$ Lipschitz gradients, which maps to our assumption the same way as Palaniappan & Bach (2016),
> 4. Def 2.2 in Han et al., (2021) that studied lower bounds for convex-concave problems.
> - *Assumption 3: Cocoercivity in average* (holds trivially when all component operators are cocoercive).
> 1. Assumption 3.1(component cocoercivity) and 3.2 in Morin & Giselsson (2022), and Sec 3.2 in Davis (2023) directly imply our  Assumption 3.
> 2. (1.3) in Davis (2023) and Assumption 3.3 in Loizou et al. (2021) can be seen as relaxing our Assumption 3 to hold only with respect to the solution point, However, Loizou et al. (2021) additionally assumed bounded variance (and quasi strong monotonicity) at the solution point which we do not assume in our paper. On the other hand, no non-asymptotic rate is proved under only (1.3) in Davis (2023), and they only show non-asymptotic rate with additional quasi-strong monotoncity assumption which is also not required by our paper.
>
> Hence we can conclude that the improvements are **not** due to extra assumptions on the template but to our analysis. The exact same assumptions, as we pointed out in the references above, are made in the related work.
>
> - Is the reviewer convinced that our assumptions are exactly the same as Palaniappan & Bach (2016), Carmon et al. (2019), Alacaoglu and Malisky (2022)? If not, can they justify how they think our assumptions differ?
>
> [Han et al., (2021)] Yuze Han, Guangzeng Xie, Zhihua Zhang. “Lower Complexity Bounds of Finite-Sum Optimization Problems: The Results and Construction.” arXiv preprint arXiv:2103.08280, 2021.

---

> ### Author Response · Authors · 2023-11-15
> **Rebuttal by Authors (2/3)**
>
> ##### **Presentation and Exposition**:
>
> > The paper's presentation of the problem setting, such as "graph of maximal monotone operators" in Section 2 is unclear. More precise definitions, without obscured notations, would greatly aid in comprehension.
>
> We use the standard, fundamental and not an obscure definition of the graph of an operator in mathematics. For $T: X \rightrightarrows X$, the graph of $T$ is defined by $gra T = \\{(\bar u, u) \in X \times X | \bar u \in T(u)\\}$. $T$ is maximal monotone if there is no monotone operator that properly contains it. In other words, if $T$ is not maximal monotone, then there exists $(\bar v, v)$ with $\bar v \notin T(v)$ such that $<u - v, \bar u - \bar v> \geq 0$ for all $(\bar u, u) \in gra T$. We have added the definition of graph in Section 2 in our revision, highlighted in red.
>
> > The discussion post Lemma 3.1 could benefit from elaboration on what is meant by "going beyond" the deterministic setting.
>
> We didn’t elaborate in the paper due to space constraints. Briefly, as is standard in stochastic optimization literature, one need to handle the variance $\mathbb{E}\lVert F(u_k) - \tilde{F}(u_k)\rVert^2$ introduced by the operator estimator $\tilde{F}$. Vanilla extension of the deterministic algorithm in Kovalev & Gasnikov (2022) with a mini-batch estimator will not lead to improved complexity. That is why we include the additional variance term $c_k\mathbb{E}\lVert F(u_k) - \tilde{F}(u_k)\rVert^2$ in our potential function and incorporate recursive variance reduction to still obtain the contraction in expectation. We will add it in later revisions.
>
> > Clarification on the nature and implications of the "more complicated induction-argument" that is avoided may also add to the reader's understanding.
>
> Unlike our Lemma 3.1 that shows the contraction of the potential function in expectation, Cai et al. (2022a) chose the batch size dependent on the distance between successive iterates to ensure $\mathbb{E}\lVert F(u_k) - \tilde{F}(u_k)\rVert^2 = O(\epsilon^2/k)$ in expectation, and the change of the potential function therein requires bounding $\lVert F(u_k)\rVert^2$ and $\lVert F(u_k) - \tilde{F}(u_k)\rVert^2$ at the same time, so a more complicated induction-based argument is needed there. We avoid such arguments due to the newly designed potential function and the properly chosen probability of full operator evaluation $p_k$. We will add a more detailed discussion in a later revision. More importantly, our analysis helps us get better complexity in the *finite-sum* case that we study.
>
> > A table format could be the most effective way to present this information, providing a quick reference to understand the advancements made.
>
> We have added Table 1 for the comparison of our results with the state of the art in Appendix A in the revision. For reviewer’s convenience, we also provided the table in markdown in our **General Response** above.
>
> ---
>
> ##### **Numerical Experiments**:
>
> We have explicitly stated our goal in the first paragraph of Section 5, and have explained the numerical results in the last paragraph of Section 5. Could the reviewer please let us know which part needs to be elaborated? We are happy to revise.
>
> ---
>
> ##### **Clarification on Algorithms**:
>
> > Considering the significant overlap with prior works on Algorithms 1, 2 and 3, would it be possible for the authors to detail the specific analysis differences and their implications?
>
> We have precisely discussed the differences in the paragraphs after Algorithms 1, 2 and 3 in our paper and also repeated these points in our **General Response** above. The main contribution of our paper is the new analysis and the resulting first-of-its-kind optimal complexities in our setting.
>
> ---
>
> ##### **Originality**:
>
> > Specifically, what is the new insight and proof technique that lead to the reported improvement in sample complexity? Currently, it seems to be mere artifact of the specific assumptions for this problem setting.
>
> This statement is not true. Please see our response for **Assumptions** above, i.e., the improvements are **not** a mere artifact of assumptions: We use the exactly same standard assumptions as prior works on variance reduced min-max algorithms.
>
> At the same time, we have described our proof techniques and discussed the differences between our algorithms/analysis and the related literature in Sections 3 and 4. For your convenience, we repeated these points in our **General Response** above. Could the reviewer please let us know which part of our techniques requires more elaboration? We are happy to answer.

---

> > ### Author Response · Authors · 2023-11-15
> > **Rebuttal by Authors (3/3)**
> >
> > #### **Questions**
> >
> > Please refer to our responses to the weaknesses part. Here we include a further summary to address the reviewer’s questions.
> >
> > > The paper should address whether those cited works indeed operate under analogous assumptions or if there are misalignments which may affect the interpretation of the current work's contributions.
> >
> > We would like to re-emphasize that we are using the same standard assumptions as prior works on variance reduction. Please see our response above for precise references to existing work and the table in our paper which is also provided in our **General Response**.
> >
> > > The paper would benefit from a critical examination of its assumptions, particularly Assumptions 2 and 3. Since the paper aims to address variance reduction, assuming bounded second moments could be contradictory.
> >
> > We repeat, this statement is not correct: we do not assume bounded second moments anywhere in the paper. As explained above, our standard assumptions do **not** imply bounded second moments. Also, since our assumptions are quite standard and prevalently used (as justified above and **General Response** with references to existing work where the same assumptions are done), we do not believe further critical examination is needed. Could the reviewer clarify how they arrive at their conclusion? We would be happy to discuss.
> >
> > > Besides the strict assumptions, it is not immediately clear what novel proof techniques or insights contribute to the improved sample complexity bound.
> >
> > First, we completely disagree with the comment *``the strict assumptions’’*, as our assumptions are all standard. Please see our earlier comments and **General Response** above. Moreover, we believe that we have already discussed our proof techniques and insights in Sections 3 and 4 (see also the **General Response** where we repeated these). If there is further information that would be useful, please let us know.
> >
> > (Please see our paper for other citations.)

---

> ### Comment · Reviewer_wjm9 · 2023-11-22
> **Feedback on author rebuttal**
>
> Thank you for the clarifications and comparative analysis with existing literature, which has enhanced the presentation of the paper. Accordingly, I have revised my evaluation score to 5. However, I must express that despite the improvement, I maintain disagreements that preclude a higher score.
>
> For instance, in their $\textbf{General Response}$, the authors made the following claim (*verbatim*):
>
> *“we show the first optimal complexity results that hold with **high probability** (depending logarithmically on the confidence level) for both the gap and operator residual measures”*.  $\Rightarrow$
>
> The statement is incorrect, as the results in the paper are limited to bounding **only** the **expected values**. Furthermore, the sources of randomness are not clearly defined. Fundamentally, this approach is less convincing than high-probability bounds, which are critical for understanding the full distribution of outcomes, especially the tail regions.
>
> In stochastic problems with a large variance, it's vital to analyze the likelihood and magnitude of deviations beyond the expected value, particularly when the underlying distributions are not tightly concentrated around their mean – a typical scenario in modern machine learning applications.
>
> Additionally, the robustness of those deviations in scenarios with arbitrarily large \\( n \\) or \\( L \\) remains unaddressed, which adds to my skepticism of the other comment in authors' rebuttal (*again, verbatim*):
>
> *“Our assumption does not require a bounded second moment. Here is an obvious counterexample: Take \\( F(x) = Ax \\) with \\( A \in R^{n \times n} \\) and \\( F_i(x) = nA_ix_i \\), where \\( A_i \\) is the i-th column of the matrix \\( A \\) and \\( x_i \\) is the i-th element of the vector \\( x \\), and take the oracle distribution \\( Q \\) to be the uniform distribution over \\([n]\\). Then, our Assumption 2 holds with \\( L = \sqrt{n} \ max_i \|| A_i \|| \\)”*. $\Rightarrow$
>
> That example is somewhat misplaced, because the authors embed crucial aspects of the problem size, represented by the \\( \sqrt{n} \\) factor, within the Lipschitz constant \\( L \\).
>
> In my experience, the Lipschitz factor should be constant, i.e., independent of \\( n \\). Allowing \\( L \\) to depend on multiples of \\( \sqrt{n} \\) would undermine the main results of the paper (i.e., tightness around sample complexity bound of \\( \mathcal{O}(n + \sqrt{n}L/\varepsilon)  \\)). Admittedly, we often observe similar issues in other ML papers. In general, manuscripts seem to **blur** the distinction between multiple Lipschitz constants and other problem sizes. While I cannot definitively assess the specific literatures cited in this paper, mishandling constant factors attributes to a growing disparity between theoretical ML and practical applications, especially on large datasets.
>
> Overall, the algorithms presented here have noteable overlaps with prior works, and relies on standard proof techniques already established in ML theory. Given these factors on its own merits, I believe the paper is not ready for publication and inclined to vote for a borderline **reject**. Nonetheless, I encourage the authors to continue refining their manuscript for future submissions.

---

> ### Author Response · Authors · 2023-11-22
> **Follow-up on the reviewer's feedback (1/3)**
>
> **We address the reviewer's further concerns below.**
>
> ---
> ### High-probability results
> > The statement is inaccurate, as the results in the paper are limited to bounding only the expected values.
>
> We first wish to emphasize that our result is **the first-of-its-kind even for expectation guarantees**. We are the first to provide variance-reduced guarantees in the residual for finite-sum min-max problems and more generally for monotone inclusions. Previous works focused on the duality gap which is not a computable quantity in general. Residual is computable and we also show up to $\sqrt{n}$ improvements compared to the previous SOTA for residual guarantees. In particular, please see the newly added Appendix C.4 where we explicitly derive Lipschitz constants for our method and deterministic methods (that were SOTA before our works) for showing the improvement in residual guarantees. Below, we also have explanations about this comparison in the context of our counterexample.
>
> We also believe that **the reviewer might have missed the paragraph after Remark 4.4 in our paper where we discuss high-probability results and make this claim precise**. In particular, our result in expectation implies by Markov’s inequality that $\mathrm{Res_{F+G}(x)} \leq \varepsilon$ with a constant probability, take $1/2$. Then running the algorithm $t$ times gives the failure probability as $1/(2^t)$. Then the most crucial point here is that since residual is **computable**, we can select the point with the smallest residual. For getting a result with failure probability $\delta$, we need to only run the algorithm $t=\log(1/\delta)$ number of times adding only a logarithmic term to the complexity bound.
>
> As we also cited in our paragraph after Remark 4.4, this is a rather standard reduction. It is worth noting that **this cannot be done with previous results for variance reduced min-max algorithms** simply because the guarantees in these works are on the duality gap which is **not computable**. Hence, such high probability reduction cannot be done there. This is showing that even for the duality gap, we can provide the first high probability result. As the reviewer believes in the value of high probability results, we hope that our simple corollary can help them re-evaluate our work.
>
> Is the reviewer convinced about the high probability corollary of our paper or do they require any further clarifications?

---

> ### Author Response · Authors · 2023-11-22
> **Follow-up on the reviewer's feedback (2/3)**
>
> ### How $L$ depends on $n$
>
> > That example is somewhat misplaced, because the authors embed crucial aspects of the problem size $n$, represented by the factor, within the Lipschitz constant $L$.
>
> We did not entirely understand what the reviewer meant by “misplaced”. We clearly show that the statement “the paper requires bounded second moment” is incorrect: we do **not** require a bounded second moment.
>
> We sincerely agree with the reviewer that the dependence on $n$ is important. This is why, in Appendix C.4, for special cases of matrix games and linearly constrained optimization, we derive explicitly the Lipschitz constants by the norms of the included matrix. Hence, we show in this case the complexities **with no hidden dependencies**. Depending on the relationship between the spectral and the Frobenius norm, our results can provide improvements up to a factor of $\sqrt{n}$.
>
> Reviewer is right that $L$ is now dependent on $n$ but we **do** take this into account **unlike** the papers the reviewer refers to. Our oracle complexity (as defined in the paper; here it corresponds to the number of times we access or multiply a matrix column by another vector) is
> $$
> O\left( n+\frac{n\max_i\lVert A_i\rVert}{\varepsilon} \right),
> $$
> whereas the arithmetic complexity of deterministic methods is
> $$
> O\left(\frac{n\lVert A\rVert}{\varepsilon} \right).
> $$
>
> Now, it is possible that the spectral norm is of the order $\sqrt{n}$ whereas the maximum row norm is of order $1$. Take for example a rank-1 matrix whose row norms are all order $1$, then Frobenius and spectral norms are both order-$\sqrt{n}$ and $\max_i\lVert A_i\rVert =O(1)$ leading to the claimed $\sqrt{n}$ factor improvement that we have. Of course, this is one extreme case and the other extreme is when both complexities can be comparable, this is why we refer to our improvement as **up to a factor of $\sqrt{n}$**. Note that this is the same case across all the variance reduced methods in this space, in agreement with previous works (Balamurugan & Bach, (2016), Carmon et al., (2019), Alacaoglu, Malitsky, (2022)). Our novelty is that we can show the improvements for the **computable** residual.
>
> We emphasize that this does **not undermine** the tightness of our result because the $L$ that we use in Assumption 2 is exactly the same $L$ that the lower bounds are derived for. Please see Def 2.2 and Table 1 in Han et al., (2021). Note that the dominant term for the $\mu_x = \mu_y=0$ in Table 1 of Han et al., (2021) has the same dependence on $n, L, \varepsilon$ as our paper. Of course as we provided references, the same $L$ is used in all the previous literature for variance reduction for min-max.
>
> In summary: our results have the **same $L$ dependence** as the existing best results for variance reduction for duality gap guarantees and the lower bounds (in terms of the dominant term).
>
> Is the reviewer convinced about fairness of the complexity comparison we are doing or do they require any further clarifications?
>
> [Han et al., (2021)] Yuze Han, Guangzeng Xie, Zhihua Zhang. “Lower Complexity Bounds of Finite-Sum Optimization Problems: The Results and Construction.” arXiv preprint arXiv:2103.08280, 2021.
>
> (Please see our paper for other citations.)

---

> > ### Author Response · Authors · 2023-11-22
> > **Follow-up on the reviewer's feedback (3/3)**
> >
> > ### Proof techniques
> >
> > > Overall, the algorithms presented here have noteable overlaps with prior works, and relies on standard proof techniques already established in ML theory.
> >
> > We politely disagree with such a generic point of view. First, on a philosophical level, we would argue that getting new and important results with standard proof techniques is a strength because it means that the authors found a way to combine these techniques in the **correct** way. In mathematics, it is common that simplicity of a proof is valued rather than penalized. Next, we already stated in our response the novelties in our analysis, including the detailed differences from previous works that are also explicitly discussed in our paper. If the reviewer disagrees with any of the novelties we claim, they can precisely describe why; otherwise, it is not clear what the criticism is.
> >
> > We are concerned that such a generic statement as “relies on standard proof techniques” can be used to reject almost all the **published** papers at ML conferences because it is such a **subjective** remark.

---

### Official Review · Reviewer_P51Q · 2023-11-10

**Soundness:** 3 good
**Presentation:** 3 good
**Contribution:** 2 fair
**Rating:** 6
**Confidence:** 3

**Summary:**

The paper clearly presents two algorithms for monotone-inclusion problems in different conditions with the new analysis that improves oracle complexity.  In the cocoercive setting, the paper uses a single loop Halpern iteration with variance reduction to achieve the oracle complexity of O(n+ \sqrt{n}L/\epsion). In the Lipschitz case, the author uses the inexact Halpern iteration and computes the resolvent approximation by VR-FoRB method. It achieves the oracle complexity of $O(n+\sqrt{n}L/\mu)$.

**Strengths:**

1. The paper proposed two algorithms in the cocoercive case and the monotone Lipschitz case, respectively.
2. The new algorithms improve oracle complexity by a factor of $\sqrt{n}$ compared with existing methods on some conditions.
3. Numerical experiments are presented to further show the improvement of the new algorithms.
4. The paper is clearly written and easy to follow.

**Weaknesses:**

1. Some concepts. E.g., monotonicity, maximal monotone, are not explicitly defined in the paper, which slightly impairs the completeness of the paper.
2. Both Algorithms 1 and 3 are variants of existing algorithms. The Algorithm 1 is a simpler version of Cai et al. (2022a), while there is not enough comparison to present the novelty and advantage of the new algorithm. The Algorithm 2 is a combination of inexact Halpern iteration and VR-FoRB (Alacaoglu & Malitsky (2022)), which still doesn’t present much novelty. Although there are improvements on the oracle complexity bound, it seems that they mainly come from the analysis side and assumptions side instead of the algorithm side. Also, it would be good to discuss whether the improvements stem from specific assumptions.
3. The new oracle complexity bound has an additional term of $n$. It may be beneficial to discuss why the new analysis introducing this term.

**Questions:**

1. Can the author propose a framework to unify the two algorithms? Two different algorithms for two different conditions are not concise enough.
2. Can the author describe the experiments more clearly? Like providing more details of matrix games and other tasks?

---

> ### Author Response · Authors · 2023-11-15
> **Rebuttal by Authors (1/3)**
>
> **Thanks for the questions you raised. We hope that in light of clarifications provided below you would consider reevaluating our work. Please let us know if there is any additional information you would find helpful.**
>
> ---
>
> #### **Strengths**
>
> > In the Lipschitz case, the author uses the inexact Halpern iteration and computes the resolvent approximation by VR-FoRB method. It achieves the oracle complexity of $O(n + \sqrt{n}L/\mu)$.
>
> We would like to point out that our main contribution in the monotone Lipschitz setting is to provide the near-optimal complexity $\widetilde O(n + \sqrt{n}L/\epsilon)$. The complexity result $O(n + \sqrt{n}L/\mu)$ the reviewer mentioned is for VR-FoRB as a byproduct of our main results, which constitutes another minor contribution of this paper to provide non-asymptotic rates for a single-call method for strongly monotone inclusions.
>
> ---
>
> #### **Weaknesses**
>
> > Some concepts. E.g., monotonicity, maximal monotone, are not explicitly defined in the paper, which slightly impairs the completeness of the paper.
>
> Monotonicity and maximal monotonicity are explicitly defined in the first paragraph of Section 2, which we also highlight in the updated revision. These are standard definitions that can be found virtually in all the papers solving monotone variational inequality problems. Could the reviewer please let us know what other concept they believe is not explicitly defined or what is not “explicit” in our definition? We would be happy to provide further pointers or explanations.
>
> > The Algorithm 1 is a simpler version of Cai et al. (2022a), while there is not enough comparison to present the novelty and advantage of the new algorithm.
>
> Please see **General Response** for our main contribution: first-of-its-kind complexity results for convex-concave min-max problems and more general problems.
> - The paragraph after Algorithm 1 provides the precise comparison between this algorithm and the one of Cai et al. (2022a), from which the reviewer quoted the first sentence out of context. One may note that Algorithm 1 takes completely different choices including but not limited to batch sizes, batch sampling strategy and the probability $p_k$ of full operator evaluation. Further, we use a new potential function and a completely different analysis to obtain the best-known complexity in this problem setting whereas the algorithm in Cai et al. (2022a) is suboptimal. Our result also has implications on best known complexity in finite-sum minimization (discussed in the last paragraph of Section 3).
> - The techniques we use are quite distinct from the infinite-sum version (Cai et al., 2022a), even in the cocoercive case. Cai et al., 2022a handles the variance reduction by separating the variance term, and their analysis heavily depends on using large batches (depending on $\varepsilon$) to ensure $\mathbb{E}[\|F(u_k) - \tilde{F}(u_k)\|^2] = \mathcal{O}(\epsilon^2/k)$. It is obvious that one cannot obtain such a variance bound in the finite-sum setting. Instead, we include the variance in our potential function and show that it decreases without batch sizes depending on $\varepsilon$ but only depending on $\sqrt{n}$, by using the structure of finite-sum and taking different $p_k$ (see Alg.1 and Lemma B.1 in Appendix B) than the original PAGE estimator. In the monotone case, our Algorithm 2 is completely different from the one in Cai et al., 2022a, and our result further handles the constrained setting. Additionally, while the complexity results we obtain are near-optimal (optimal up to a log factor) in the finite-sum setting we consider, the result of Cai et al., 2022a for the infinite-sum setting is *suboptimal* (their oracle complexity is $1/\epsilon^3$, whereas the optimal oracle complexity for the infinite sum setting is $1/\epsilon^2$). It would be highly unusual that transferring results from one domain (infinite-sum) to another (finite-sum) would suddenly change the results from highly suboptimal to (near-)optimal.

---

> ### Author Response · Authors · 2023-11-15
> **Rebuttal by Authors (2/3)**
>
> > Algorithm 2 is a combination of inexact Halpern iteration and VR-FoRB (Alacaoglu & Malitsky (2022)), which still doesn’t present much novelty.
>
> Please see **General Response** for our main contribution: first-of-its-kind complexity results for convex-concave min-max problems and more general problems.
>
> Again, the paragraph after Algorithm 2 precisely describes the important differences in our algorithmic approach compared to the deterministic inexact approach of Diakonikolas (2020). Briefly:
> - Our Algorithm 2 is designed for a more general composite problem setting and takes a completely different inexactness criterion for the stopping criterion of inner loops, provides a non-trivial extension from deterministic inexact Halpern iteration to a stochastic one. This is due to the difficulty of using stopping criteria in stochastic algorithms, please also see **General Response**.
> - We adopt the proper-chosen stepsize $\eta$ in the resolvent (backward step) (which comes from our analysis) to obtain the desired complexity, in contrast to the $\eta \equiv 1$ in the inexact Halpern iteration from Diakonikolas (2020).
> - Already explicitly discussed in the paragraph after Theorem 4.6, only almost sure convergence is proved for VR-FoRB in Alacaoglu & Malitsky (2022), while we show its non-asymptotic rate for the first time and this constitutes another (although not major) contribution of our paper.
>
> > Although there are improvements on the oracle complexity bound, it seems that they mainly come from the analysis side and assumptions side instead of the algorithm side.
>
> We disagree with this comment. We believe that there is as much value in proving interesting theoretical results about existing algorithms as there is in proposing new algorithms. With the same logic, does the reviewer think that all the papers providing new analysis for SGD, accelerated gradient descent (AGD) or SVRG should be rejected? We (and the ML community, judging by the number of ML conference papers focused on analyzing these existing algorithms) definitely do not agree with this since the new insights on existing algorithms have provided and continue to provide invaluable knowledge.
>
> Also, as clarified in our responses to the reviewer’s other points, our proposed algorithms differ from prior works on subtle but important aspects and we present the first (near-)optimal oracle complexity for the general finite-sum monotone inclusion problem settings considered in the paper.
>
> > Also, it would be good to discuss whether the improvements stem from specific assumptions.
>
> Our assumptions are **exactly the same** as those in the existing literature on variance reduced algorithms for min-max problems. Please see our **General Response** for precise references to these works.
>
> Hence our improvements compared to other variance reduced algorithms do **not** stem from specific assumptions.
>
> All the references that the reviewer pointed out have the exact same assumption where we provide pointers:
>
> - Assumption 1(iv) in Alacaoglu and Malisky (2022),
> - (C) in Section 2 of Palaniappan & Bach (2016) where each operator is assumed to be $L_i$-Lipschitz which satisfies our assumption with $L=\mathbb{E}_{i\sim Q} L_i$,
> - Sec 5.4 in Carmon et al. (2019) where they discuss convex-concave problems, they assume that each function has $L_k$ Lipschitz gradients, which maps to our assumption the same way as Palaniappan & Bach (2016),
> - Def 2.2 in Han et al., (2021) that studied lower bounds for convex-concave problems.
>
> On the other hand, it is worth noting that both our work and all the existing variance reduced algorithms (see the above references) do use a stronger assumption compared to deterministic algorithms. In particular, deterministic algorithms use Lipschitzness of the full operator whereas variance reduced algorithms use Lipschitzness of the components. As shown in the work of Han et al., (2021) under Lipschitzness of the full operator, deterministic operators have optimal complexity and with the additional assumption on expected Lipschitzness, our complexity is also optimal.
>
> [Han et al., (2021)] Yuze Han, Guangzeng Xie, Zhihua Zhang. “Lower Complexity Bounds of Finite-Sum Optimization Problems: The Results and Construction.” arXiv preprint arXiv:2103.08280, 2021.

---

> > ### Author Response · Authors · 2023-11-15
> > **Rebuttal by Authors (3/3)**
> >
> > > Can the author propose a framework to unify the two algorithms? Two different algorithms for two different conditions are not concise enough.
> >
> > Our Algorithm 2 can also be seen as a “unified algorithm” to get the improved complexity bound for both cases. This is because cocoercivity directly implies Lipschitzness and monotonicity. We provide Algorithm 1 since it is simpler: a direct way (single loop) to get this result in the cocoercive case. We respectfully disagree with “not concise enough” as a reason for rejection since we only have two algorithms in the paper and “concise enough” is a highly subjective concept in this context.
> >
> > > Can the author describe the experiments more clearly? Like providing more details of matrix games and other tasks?
> >
> > We have defined the matrix game and the quadratic program as well as the important setups in the second paragraph of Section 5. More experimental details such as the data matrices are provided in Appendix D due to space constraints, which is also explicitly referred to in the last sentence of our second paragraph in Section 5. Can the reviewer please clarify what additional details they think are needed?
> >
> > > The new oracle complexity bound has an additional term of $n$. It may be beneficial to discuss why the new analysis introducing this term.
> >
> > It is not clear to us which $n$ term the reviewer means. However, if it is the additional $n$ in $\widetilde{O}(n+\sqrt{n}L/\epsilon)$, it is because our algorithm needs to compute the full gradient in the beginning of the run of the algorithm. This term is extremely prevalent in variance reduction both for min-max or minimization problems.
> >
> > (Please see our paper for other citations.)

---

### Author Response · Authors · 2023-11-15
**General Response (1/3)**

Dear reviewers and AC,

Thank you for your time reviewing our paper. We have uploaded a revision, where the main change is Table 1 added in Appendix A which we refer to in the introduction for a clear comparison of our results and assumptions with the state-of-the-art in the monotone Lipschitz settings, addition of the definition of graph of an operator as requested by **Reviewer wjm9**, and derivation of Lipschitz constants as requested by **Reviewer 9YKM**. We will move Table 1 to Section 1 by reorganizing our content for the page limit in later revisions.

We have highlighted major parts of the revision in red for easy search. For reviewers’ convenience, we have also included Table 1 in this general response.

---

First of all, we would like to restate the following strengths of our paper:
- All the reviewers mentioned that we provide improved and new oracle complexity results. In fact, our results are the first optimal (up to a log factor) complexities achieved for a computable optimality measure (operator residual) for convex-concave min-max problems and more generally finite-sum monotone inclusion problems.
- Two out of three reviewers agreed that our paper is well written and easy to follow, with scores Soundness 3 and Presentation 3.
    - “The paper is well-written and explains every detail of the algorithms and their contributions. The discussions and comparisons to previous works clearly reflect their differences and improvements.” (**Reviewer 9YKM**)
    - “The paper is clearly written and easy to follow.” (**Reviewer P51Q**)

---

Moreover, to prevent possible misunderstandings of our work, we provide further clarifications of our assumptions and techniques here.

**[Assumptions]** All the assumptions made in this paper are standard and prevalently used in previous works on variance reduction. Hence, unlike **Reviewers wjm9, P51Q** **incorrectly** claim, the improvements in our paper are **not** due to stronger assumptions. We provide precise pointers to existing variance reduced methods:
- Expected Lipschitzness (Assumption 2) trivially holds when each component operator is Lipschitz (for comparison in minimization, this is similar to the common assumption that each objective is smooth). Our assumption is either the same or weaker than:
    - Assumption 1(iv) in Alacaoglu and Malisky (2022),
    - (C) in Section 2 of Palaniappan & Bach (2016) where each operator is assumed to be $L_i$-Lipschitz, which satisfies our assumption with $L=\mathbb{E}_{i \sim Q} L_i$,
    - Section 5.4 in Carmon et al. (2019) where they discuss convex-concave problems. They assume that each function has $L_k$ Lipschitz gradients, which maps to our assumption the same way as Palaniappan & Bach (2016).
- Average Cocoercivity (Assumption 3) holds trivially when each component operator is cocoercive (for comparison in minimization, this is similar to the common assumption that each objective is smooth and convex). Our assumption is either the same or weaker than:
    - Assumption 3.1 in Morin & Giselsson (2022),
    - Sec 3.2 in Davis (2023).

Hence, as the response to **Reviewer wjm9** and **P51Q**, the improvements compared to existing variance reduction schemes are definitely **not** due to extra assumptions on the template but due to our new analysis. The exact same assumptions, as we provided precise references above, were made in the related work on variance reduction.

---

> ### Author Response · Authors · 2023-11-15
> **General Response (2/3)**
>
> **[Techniques]** As discussed in Sections 3 and 4 in the paper, our paper presents algorithms with many new techniques adding to the previous works.
> - (Cocoercive) To take advantage of the finite-sum structure and obtain the improved complexity, we adopt a two-stage full operator evaluation strategy for the PAGE estimator and design a new potential function (see Alg. 1, Lemma 3.1 and Lemma B.1) to prove the contraction of the potential function in expectation. These are obviously different from traditional recursive variance reduction literature and the more relevant work of Cai et al. (2022a), where they used a different potential function and needed a more complicated induction-based argument to get a complexity result that is **suboptimal** for the finite-sum case.
> - (Monotone)
>     - Compared to the deterministic inexact Halpern iteration with unit backward stepsize (stepsize for the resolvent) in Diakonikolas (2020), we propose a new inexact scheme that handles the stochastic error and we adopt the properly chosen (from our analysis) backward stepsize for improving complexity.
>     - To deal with the well-known issues with using stopping criteria in stochastic settings (not computable since the guarantees are given in expectation), we set the inexactness criterion such that the required level of inexactness is **guaranteed** after a **computable** number of inner iterations which only depends on $L, n, k$. This sidesteps the need to check a stopping criterion since our analysis **proves** that after running the inner loop for the given number of iterations, the required stochastic inexactness will hold.
>     - This, as a byproduct, also gives us an **any-time** algorithm where the final accuracy is not set in advance, differing from the work of Diakonikolas (2020) that focused on deterministic algorithms.
>     - Another byproduct of our analysis and a contribution of our work is the non-asymptotic linear convergence rate for the single-call method VR-FoRB under strong monotonicity. Note that in Alacaoglu & Malitsky (2022) only almost-sure convergence was proved.
>     - An important corollary of our result, due to our focus on deriving guarantees on the computable operator residual is that we show **the first optimal complexity results that hold with high probability** (depending logarithmically on the confidence level) for both the gap and operator residual measures. Such a result was not possible with the previous variance reduced algorithms that made the same assumptions as us.
>
> **We hope that our explanations above and the individual responses we provide below help clarify the misunderstandings of the reviewers on key aspects of our paper.
> We look forward to interacting with the reviewers and providing any further explanations as necessary.**
>
>
> Sincerely,
>
> Authors
>
>
> (Please refer to our paper for the citations.)

---

> > ### Author Response · Authors · 2023-11-15
> > **General Response (3/3)**
> >
> > ## Table: Comparison of Our Results with State-of-the-Art in Monotone Lipschitz Settings
> >
> > Refer to Introduction Section for the discussion regarding the difference in optimality measures $\mathrm{Res}_{F+G}$ and $\mathrm{Gap}$ and the importance of getting results on residue guarantees.
> >
> >
> > | Paper                         | Complexity for $\mathrm{Res}_{F+G}$                         | Complexity for $\mathrm{Gap}$                         | Assumption | High Probability Result |
> > |-------------------------------|------------------------------------------------------------|-------------------------------------------------------|------------|-------------------------|
> > | KOVALEV & GASNIKOV (2022)     | $\mathcal{O}\big(\frac{nL_F}{\epsilon}\big)$                         | $\mathcal{O}\big(\frac{nL_F}{\epsilon}\big)$                  | Assumption 1 | N/A                   |
> > | NEMIROVSKI (2004)             | $\mathcal{O}\big(\frac{nL_F^2}{\epsilon^2}\big)$                     | $\mathcal{O}\big(\frac{nL_F}{\epsilon}\big)$                  | Assumption 1 | N/A                   |
> > | CAI ET AL. (2022A)            | $\mathcal{O}\big(\frac{\sigma^2L}{\epsilon^3} + \frac{L^3}{\epsilon^3}\big)$ | $\mathcal{O}\big(\frac{\sigma^2L}{\epsilon^3} + \frac{L^3}{\epsilon^3}\big)$ | Assumption 1, 2, $G \equiv \mathbf{0}$, $\mathbb{E}_i \lVert F_i(x)-F(x) \rVert^2 \leq \sigma^2$ | - |
> > | LUO ET AL. (2021)             | $\widetilde{\mathcal{O}}\big(\frac{\sigma^2}{\epsilon^2} + \frac{L_F}{\epsilon}\big)$ | $\widetilde{\mathcal{O}}\big(\frac{\sigma^2}{\epsilon^2} + \frac{L_F}{\epsilon}\big)$| Assumption 1, $G \equiv \mathbf{0}$, $F = \binom{\nabla_x \Phi(x, y)}{-\nabla_y \Phi(x, y)}$, $\mathbb{E}_i \lVert F_i(x)-F(x)\rVert^2 \leq \sigma^2$ | - |
> > | CARMON ET AL. (2019)         | -                                                          | $\widetilde {\mathcal{O}}\big(n + \frac{\sqrt{n}L}{\epsilon}\big)$ | Assumption 1, 2, Bounded Domain, cf. Sec 5.4 in CARMON ET AL. (2019) | - |
> > | PALANIAPPAN & BACH (2016)     | -                                                          | $\widetilde {\mathcal{O}}\big(n + \frac{\sqrt{n}L}{\epsilon}\big)$ | Assumption 1, 2, Bounded Domain, cf. (C) in Sec. 2 in PALANIAPPAN & BACH (2016) | - |
> > | ALACAOGLU & MALITSKY (2016)   | -                                                          | $ \mathcal{O}\big(n + \frac{\sqrt{n}L}{\epsilon}\big)$        | Assumption 1, 2, cf. Assumption 1(iv) in ALACAOGLU & MALITSKY (2016) | - |
> > | **[Ours]**                    | $\widetilde {\mathcal{O}}\big(n + \frac{\sqrt{n}L}{\epsilon}\big)$     | $\widetilde {\mathcal{O}}\big(n + \frac{\sqrt{n}L}{\epsilon}\big)$| Assumption 1, 2 | $\checkmark$         |

---

### Author Response · Authors · 2023-11-20
**Reminder on the rebuttal**

Dear reviewers,

We would like to kindly remind you of the upcoming deadline for discussions -- Nov 22nd. We believe that much of the criticism of our work comes from a misunderstanding that we hope to have clarified in our response. Could you please let us know if our response addresses your concerns, or if there is any other information you would find useful?

Regards,

Authors

---

### Meta-Review · Area_Chair_G56K · 2023-12-06

**Metareview:**

The paper studies finite-sum monotone inclusion problems and presents variants of the classical Halpern iteration that utilize variance reduction, leading to enhanced complexity guarantees. After author response and reviewer discussion, the paper receives generally unanimous support from the reviewers. Thus, I recommend acceptance.

**Justification For Why Not Higher Score:**

The reviewers are not super excited about the novelty of the work.

**Justification For Why Not Lower Score:**

The paper receives generally unanimous (weak) support from the reviewers.

---

### Decision · Program_Chairs · 2024-01-16

Accept (poster)